# CollabBench: Benchmarking and Unleashing Collaborative Ability of LLMs with Diverse Players via Proactive Engagement

Hong Qian [1 2]   Yuanhao Liu [1 3]   Zihan Zhou [1]   Zongbao Zhang [1 2]   Hanjie Ge [1]   Haotian Shi [1]   Liang Dou [1]
Xiangfeng Wang [1]   Jingwen Yang [* 3]   Aimin Zhou [1 2]

## Abstract

While LLM-based agents excel at individual tasks, effective collaboration with realistic human partners remains challenging. Most of the existing conversation-level collaborative studies lack grounded interaction and behavioral execution, motivating the need for cooperative game environments that enable contextualized and immersive collaboration. To this end, this paper proposes CollabBench, a benchmark for evaluating and training collaborative agents in cooperative games. CollabBench features a Diverse Player Profile Simulation pipeline to model varied players behaviors, and a Collaborative Agentic Training paradigm that unifies reasoning, communication, and action via agentic rollouts, optimized with a hybrid reward balancing task efficiency and affective adaptation. We further extend classic environments to CWAH-MultiPlayer and Cook-MultiPlayer for systematic evaluation under diverse personalities. Experiments with efficiency and affective metrics show that our trained models outperform base models, achieving 19.5% higher efficiency and 24.4% improved affective performance. Further analysis reveals key collaborative limitations of existing models and offers insights for future collaborative training.

## 1. Introduction

With the rapid advancement of Large Language Models (LLMs), LLM-based agents (Wang et al., 2024b; Xi et al., 2025a; Zhang et al., 2026) have demonstrated strong capabilities in individual tasks such as deep research, mathematical reasoning, and coding. As these abilities expand, a critical research challenge is enabling agents to collaborate effectively with human partners exhibiting diverse personalities and behavioral patterns. Most existing human–agent collaboration studies focus on conversation-level tasks (Wang et al., 2024d; Wu et al., 2025; Zhang et al., 2025a), such as dialogue, document editing, and mathematical problem solving, which typically exhibit weak context and interaction, often detaching communication from the immediate shared situation and lacking grounded behavioral execution. This motivates the development of agents operating in immersive cooperative game environments (Hu et al., 2024) for context scaling (Wang et al., 2026), requiring agents to share a grounded environment and engage in proactive interaction beyond purely linguistic exchange, thereby unleashing agent's contextualized collaborative intelligence in both efficient task execution and affectively adaptive engagement with diverse human partners.

Despite growing interest, several challenges remain. First, simulating players with diverse personalities and behavioral styles through LLMs in game environments remains difficult (*Challenge 1*). Existing role-playing and user profiling approaches (Wang et al., 2024c; Tu et al., 2024; Wang et al., 2025a; Salemi et al., 2024) largely rely on predefined characters or static user data, and fail to capture grounded, action-level behaviors in interactive games. Second, unleashing LLM-based agents collaboration with diverse partners in strong context such as game environments is largely unexplored (*Challenge 2*). Prior work predominantly focuses on improving single-agent capabilities (Feng et al., 2025; Wang et al., 2025b; Xi et al., 2026) or multi-agent architectures (Zhang et al., 2024b;a; Seo et al., 2025), with limited emphasis on learning collaborative awareness and adaptability within LLMs. Third, current criteria for LLM-based agents in game environments (Costarelli et al., 2024; Xi et al., 2025b; Hu et al., 2026) are often restricted to efficiency-based metrics, such as final game scores (*Challenge 3*), overlooking affective and social qualities that are essential for effective human-agent collaboration.

Therefore, this paper introduces CollabBench (CB), a bench-

---

[1]Shanghai Institute of AI for Education, and School of Computer Science and Technology, East China Normal University, Shanghai, China [2]Shanghai Innovation Institute, Shanghai, China [3]Tencent Inc., Shenzhen, China. Correspondence to: Jingwen Yang <jingwenyang@tencent.com>.

*Proceedings of the 43rd International Conference on Machine Learning*, Seoul, South Korea. PMLR 306, 2026. Copyright 2026 by the author(s).

**(a) CB-Efficiency Challenge**

Compared to the default game environment, our **personality-driven** setting introduces **additional collaboration difficulty** in task performance.

**Personality-driven Game Environment**

- CWAH-MultiPlayer (Less is Better)
- Cook-MultiPlayer (More is Better)

**(b) CB-Affective Challenge**

**Default Game Environment**

$P_{sim}$ — Lack of Profiles — Homogenization ⟷ Game Goal $G$ ⟷ Focus on Efficiency — Weak Interactivity — $P_{target}$

I'll **check** the kitchen cabinets for pudding, juice, and cupcakes next. **Can you search** the bathroom? **Let me know** if you find anything.

I have the pudding. **Can you grab** the pudding and the other juice and start taking them to the coffeetable? **I'll find** the remaining items.

**I have** juice here, but I still need two cupcakes. **Please put** the items you have on the coffeetable, and **I'll go check** the kitchen cabinets to see if there are any cupcakes.

I **found** all goal items here except we need one more cupcake. I'll **check** the stove and microwave next. **Can you take** your items to the coffeetable in the livingroom?

**Action** [walktowards] <kitchen> ⟵ ⟶ [walktowards] <stove> **Action**

Focus on tasks, Command-based, Lack of empathetic feedback
Low Helpfulness · Low Trustfulness · Low Empathy

**Personality-driven Game Environment**

$P_{sim}$ — Diverse Profiles — Diversity ⟷ Game Goal $G$ ⟷ Efficiency $R_s(r \mid G)$ Affective $R_a(t_t \mid P_{sim})$ — Optimization — $P_{target}$

Hi Alice, I'm in the livingroom here. **I'm worried** I might choose wrong and waste time. **What do you think** I should do? Maybe you have a good idea?

Of course, Bob! I **completely understand your anxiety**. I suggest you **explore the kitchen first**, as it is **likely to contain the items we need**. I'm in the bathroom, and I'll **keep you updated** if I find anything.

**Thank you** for your advice. I'll go to the kitchen to look for the items, but I'm still **anxious about our progress**. Could you go to the bedroom and see if there's anything we need there?

Of course, Bob! **I'll go to the bedroom to check the cabinet** to look for the goal items. Please **focus on finding** the item we're looking for and **placing it on the coffeetable**. **I believe in you!**

**Action** [walktowards] <kitchen> ⟷ [walktowards] <bedroom> **Action**

High Anxiety, Indecisive behavior pattern, Often sending messages
High Helpfulness · High Trustfulness · High Empathy

*Figure 1.* The motivation of this work: challenges of collaboration between real human and agent. **(a) CB-Efficiency Challenge**: Introducing diverse players significantly increases task difficulty. The Personality-driven setting (red) degrades performance compared to standard homogeneous multi-agent cooperation (blue), resulting in more steps in CWAH (+9.3%) and lower scores in Overcook (-23.1%). **(b) CB-Affective Challenge**: The Default environment (left) prioritizes efficiency, leading to command-based, low-empathy interactions. In contrast, the Personality-driven environment (right) introduces diverse teammates, requiring the agent to optimize both efficiency and affective engagement. Detailed results on motivation are in Appendix A.

mark for systematically evaluating and training LLM-based agents to collaborate proactively with diverse players. To address Challenge 1, we develop a Diverse Players Profiles Simulation pipeline that generates heterogeneous behaviors grounded in Big Five personality theory, followed by high-fidelity profile modeling that enforces explicit personality-behavior mappings and applies interaction-based filtering to ensure realism and consistency. Based on this environment, we observe that player heterogeneity substantially increases task difficulty compared to standard homogeneous multi-agent settings (Figure 1(a)), and that efficiency-driven interaction strategies fail to satisfy affective needs of diverse partners (Figure 1(b)). Motivated by these findings, we propose Collaborative Agentic Training to address Challenge 2, featuring a unified agentic rollout that integrates reasoning, communication, and action, together with a hybrid reward mechanism that jointly optimizes trajectory-level task efficiency and step-level affective alignment via agentic RL.

Building on our diverse players profiles simulation, we extend two classic environments to construct CWAH-MultiPlayer and Cook-MultiPlayer for both training and evaluation, introducing substantially more challenging and realistic collaborative settings with heterogeneous players. To address Challenge 3, we design a comprehensive evaluation protocol combining efficiency metrics (e.g., game score, variance and token usage) with affective metrics (Helpfulness, Trustfulness, and Empathy). Empirical results show that our collaborative training paradigm yields substantial gains, improving average game scores and affective metrics by 19.5% and 24.4%, respectively, over the base model. We further provide detailed analyses revealing the collaborative

limitations of existing LLMs and offer extensive experimental insights into the efficacy of collaborative training.

## 2. Related Work

**Personality-Aware Human Simulation.** Traditional human simulators are learned from collected behavior data (Carroll et al., 2019). Recent work shows that LLMs can flexibly generate diverse personas, and has applied this capability to role-playing and personalized services using predefined character descriptions or user profiles (Kosinski, 2024; Salminen et al., 2024; Shin et al., 2024; Wang et al., 2024c; Tu et al., 2024; Wang et al., 2025a; Salemi et al., 2024). However, such approaches are limited for cooperative game environments, where data for diverse, grounded behaviors is scarce. Existing human-LLM collaboration systems rely on task-driven, conversation-level simulators with limited personality diversity (Wu et al., 2025), while domain-specific LLM-based simulations in dialogue, social activities, and education (Wang et al., 2024d; Zhang et al., 2025a; Park et al., 2023; Liu et al., 2024) do not readily transfer to cooperative game tasks.

**Agentic Training for LLMs in Games.** Recent work has demonstrated strong performance of agentic training for LLMs in domains such as deep research, mathematical reasoning, and coding (Jin et al., 2025; Guo et al., 2025; Jiang et al., 2026). Game environments provide a practical and realistic testbed for this paradigm (Chevalier-Boisvert et al., 2019; Carroll et al., 2019; Shridhar et al., 2021; Puig et al., 2021; Wang et al., 2024a). Prior studies predominantly focus on enhancing single-agent capabilities through

architecture design or agentic reinforcement learning, including VOYAGER (Wang et al., 2024a), RAGEN (Wang et al., 2025b), and AgentGym-RL (Xi et al., 2026). Research on collaborative agents remains largely architecture-centric, as exemplified by ProAgent (Zhang et al., 2024a) and CoELA (Zhang et al., 2024b), which improve coordination via specialized modules. However, these approaches generally lack explicit mechanisms for training collaboration awareness within LLMs and overlook the diversity of teammate personalities in real-world collaboration.

**Evaluation of LLM-based Agent and Human Collaboration.** As LLM capabilities advance, effective collaboration has become a critical objective, where efficiency alone is insufficient as an evaluation criterion (George, 1990; McAllister, 1995; Salas et al., 2005). Recent work has begun to adopt multidimensional evaluation frameworks, such as LLM-as-a-judge for assessing interactivity (Wu et al., 2025) and empathy (Zhang et al., 2025a). However, gamified evaluations of LLM-based agents still predominantly rely on efficiency-oriented metrics, including game scores and success rates (Costarelli et al., 2024; Hu et al., 2026; Sun et al., 2025; Zhang et al., 2025b). Although some cooperative game studies incorporate affective aspects via user surveys (Siu et al., 2021; Zhang et al., 2024b), the lack of systematic, scalable evaluation protocols limits large-scale benchmarking and complicates reward design for agentic reinforcement learning. Detailed related work can be found in the Appendix B.

# 3. Problem Formulation

Distinct from traditional homogeneous, performance-driven multi-agent game systems, we study a heterogeneous collaboration setting in which two agents cooperate to achieve a shared goal $G = \{g_1, \ldots, g_K\}$ in a partially observable environment, where $K$ means the number of subgoals in a task. One agent acts as a simulated player $P_{sim}$ that exhibits diverse human personalities and behaviors, while the other is a target collaborative agent $P_{target}$ whose policy is optimized. Interaction is represented by a trajectory $\tau = \{t_1, \ldots, t_H\}$, where $H$ means the number of turns in a trajectory and each turn $t_i = \{s_i, r_i, c_i, a_i\}$ involves a partial environment observation $s_i$, internal reasoning $r_i$, natural language communication $c_i$ and executable actions $a_i$. To account for partner heterogeneity, $P_{target}$ is optimized with respect to two objectives: a trajectory-level efficiency reward $R_e(\tau|G) = score(\tau, G)$, measuring overall goal completion, and a step-level affective reward $R_a(t_i|P_{sim})$ capturing alignment with the simulated players profile. The overall objective $R^* \simeq R_e + R_a$ balances task efficiency with per-step player affective experience.

# 4. CollabBench: Benchmark, Method and Evaluation

## 4.1. Diverse Players Profiles Simulation

To enable $P_{target}$ to acquire collaborative competence in real-world scenarios, it is essential to simulate diverse players $P_{sim}$. However, cooperative game environments pose two key challenges: (1) the scarcity of real human gameplay trajectories limits the coverage and fidelity of teammate personality distributions, and (2) bridging abstract personality representations with concrete, game-specific behaviors across diverse games is inherently difficult. These challenges motivate a unified pipeline that converts personality distributions into executable behaviors for scalable, game-adaptive player simulation.

### 4.1.1. DIVERSE SIMULATED PLAYERS TRAJECTORY DATA CONSTRUCTION

**Personality-Driven Players Profiles Construction.** Simulating teammates with diverse personalities and behavioral styles in game environments is challenging due to the need for continuous, authentic interactions and the high cost and limited coverage of real player trajectories. To address this, we adopt the widely recognized Big Five personality theory (McCrae & Costa Jr, 1999) and assign low, medium, or high levels to each dimension in prompts to ensure broad behavioral coverage. Furthermore, each trait is grounded in expert-validated gameplay logic, allowing personality differences to consistently manifest as observable behavioral patterns during interaction. These personality descriptors are used solely to induce behavioral diversity during data generation, rather than serving as final player profiles.

**Game-Specific Behavioral Trajectory Generation.** To mitigate personality bias induced by a single LLM and enhance trajectory diversity, we employ multiple LLMs as diverse players $P_{sim}$ instantiated from our personality-driven player profiles. These agents interact with diverse game environments via a ReAct-style mechanism, producing trajectories that include both internal reasoning and executable actions. Notably, the reasoning traces are critical for extracting high-quality behavioral patterns. This process yields a highly diverse behavior database, providing a solid foundation for constructing and validating mappings between personality profiles and behavioral patterns.

### 4.1.2. HIGH-FIDELITY PROFILE MODELING

Although diverse behavioral trajectories support fine-grained modeling of players diversity, they are generated solely from personality-based prompts and lack explicit alignment between personality traits and observed behaviors in specific games, which may introduce inconsistencies (a player with low openness frequently sending messages).

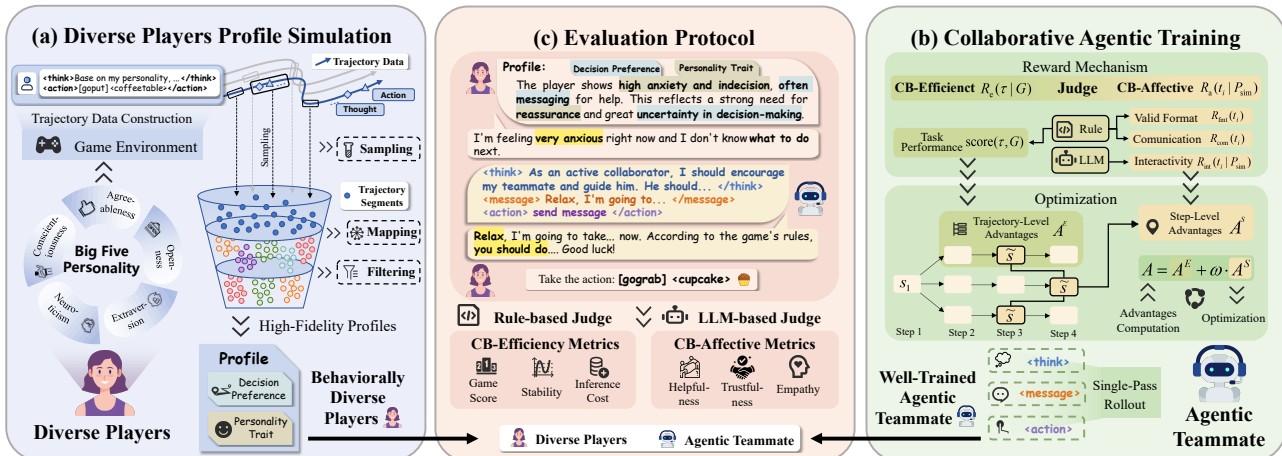

*Figure 2.* The framework of the proposed CollabBench. **(a) Diverse Players Profiles Simulation**: A pipeline that constructs high-fidelity, behaviorally diverse player profiles ($P_{sim}$). **(b) Collaborative Agentic Training** : A training paradigm for the target collaborative agent ($P_{target}$). **(c) Evaluation Protocol**: A comprehensive framework that assesses the interaction quality between the target collaborative agent ($P_{target}$) and diverse simulated teammates ($P_{sim}$) using both efficiency and affective metrics.

We therefore construct a high-quality mapping and apply rigorous filtering to obtain high-fidelity player profiles.

**Personality-Behavior Mapping Construction.** To construct personality-behavior mappings, we first encode collected textual trajectory segments and cluster the LLM-based embeddings to identify similar behavior patterns. For each cluster, an LLM summarizes the associated personality traits, reasoning content, and actions to produce a unified mapping that captures thinking patterns and action preferences. We then filter redundant mappings to ensure diversity, yielding the final high-fidelity player profiles. The clustering process and the prompts for summarizing profiles are detailed in Appendix C.1 and Appendix E.1, respectively.

**Interaction-Based Filtering.** We hypothesize that a high-fidelity player profile requires not only a well-constructed mapping between personality traits and behaviors, but also that when the profile serves as a prompt to a capable LLM, the reasoning and actions faithfully reflect its encoded personality traits and behavioral patterns. Therefore, we evaluate inconsistencies by deploying these profiles to drive player agents through ReAct-style game interactions, and apply two dimensions of criteria to further filter the profiles:

• **Personality-Reasoning Consistency**: The reasoning content of the player agent should accurately reflect the personality traits described in the corresponding profile.

• **Reasoning-Action Consistency**: The reasoning processes of the player agent during interactions should align with its actual action decisions.

Inspired by (Wang et al., 2025a), we employ a penalty-based LLM judge to automatically detect deviations during interactions. Specifically, an interaction trajectory is segmented

into multiple fixed-size time windows, each serving as the minimal evaluation unit. The judge evaluates each segment and assigns a severity score to each deviation, ranging from 1 (minor) to 5 (severe). The score can be formalized as:

$$ S_\eta = \frac{1}{|\Omega_\eta|} \sum_{i=1}^{|\Omega_\eta|} 5 - \alpha_p \times D_i - \alpha_p^m \times D_i^m + \alpha_r \times L_i, \quad (1) $$

where $\Omega_\eta$ denotes the game set under profile $\eta$, while $D_i$ and $D_i^m$ denote the total and maximum deviation penalties for the $i$-th game, respectively. $L_i$ denotes the number of LLM reasoning steps. $\alpha_p$, $\alpha_p^m$ and $\alpha_r$ are their corresponding penalty and reward coefficients. The final term rewards longer reasoning trajectories to curb deviation accumulation, ensuring fairness across varying interaction lengths. The detailed scoring prompt can be found in the Appendix E.2.

Based on Eq.(1), we compute the personality-reasoning consistency score $S_\eta^{\text{P-R}}$ and reasoning-action consistency score $S_\eta^{\text{R-A}}$ for each profile $\eta$. The final filtering score is defined as $S_\eta^{\text{ALL}} = \beta \times S_\eta^{\text{P-R}} + (1 - \beta) \times S_\eta^{\text{R-A}}$, where $\beta$ is a weighting coefficient. We retain the top-$k$ highest-rated profiles for each game subtask to model diverse players $P_{sim}$ by the final score $S_\eta^{\text{ALL}}$, which are subsequently verified by experts. Additional quantitative validation of filtering is provided in Appendix C.2. And detailed experiments for diversity, robustness and realism of our simulated players profiles are provided in Section 5.2.2 and Appendix D.3.

### 4.2. Collaborative Agentic Training

#### 4.2.1. AGENTIC ROLLOUT

**Environment Input.** To train the collaborative agent $P_{target}$, our environment comprises a game environment

and a set of diverse players $P_{sim}$ as described in Section 4.1. During training, $P_{target}$ interacts and cooperates with players of varying personalities and behavior patterns. Focusing on improving the LLMs capabilities rather than the agent architecture, we adopt the same observation format $s_i$ as in prior work (Zhang et al., 2024a;b).

**Agent Output.** To capture how LLM-based agents reason about collaboration, we adopt a single-pass rollout that unifies reasoning, communication, and action at each interaction step, marked by `<think>`, `<message>`, and `<action>`. Unlike prior segmented approaches (Zhang et al., 2024b), the agent generates all collaboration-relevant outputs simultaneously, including a message at every step to reflect communication reasoning even when final action does not choose to send message. This design reduces token usage and latency, encourages joint reasoning over communication and action, and provides interpretable signals of collaborative intent. Only the `<action>` output is executed in the environment.

### 4.2.2. REWARD MECHANISM

While prior work (Feng et al., 2025; Wang et al., 2025b; Xi et al., 2026) shows that optimizing trajectory-level efficiency rewards improves LLM-based agent performance, collaborative tasks require additional affective capabilities, such as communication willingness, partner awareness, and empathy. These local, context-sensitive skills cannot be captured by sparse trajectory-level rewards alone. We therefore decompose the reward into two components: a trajectory-level efficiency reward for global task performance and a step-level affective reward to guide collaborative behavior at each interaction step.

**Efficiency Reward.** Formally, given a trajectory $\tau$ and a task goal $G$, we define the efficiency reward as a sparse trajectory-level signal

$$R_{\text{e}}(\tau \mid G) = \text{score}(\tau, G), \qquad (2)$$

where $\text{score}(\cdot)$ denotes the environment-specific task performance metric. This reward encourages efficient task completion but provides no direct supervision over intermediate collaborative behaviors.

**Affective Reward.** We define a dense, step-level affective reward to supervise the agents collaborative behaviors at each interaction step. At each time step $t_i \in \tau$, the affective reward is formulated as

$$R_{\text{a}}(t_i \mid P_{\text{sim}}) = R_{\text{fmt}}(t_i) + R_{\text{com}}(t_i) + R_{\text{int}}(t_i \mid P_{\text{sim}}). \quad (3)$$

We define the agents executable action space as $\mathcal{A} = \mathcal{A}_{\text{int}} \cup \mathcal{A}_{\text{com}}$, where $\mathcal{A}_{\text{int}}$ denotes interaction actions and $\mathcal{A}_{\text{com}}$ denotes communication actions (e.g. send message) .

And three reward components can be jointly specified as

$$\begin{cases} R_{\text{fmt}}(t_i) = \mathbb{I}\left[f_{\text{struct}}(t_i) = 1 \wedge a_i \in \mathcal{A}\right] \\ R_{\text{com}}(t_i) = \mathbb{I}\left[a_i \in \mathcal{A}_{\text{com}}\right] \\ R_{\text{int}}(t_i \mid P_{\text{sim}}) = J_{\text{LLM}}(t_i, P_{\text{sim}}) \end{cases} . \quad (4)$$

The format reward $R_{\text{fmt}}(t_i)$ ensures output validity by verifying well-formed and executable responses, including required token structure via $f_{\text{struct}}(\cdot)$ and that the action $a_i \in \mathcal{A}$. The communication reward $R_{\text{com}}(t_i)$ encourages proactive dialogue by incentivizing actions in the communication space $\mathcal{A}_{\text{com}}$. The interactivity reward $R_{\text{int}}(t_i \mid P_{\text{sim}})$ assesses step-level collaboration using an LLM-based judge conditioned on the partner profile $P_{\text{sim}}$, scoring reasoning, messages, and actions on affective criteria such as helpfulness, trustfulness, and empathy, on a normalized $[0, 1]$ scale with ten discrete levels. By combining communication rewards with subjective affective judgments, these designs cross-validate communication frequency and quality, discouraging reward hacking such as excessive but uninformative communication. Details of the LLM judge prompt are provided in Appendix E.3.

### 4.2.3. OPTIMIZATION

To tune the model with structured rewards, we employ GIGPO (Feng et al., 2025), a variant of GRPO (Guo et al., 2025) that introduces a hierarchical advantage estimation structure combined with both trajectory-level and step-level rewards. We sample a group of $N$ trajectories $\{\tau_1, \ldots, \tau_N\}$ and combine a global trajectory-level relative advantage $A^T(\tau_n)$, which captures trajectory-level task efficiency $R_{\text{e}}(\tau_n \mid G_n)$, with a local step-level advantage $A^S(t_i^{(n)})$ derived from fine-grained affective collaborative performance $R_{\text{a}}(t_i^{(n)} \mid P_{\text{sim}}^{(n)})$. This can be formalized as

$$A(t_i^{(n)}) = A^T(\tau_n) + \omega \cdot A^S(t_i^{(n)}), \qquad (5)$$

where $\omega$ balances global task efficiency and local affective interaction quality. This advantage is then used to optimize the clipped policy objective:

$$\mathcal{J}(\theta) = \mathbb{E}_{\tau \sim \pi_{\theta_{\text{old}}}} \left[ \frac{1}{NH} \sum_{n=1}^{N} \sum_{i=1}^{H} \min\left( \rho_\theta\left(t_i^{(n)}\right) A\left(t_i^{(n)}\right), \right. \right.$$
$$\left. \left. \text{clip}\left(\rho_\theta\left(t_i^{(n)}\right), 1 - \epsilon, 1 + \epsilon\right) A\left(t_i^{(n)}\right)\right)\right], \quad (6)$$

where $\pi_\theta(t_i^{(n)})$ represent the current collaborative agent $P_{target}$'s policy, which is calculated over the entire generated sequence $t_i^{(n)}$ by $P_{target}$. And $\rho_\theta(t_i^{(n)}) = \frac{\pi_\theta(t_i^{(n)})}{\pi_{\theta_{\text{old}}}(t_i^{(n)})}$ is the importance sampling ratio.

## 4.3. Evaluation Protocol

Evaluating contextualized collaborative intelligence requires a holistic view beyond task performance. Grounded in organizational psychology and human-AI interaction theories (George, 1990; McAllister, 1995; Salas et al., 2005), we define collaboration along two dimensions: *Taskwork* (operational efficiency) and *Teamwork* (interpersonal interaction quality). Accordingly, CollabBench uses two major metrics with several submetrics, CB-Efficiency and CB-Affective, to assess whether agents balance efficient execution with affectively proactive collaboration across diverse partners.

**CB-Efficiency.** We assess task performance, robustness, and interaction cost. Specifically, CB-Efficiency reports task completion steps or final game scores, together with their standard deviations across diverse player profiles, as well as the average number of generated tokens per step as a proxy for communication and computational cost.

**CB-Affective.** We evaluate the affective quality of collaboration under CollabBench using three sub-dimensions widely adopted in psychological studies: **Helpfulness** evaluates whether $P_{target}$ provides relevant, actionable, and clear outputs that accurately capture the players intentions to support task completion. **Trustfulness** evaluates whether $P_{target}$ can reliably interpret and execute the players instructions, and respond in a stable and timely manner during interaction. **Empathy** evaluates whether $P_{target}$ can perceive the simulated player $P_{sim}$'s profile and emotional state, and adapt its behavior to offer supportive and encouraging responses under confusion or failure. All scores are computed using a penalty-based LLM judge (Wang et al., 2025a). Following Section 4.1.2, segment-level evaluations are aggregated into trajectory-level, dimension-wise scores to reduce fine-grained subjective bias (Appendix C.3).

**Environments.** CollabBench is built by extending two multi-agent games, Communicative Watch-And-Help (CWAH) (Zhang et al., 2024b) and Overcooked-AI (Cook) (Carroll et al., 2019), into CWAH-MultiPlayer and Cook-MultiPlayer for multi-turn collaboration with diverse player profiles. CWAH-MultiPlayer includes five tasks with two layouts and fifteen profiles each (150 trajectories), while Cook-MultiPlayer contains five layouts with a "send message" action and fifteen profiles per layout (75 trajectories).

## 5. Experiments

This section primarily presents the performance of mainstream LLMs on CollabBench, evaluated within prior SOTA agent frameworks. It also provides an analysis of CollabBench itself and compares LLM performance before and after our collaborative agentic training for future training insights. We will answer the following four key research questions:

• **RQ1**: How do mainstream foundation models perform alongside our trained models on CollabBench?

• **RQ2**: How does CollabBench capture the diversity, robustness, and realism of anthropomorphic collaboration behaviors?

• **RQ3**: What insights do persona diversity, efficiency and affective rewards provide for collaborative agentic training?

• **RQ4**: How does our trained agent perform compared with the base model in real human-AI collaboration?

The codes are available at `https://github.com/BW297/CollabBench`.

### 5.1. Experimental Setup

**Players Simulator $P_{sim}$ Settings.** To build a two agent cooperative game environment, we employ DeepSeek-V3.1 as teammate simulator LLM to role-play diverse player behaviors, given observation $s_i$ of each turn and different player profiles provided in Section 4.1.

**Collaborative Agent $P_{target}$ Baselines.** For CWAH-MultiPlayer, We adopt CoELA (Zhang et al., 2024b) as the agent framework and instantiate it with GPT-5.2, DeepSeek-V3.1, and Qwen2.5-72B-Instruct as backbone LLMs. For Cook-MultiPlayer: We use ProAgent (Zhang et al., 2024a) as the underlying framework, instantiated with the same set of backbone LLMs. For our collaborative agentic training, we compare Qwen2.5-7B-Instruct before and after training to evaluate the effectiveness of our method.

**Implementation Details.** We apply full-parameter reinforcement learning without supervised fine-tuning. Agent 1 serves as $P_{target}$ and Agent 2 as $P_{sim}$, with all trainable parameters inherited from $P_{target}$. We employ DeepSeek-V3.1 for affective evaluation. For Cook-MultiPlayer, all five scenarios are used for training, and final performance is measured by game scores. For CWAH-MultiPlayer, 40 scenarios are used for training and the remaining 10 for testing, with final performance measured by task completion steps. Notably, in CWAH-MultiPlayer, the player profiles in the 40 training scenarios are generated by slightly perturbing the behavior patterns of the 15 test profiles excluded from evaluation, ensuring no overlap with test-time player instances. Additional details are provided in Appendix D.1.

### 5.2. Experimental Results

5.2.1. MAIN RESULTS ON COLLABBENCH (TO RQ1)

We present the results in Table 1 and key findings are:

**Limitations in Balancing Affective and Efficient Collaboration.** Current LLMs struggle to balance efficiency and affective capabilities, with notably weak performance in

*Table 1.* Evaluation Results on CollabBench. Blue Zone : Performance of the base model and our trained model. Red Zone : Relative Improvements indicates the relative improvements of our trained model over the base model. Oracle: Affective performance upper bound with ground-truth opposite player profile and scoring principle. Agent 1 or 2 indicates the role assumed by the collaborative agent $P_{target}$ in the game. All results are reported as averages over all evaluation trajectories. Detailed results are provided in Appendix D.2.

| | | CWAH-MultiPlayer | | | | | | | | | | |
|---|---|---|---|---|---|---|---|---|---|---|---|---|
| | | CB-Efficiency | | | | | | CB-Affective | | | | |
| Metric | | Step ↓ | | Std. ↓ | | #Tokens(k) ↓ | | Helpfulness ↑ | | Trustfulness ↑ | | Empathy ↑ | |
| Method | LLMs | Agent 1 | Agent 2 | Agent 1 | Agent 2 | Agent 1 | Agent 2 | Agent 1 | Agent 2 | Agent 1 | Agent 2 | Agent 1 | Agent 2 |
| Oracle | GPT-5.2 | 60.91 | 60.29 | 22.26 | 25.01 | 0.16 | 0.16 | 2.81 | 2.96 | 3.77 | 4.01 | 3.69 | 3.53 |
| Base | GPT-5.2 | 67.49 | 62.98 | 25.84 | 20.70 | 0.21 | 0.21 | 2.66 | 2.72 | 3.74 | 3.66 | 3.07 | 3.32 |
| | DeepSeek-V3.1 | 69.26 | 65.75 | 25.17 | 26.28 | 0.41 | 0.43 | 2.32 | 2.45 | 3.35 | 3.50 | 2.87 | 3.30 |
| | Qwen2.5-72B-Instruct | 68.68 | 66.54 | 24.36 | 23.65 | 0.29 | 0.29 | 2.41 | 2.51 | 3.61 | 3.71 | 3.23 | 3.39 |
| | Qwen2.5-7B-Instruct | 84.51 | 90.03 | 33.23 | 31.62 | 0.24 | 0.24 | 1.22 | 1.04 | 2.58 | 2.19 | 2.50 | 2.30 |
| Trained | Qwen2.5-7B-Instruct | 71.64 | 63.65 | 25.16 | 22.80 | 0.23 | 0.23 | 1.43 | 1.45 | 3.03 | 3.02 | 3.33 | 3.02 |
| Relative Improvements | | 15.2% | 29.3% | 24.3% | 27.9% | 4.2% | 4.2% | 17.2% | 39.4% | 17.4% | 37.6% | 33.5% | 31.5% |

| | | Cook-MultiPlayer | | | | | | | | | | |
|---|---|---|---|---|---|---|---|---|---|---|---|---|
| | | CB-Efficiency | | | | | | CB-Affective | | | | |
| Metric | | Score ↑ | | Std. ↓ | | #Tokens(k) ↓ | | Helpfulness ↑ | | Trustfulness ↑ | | Empathy ↑ | |
| Oracle | GPT-5.2 | 143.47 | 134.90 | 53.02 | 33.56 | 0.30 | 0.20 | 2.68 | 2.34 | 3.84 | 3.45 | 3.47 | 3.46 |
| Base | GPT-5.2 | 135.20 | 137.12 | 41.96 | 42.40 | 0.20 | 0.20 | 1.63 | 1.88 | 2.89 | 3.10 | 2.27 | 2.52 |
| | DeepSeek-V3.1 | 136.53 | 136.80 | 40.30 | 46.24 | 0.31 | 0.31 | 1.79 | 2.07 | 2.97 | 3.04 | 2.67 | 2.77 |
| | Qwen2.5-72B-Instruct | 135.47 | 114.13 | 41.99 | 39.45 | 0.27 | 0.26 | 1.37 | 1.26 | 2.77 | 2.65 | 2.45 | 2.48 |
| | Qwen2.5-7B-Instruct | 86.93 | 85.87 | 33.30 | 33.73 | 0.23 | 0.23 | 0.45 | 0.53 | 1.92 | 1.87 | 1.86 | 1.88 |
| Trained | Qwen2.5-7B-Instruct | 99.20 | 102.40 | 34.03 | 34.76 | 0.23 | 0.22 | 0.74 | 0.55 | 2.26 | 1.99 | 2.12 | 2.08 |
| Relative Improvements | | 14.1% | 19.3% | -2.2% | -3.1% | 0.0% | 4.3% | 62.7% | 4.6% | 17.7% | 6.5% | 14.0% | 10.4% |

helpfulness and empathy. Models often rely on superficial collaboration strategies and show limited sensitivity to communication timing and human partners intentions (Detailed case study in Appendix D.3.5). Trustfulness scores are relatively higher, likely due to instruction-following and alignment training. While game scores show no significant differences across models, proprietary models (e.g., GPT-5.2) outperform open-source ones on most affective dimensions. Increased token generation by open-source models does not translate into improved affective performance.

**Affective Sensitivity to Interaction Dynamics.** Game genre and interaction dynamics strongly influence affective performance. Models exhibit better affective behaviors in the slower-paced CWAH-MultiPlayer than in the time-critical Cook-MultiPlayer. In high-frequency interactions, ill-timed actions and disruptive error correction degrade gameplay flow and player experience.

**Training for Joint Efficiency and Affective.** After affect-aware collaborative training, Qwen2.5-7B-Instruct consistently improves across all evaluated dimensions, achieving a balanced gain in both efficiency and affective capabilities. Training dynamics and insights are analyzed in Section 5.2.3 and Appendix D.4.

5.2.2. ANTHROPOMORPHIC ANALYSIS ON
         COLLABBENCH (TO RQ2)

To evaluate the anthropomorphic behavior of CollabBench in diversity, robustness, and realism, we take CWAH-

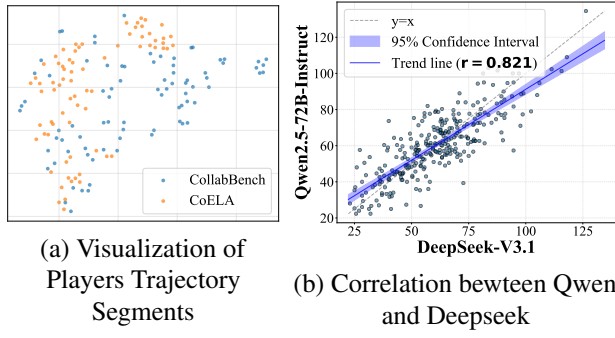

(a) Visualization of Players Trajectory Segments

(b) Correlation between Qwen and Deepseek

*Figure 3.* Diversity and robustness analysis on CWAH-MultiPlayer.

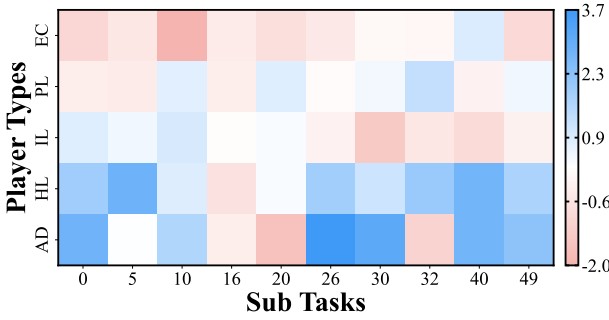

*Figure 4.* Heatmap of score distributions on CWAH-MultiPlayer.

MultiPlayer as an example for analysis. By default, $P_{sim}$ and $P_{target}$ use DeepSeek-V3.1, with Agent 1 as $P_{sim}$ and Agent 2 as $P_{target}$. Results for Cook-MultiPlayer are provided in the Appendix D.3.1.

**Diversity Analysis**. CollabBench leads to greater variabil-

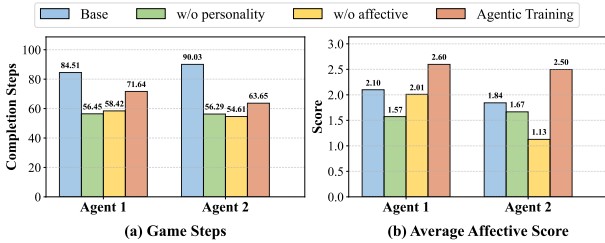

*Figure 5.* Ablation study of collaborative agentic training on CWAH-MultiPlayer.

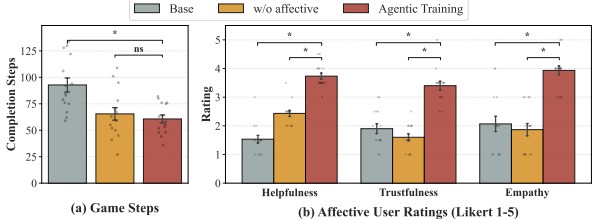

*Figure 6.* Human evaluation results of CWAH in the user study. Error bars denote standard error. Statistical significance is determined by $t$-tests: "*" means statistically significant ($p < 0.01$) and "$ns$" means non-significant.

ity in performance than the baseline methods, evaluated using standard deviation. Due to space limitations, detailed results are provided in Appendix D.3.2. In addition, we embed and visualize trajectory segments of players using t-SNE (Van der Maaten & Hinton, 2008) in Figure 3 (a), showing that CollabBench covers a broader range of interaction patterns. The corresponding quantitative metrics and comparison results are provided in Appendix D.3.3. These results demonstrate that CollabBench captures high diversity in anthropomorphic collaborative behaviors.

**Robustness Analysis**. To evaluate the robustness, we use Qwen2.5-72B-Instruct and DeepSeek-V3.1 to drive the players, and compute the pearson correlation coefficients between their game scores. The results in Figure 3 (b) indicate that CollabBench maintains a consistent relative performance across LLMs, with 0.821 on Cwah-MultiPlayer.

**Realism Analysis**. To validate realism, we embed trajectory segments from different player profiles using Qwen3-Embedding-4B and employ K-means clustering to group them into five player types. For each cluster, we select representative profiles and visualize their game score distributions, where the player types are labeled by double uppercase letters. As shown in Figure 4, different player profiles lead to markedly distinct and reasonable performance. The results indicate that proactive communication and structured task allocation lead to superior performance. However, hesitant execution and communication-avoidant behaviors lead to degraded performance. In less collaboration-dependent settings, independent action preference can still perform well. These observations align well with intuitive collaborative dynamics. The corresponding player type descriptions and example profiles are provided in Appendix D.3.4.

Furthermore, we analyze the interaction records in lowest-performing scenarios to uncover causes of performance degradation and offer insights for improving LLM collaboration. The detailed analyses are provided in Appendix D.3.5.

### 5.2.3. ABLATION STUDY ON COLLABORATIVE AGENTIC TRAINING (TO RQ3)

In Figure 5, we present ablation studies of collaborative agentic training on Qwen-2.5-7B-Instruct in CWAH-

MultiPlayer. Here, *w/o personality* indicates that during training of $P_{target}$, $P_{sim}$ is not assigned any profile and was trained using standard self-play setting; *w/o affective* indicates that only the efficiency reward was used, omitting the affective reward. The average affective score is defined as the average over three affective dimensions. We find that both ablations exhibit a clear limitation: while efficiency improves substantially, affective performance drops sharply. In particular, models gradually lose their communication capability, treating the game as effectively single-player, which severely degrades the player experience. Further details on model types, model sizes, training behaviors and case study are provided in the Appendix D.4.

### 5.2.4. USER STUDY (TO RQ4)

To evaluate the practical effectiveness and user experience of collaborative agentic training in real-world human-AI collaboration, we conducted a user study with 15 participants (12 male, 3 female; mean age 21.8). Each participant was assigned to one of three tasks (Task 0, 20, 40) in CWAH, with assignments evenly balanced to control task difficulty. Participants interacted with three variants of Qwen2.5-7B-Instruct: the base model, a model without affective reward, and our fully trained model. Efficiency was measured by completion steps. After interacting with each model, participants completed a 5-point Likert-scale questionnaire assessing three affective dimensions: Helpfulness (2 items), Trustfulness (2 items), and Empathy (1 item). Detailed settings are provided in Appendix D.5.

As shown in Figure 6, both the ablation and fully trained models significantly reduced completion steps compared to the base model, indicating that affective rewards do not influence efficiency seriously. In contrast, subjective evaluations revealed that the fully trained model achieved substantially higher scores across all affective dimensions. Removing affective rewards led to a marked decline in user satisfaction, with participants describing the ablation model as efficient but cold, particularly in terms of trustfulness and empathy. These results demonstrate that collaborative agentic training is crucial for producing agents that are both efficient and affectively aligned.

## 6. Limitation and Discussion

**Immersive Player Simulation and Behavioral Alignment.** While our work establishes a foundational and scalable paradigm for utilizing LLMs to generate high-quality, diverse player simulations, significant potential remains for further research in anthropomorphic modeling. Unlike standard dialogue scenarios, cooperative game environments require agents to align their self-cognition and persona not only within linguistic exchanges but also through embodied in-game actions. This "embodied" context, where communication is intertwined with execution, brings the simulation closer to real-world complexity. Future research could deepen this alignment by incorporating Reinforcement Learning or established other personality frameworks (e.g., MBTI) to create more consistent and nuanced behavioral patterns. We hope our findings provide valuable insights for subsequent work aiming to bridge the gap between conversational personas and action-oriented behavior in complex environments.

**Challenges in Collaborative Agentic Training.** There are also substantial opportunities for further exploration within collaborative agentic training. We observe a persistent gap in existing LLMs regarding the balance between maximizing task efficiency and maintaining a positive player experience. The complexity of game interactions often hinders the models ability to master communication timing and decipher the deeper intent behind user behaviors; consequently, LLMs can exhibit template-like communication with low information density. In an era dominated by task-efficiency-driven reinforcement learning, fostering such affective capabilities is inherently difficult.

A closely related challenge lies in mitigating reward hacking in affective optimization. Although our training objective jointly considers both efficiency and affective dimensions, and further introduces complementary mechanisms within the affective reward, such as combining communication rewards with subjective affective judgments to cross-validate communication frequency and quality, these designs primarily aim to reduce degenerate behaviors that optimize a single affective signal (e.g., excessive but uninformative communication). Nevertheless, our human evaluation indicates that, even after extensive affective training, the agents communication skills and empathetic understanding remain imperfect. This suggests that more robust and principled affective reward designs, capable of better resisting reward hacking while preserving genuine interaction quality, remain an open and important direction for future work.

Additionally, we note that in scenarios with high-frequency interactions, such as Cook-MultiPlayer, the training time overhead becomes substantial, highlighting the need for more computationally efficient optimization algorithms in future research.

**Refinement of Evaluation Protocols.** Regarding our evaluation protocol, while our affective metrics are grounded in psychological theory, they inevitably retain a degree of subjectivity that warrants further refinement. An important related aspect is the consistency between human judgments and LLM-based evaluations. In our setting, collaborative game trajectories are often long and complex, making exhaustive human annotation prohibitively time-consuming and costly. To alleviate this issue, we adopt an improved LLM-based evaluation strategy that segments long interaction trajectories into shorter, semantically coherent clips, which helps reduce fine-grained scoring bias in subjective assessment. Further discussions of the LLM judge for affective evaluation are provided in Appendix D.6.

Nevertheless, we believe that human calibration remains an essential component for validating affective and interaction-oriented metrics, particularly for nuanced qualities such as empathy and trustfulness. Addressing the high human labor cost associated with such calibration, and designing more efficient human-in-the-loop evaluation protocols, constitutes an important direction for our future work. For instance, this may be achieved by designing targeted interaction scenarios or specific end-game puzzles, where affective and collaborative outcomes are more clearly defined, thereby substantially mitigating the potential bias inherent in subjective LLM-based assessments.

Furthermore, regarding the breadth of our evaluation, constraints on time and resources limited our scope to a selection of representative models. Given that collaborative capability is a vital criterion for the evolution of LLMs, a more comprehensive assessment covering a wider spectrum of models remains an essential direction for future benchmarking efforts.

## 7. Conclusion

Genuine human-AI collaboration requires agents to move beyond solitary task execution and proactively adapt to the behavioral and affective needs of human partners. To overcome the limited grounding and interactivity of prior conversation-level studies, this paper proposes cooperative game environments as a principled testbed for immersive collaboration. We introduce CollabBench, a benchmark that enables collaborative agentic training through diverse player simulation and supports multi-dimensional evaluation of both efficiency and affective alignment. Extensive experiments show that agents can learn to balance task performance with empathy, offering actionable insights for future training strategies and establishing a benchmark for socially aware, proactive collaborators.

## Acknowledgments

We thank the anonymous reviewers for their constructive comments. This work is supported by the National Key Research and Development Program of China under Grant 2024YFC3308503, Tencent, and the Shanghai Municipal Special Program for Basic Research on General AI Foundation Models under Grant No. 2025SHZDZX026D08.

## Impact Statement

This paper aims to advance **socially aware** collaborative intelligent agents. In our user study, we collected participants' behavioral data and feedback to assess the affective alignment of collaborative agents. During data collection, we strictly protected participants' privacy by excluding any personally identifiable information (PII), ensuring that participation was entirely voluntary, and following principles of fair compensation.

We believe this work has a positive societal impact. First, by improving understanding and adaptability of LLMs in multi-agent collaborative tasks, it can facilitate more efficient and empathetic interactions in applications such as educational platforms and virtual assistants. More specifically, intelligent agents with social awareness and collaborative capabilities can support more affective communication, improving user engagement and learning experiences in educational settings (e.g., collaborative learning and intelligent tutoring). They may also help alleviate users psychological burden in emotionally sensitive interactions (e.g., complex decision-making and high-cognitive-load tasks). Second, we emphasize maintaining fairness and transparency throughout agent training and deployment, while mitigating potential biases and risks of emotional misalignment. These considerations are particularly important for socially interactive systems, as inappropriate affective behaviors or biased responses may influence users trust, motivation, and emotional experience.

By systematically including these considerations, our work offers a responsible reference for deploying collaborative intelligent agents in real-world applications.

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

# A. Details of Motivation Study

In this section, we provide the experimental setup and results for Efficiency Challenge in Figure 1 in Section 1. Both $P_{sim}$ and $P_{target}$ are powered by DeepSeek-V3.1. Under the default setting, we adopt CoELA as the game environment for CWAH-MultiPlayer and ProAgent for Cook-MultiPlayer. Here, we consistently adopt Agent 1 as $P_{sim}$ and Agent 2 as $P_{target}$, and conduct experiments across all 30 filtered player profiles. The complete results are reported in Figure 7 and Figure 8.

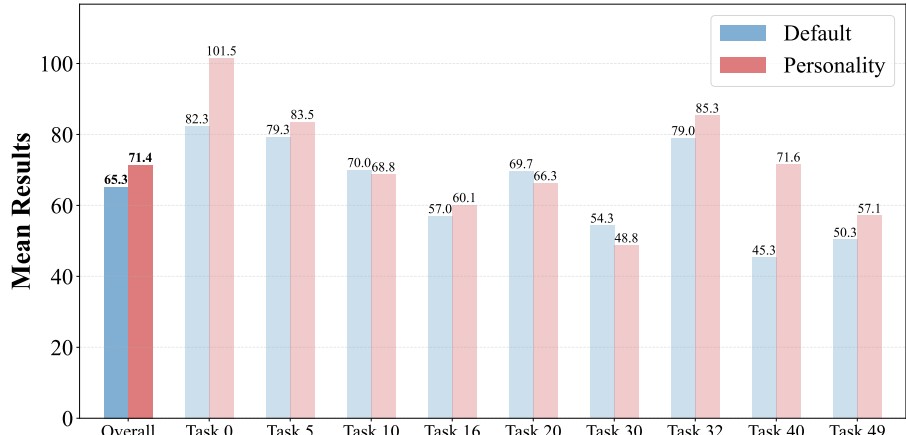

*Figure 7.* The complete comparison results for Efficiency Challenge on CWAH-MultiPlayer.

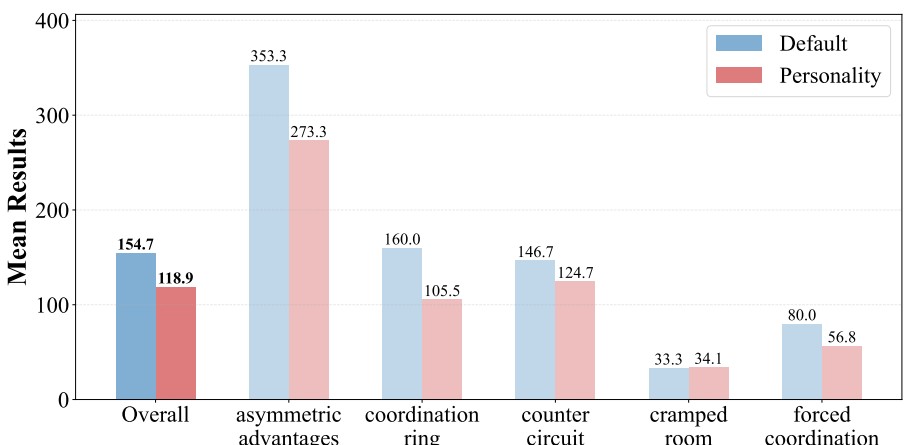

*Figure 8.* The complete comparison results for Efficiency Challenge on Cook-MultiPlayer.

# B. Related Work

### B.1. Personality-Aware Human Simulation.

Realistic human simulation is essential across a wide range of interactive settings. Traditionally, human simulators are learned by fitting models to collected human behavior data (Carroll et al., 2019). With the emergence of LLMs, it has been shown that they can generate diverse and authentic personas (Kosinski, 2024; Salminen et al., 2024; Shin et al., 2024), enabling more flexible and scalable human simulation. This capability has been widely adopted in domains such as role-playing (Wang et al., 2024c; Tu et al., 2024; Wang et al., 2025a), where LLMs simulate characters from fictional narratives and films, and personalized services (Salemi et al., 2024; Jia et al., 2026; Huang et al., 2025), where agents are conditioned on user profiles including demographics, personality traits, interests and careers. However, these approaches typically rely on predefined character descriptions or user data. In simulated embodied cooperative game environments, data for players with diverse personalities and behavioral styles remains scarce. Existing collaboration systems between

human and LLM-based agents, such as CollabLLM (Wu et al., 2025), employ user simulators for tasks like document editing, mathematics and coding, but these simulators are largely task-driven, lack personality diversity, and operate at the conversation level, limiting their ability to capture realistic human behaviors and environmental dynamics. Researches in social science have explored human simulation through LLMs with diverse personalization and dynamic actions in specific domains such as daily dialogues (Wang et al., 2024d; Zhang et al., 2025a), social activity modeling (Park et al., 2023) and education (Liu et al., 2024). Despite their success, these approaches are often domain-specific and cannot be directly applied to embodied collaborative tasks between human and LLM-based agents.

### B.2. Agentic Training for LLMs in Games.

Recent studies have demonstrated the impressive performance of LLM-based agentic training in fields like deep research (Jin et al., 2025), mathematical reasoning (Guo et al., 2025), and coding (Jiang et al., 2026). Game environments (Chevalier-Boisvert et al., 2019; Carroll et al., 2019; Shridhar et al., 2021; Puig et al., 2021; Wang et al., 2024a) offer significant practical value for applying this technology, providing a realistic platform to enhance agentic training and simulate complex real-world scenarios.

• **Single Player Training:** Early research on single agentic training primarily focused on the design of agent architectures, such as VOYAGER (Wang et al., 2024a), which introduces a novel agent architecture for embodied lifelong learning in Minecraft, combining an automatic curriculum and an evolving skill library. As agentic reinforcement learning has been validated in an increasing number of applications, recent work has also begun to apply agentic RL to train LLMs in simulated embodied game environments, such as RAGEN (Wang et al., 2025b) and AgentGym-RL (Xi et al., 2026). While these studies are crucial for enhancing the capabilities of LLMs themselves, the potential of using Agentic RL to improve the collaboration awareness remains underexplored.

• **Collaborative Teammate Training:** Most existing research on collaborative teammate agents still primarily focuses on agent architecture design (Zhang et al., 2024b;a; Seo et al., 2025). For example, in the Overcooked environment, ProAgent (Zhang et al., 2024a) employs a belief revision mechanism to create proactive agents that dynamically adapt their behaviors to enhance cooperation with teammates. In the C-WAH environment, CoELA (Zhang et al., 2024b) integrates perception, memory, execution, planning, and communication modules to cooperate with others in accomplishing long-horizon tasks. However, architecture-centric designs often lack explicit mechanisms for eliciting and training collaboration awareness within LLMs themselves, and they generally overlook the diversity of teammate personalities that commonly arises in real-world collaboration settings.

### B.3. Evaluation of LLM-based Agent and Human Collaboration.

As the capabilities of LLMs continue to grow, the collaboration ability of LLM-based agents has become increasingly important. However, when it comes to collaboration, efficiency is often not the sole metric of evaluation (George, 1990; McAllister, 1995; Salas et al., 2005). Recently, an increasing number of studies have shifted the focus from traditional efficiency-based metrics to more multidimensional evaluation frameworks. For example, the LLM-as-a-judge approach has been adopted to evaluates interactivity (Wu et al., 2025) by assessing user engagement levels or empathy (Zhang et al., 2025a) to determine whether the model can provide timely emotional support to users. However, in current gamified evaluations (Costarelli et al., 2024; Xi et al., 2025b; Hu et al., 2026; Sun et al., 2025; Zhang et al., 2025b), most assessments still rely solely on efficiency metrics such as game scores or success rates to measure the capabilities of LLMs. While some research in cooperative games has expanded beyond efficiency metrics, it typically relies on subjective user feedback through surveys (Siu et al., 2021; Zhang et al., 2024b), lacking systematic evaluation criteria and methodologies. This limitation hampers large-scale evaluations and complicates reward feedback signals in current Agentic RL training.

## C. Technical Details

### C.1. Detailed process of clustering

Here, we provide the details of clustering process. We segment LLM-based high-level trajectories into units of three time windows to maintain reasoning coherence while controlling contextual overhead. Each segment is then embedded using Qwen3-Embedding-4B and grouped using K-means clustering (Hartigan & Wong, 1979), with each cluster of segments used to construct the corresponding candidate player profiles.

### C.2. Quantitative Validation for Profile Filtering

Here, we compare the quantitative results before and after filtering to evaluate the fidelity of player profiles. The detailed metrics, implementation details, and the results analysis are as follows.

**Quantitative Metrics**.

• **Intra-Class Distance** measures the degree of clustering of trajectory segments generated by the same player profile. Assume there are $Q$ embed trajectory segments $\mathbf{x}_i \in \mathbb{R}^d$, associated with the corresponding player profiles $\eta_i$. For profile $\eta$, let $C_\eta = \{\mathbf{x}_i \mid \eta_i = \eta\}$ denote the set of their trajectory segments. The pairwise cosine distance between segments is defined as:

$$d_{ij} = 1 - \frac{\mathbf{x}_i \cdot \mathbf{x}_j}{\|\mathbf{x}_i\|\|\mathbf{x}_j\|}, \quad \text{where } i, j \in C_\eta, i \neq j. \tag{7}$$

The average intra-class distance for profile $\eta$ is:

$$D_{\text{Intra}_\eta} = \frac{2}{|C_\eta|(|C_\eta| - 1)} \sum_{i < j} d_{ij}, \quad \text{where } i, j \in C_\eta. \tag{8}$$

The global $D_{\text{Intra}}$ in profile set $\mathcal{E}$ can be defined as $D_{\text{Intra}} = \frac{1}{|\mathcal{E}|} \sum_{\eta \in \mathcal{E}} D_{\text{Intra}_\eta}$. Smaller values indicate tighter clustering of trajectory segments from the same profiles, reflecting higher intra-class consistency.

• $k$-**Nearest Neighbor Label Consistency** ($k$-**NN Label Consistency**) measures the degree to which neighboring trajectory segments share the same profile. For trajectory segment $\mathbf{x}_i$, let $N_i$ denote its $k$ nearest neighbors (excluding itself). Its local consistency is defined as:

$$\text{LC}_{\mathbf{x}_i}^k = \frac{1}{k} \sum_{j \in N_i} \mathbb{I}(\eta_j = \eta_i), \tag{9}$$

where $\mathbb{I}(\cdot)$ is the indicator function. The global $\text{LC}^k$ can be defined as $\text{LC}^k = \frac{1}{Q} \sum_{i=1}^Q \text{LC}_{\mathbf{x}_i}^k$. Larger values indicate high local consistency within the same profile.

• **Consistency Score**. Based on the two metrics above, we define Consistency Score as:

$$\text{Consistency Score} = \begin{cases} \gamma \cdot \text{LC}^k + (1 - \gamma) \cdot \frac{1}{1 + D_{\text{Intra}}}, & \text{if } D_{\text{Intra}} > 0 \\ 0, & \text{otherwise} \end{cases}, \tag{10}$$

where $\gamma$ is the weighting coefficient. The score captures both intra-class and local consistency of the trajectory segments, with larger values indicating better fidelity, providing a quantitative basis for validating filtered player profiles. Notably, profile fidelity focuses on the consistency of behavior within the same profile, rather than the distinguishability between different profiles. Therefore, inter-class measures are not incorporated in Consistency Score.

**Implementation Details**. The weighting coefficient $\gamma$ is set to 0.5. We segment LLM-based high-level trajectories into units of three time windows and embed the segments using Qwen3-Embedding-4B.

*Table 2.* The comparison results before and after player profile filtering.

| Metrics | CWAH-MultiPlayer | | Cook-MultiPlayer | |
|---|---|---|---|---|
| | Before | After | Before | After |
| Intra-class Distance $D_{\text{Intra}} \downarrow$ | 0.1237 | **0.1236** | **0.1034** | 0.1040 |
| $k$-NN Label Consistency $\text{LC}^k \uparrow$ | 0.4511 | **0.5470** | 0.5729 | **0.6760** |
| Consistency Score $\uparrow$ | 0.1006 | **0.1220** | 0.1298 | **0.1531** |

**Results Analysis**. As shown in Table 2, the results confirm that filtering enhances the fidelity of profiles compared to before. This further underscores the necessity and effectiveness of the secondary filtering step.

## C.3. Affective Metrics Scoring

Here, we provide the detailed evaluation criteria for affective metrics and description of the scoring computation. The corresponding prompt is shown in Appendix E.4.

### C.3.1. EVALUATION CRITERIA

We segment the interaction trajectories of $P_{target}$ into three time windows as the minimal evaluation unit to evaluate the affective quality of reasoning, communication, and action in the single-pass output, marked by `<think>`, `<message>`, `<action>`, respectively. The focus of each specific criterion is indicated in [·].

- **Helpfulness.** The specific dimensions are as follows:

  ○ **Task Focus.** [`<think>`] Whether $P_{target}$ allocates attention and time primarily to task progression and key steps, rather than engaging in irrelevant interactions.

  ○ **Adaptation.** [`<think>`, `<action>`] Whether $P_{target}$ proactively identifies missing information, proposes executable plans, synchronizes key progress, and dynamically adjusts its behavior according to the state and task progress of $P_{sim}$ to fill gaps while avoiding redundant effort.

  ○ **Communication Clarity.** [`<message>`] Whether the message is structured, comprehensible, and executable by $P_{target}$.

  ○ **Stuck or Negative Behaviors.** [`<action>`] Including loops, indecision, or prolonged lack of progress.

  ○ **Intent Alignment.** [`<think>`, `<action>`] Whether $P_{target}$ demonstrates understanding of the latent goals of $P_{sim}$, rather than merely executing surface-level instructions.

- **Trustfulness.** The specific dimensions are as follows:

  ○ **Interaction Responsiveness.** [`<message>`, `<action>`] Whether $P_{target}$ accurately responds to requests, promptly addresses key issues and updates, and avoids deviating from collaboration goals or forcing its own plan in the absence of consensus.

  ○ **Synchronization Latency.** [`<message>`, `<action>`] Whether key discoveries or sub goal completions are shared promptly, preventing delayed information from causing redundant effort or collaboration inefficiency.

- **Empathy.** The specific dimensions are as follows:

  ○ **Personality Inference.** [`<think>`, `<message>`] Whether $P_{target}$ adjusts its tone, expressions, and interaction strategies based on the personality traits or preferences of $P_{sim}$, avoiding clear mismatches in communication style.

  ○ **Emotional Support.** [`<message>`, `<action>`] Whether $P_{target}$ maintains politeness, encouragement, and emotional acceptance, and provides timely reassurance and constructive help when $P_{sim}$ shows frustration or uncertainty; cold, harsh, or dismissive behavior is considered negative.

It is worth noting that not every situation within a given time window fully reflects all of the aforementioned dimensions. This is precisely why we employ a penalty-based LLM judge: at each time window, we assess whether the sequence of outputs of the agent violates any of these dimensions and assign penalties accordingly.

### C.3.2. SCORING COMPUTATION

Inspired by (Wang et al., 2025a), we adopt a penalty-based LLM judge to evaluate the quality of affective interaction along three dimensions: *Helpfulness*, *Trustfulness*, and *Empathy*. Each interaction trajectory is segmented into fixed-size windows of $K{=}3$ consecutive high-level actions (LLM calls). Incomplete tail windows are discarded. Each window is independently scored for the three dimensions.

• **Window-level scoring.** For a window $w$ and dimension $d \in \{Helpfullness, Trustfulness, Empathy\}$, the judge outputs a list of violations $\mathcal{V}_{w,d}$ with deduction severities $\delta_{w,d,j} \in \{1, 2, 3\}$. The window score is computed by deducting from 5:

$$s_{w,d} = \max\Big(0,\ 5 - \sum_{j=1}^{|\mathcal{V}_{w,d}|} \delta_{w,d,j}\Big). \tag{11}$$

The judge is instructed to start from 5, deduct for each violation, and return strict JSON with evidence snippets.

• **Trajectory-level aggregation.** Let $\mathcal{W}_\tau$ be the set of windows for trajectory $\tau$, with $n = |\mathcal{W}_\tau|$. Define the total violation count $V_{\tau,d} = \sum_{w \in \mathcal{W}_\tau} |\mathcal{V}_{w,d}|$, and the worst window score gap $G_{\tau,d} = \max_{w \in \mathcal{W}_\tau}(5 - s_{w,d})$. The aggregated score for dimension $d$ is:

$$S_{\tau,d} = \text{clip}_{[0,5]}\Big(5 - a_d \frac{V_{\tau,d}}{n} - b_d\, G_{\tau,d} - P_{\text{msg}}(\rho_\tau)\Big). \tag{12}$$

Here $\rho_\tau$ is the agent send-message ratio at the trajectory level,

$$\rho_\tau = \frac{\#\, P_{target}\ \text{send\_message steps}}{\#\, P_{target}\ \text{llm\_call steps}},$$

and $P_{\text{msg}}$ is a trajectory-level penalty applied when messaging is too sparse.

$$P_{\text{msg}}(\rho_\tau) = \begin{cases} 0, & \rho_\tau \geq \varepsilon, \\ \lambda \cdot p_{\max}, & \rho_\tau < \varepsilon. \end{cases} \tag{13}$$

Here $\lambda = \dfrac{\exp(kx) - 1}{\exp(k) - 1}$ denotes the normalized exponential penalty factor. In this formulation, and $x = \dfrac{\varepsilon - \rho_\tau}{\varepsilon} \in (0, 1]$ measures the normalized shortfall of the observed message ratio relative to the minimum acceptable threshold , $\rho_\tau$ denotes the message sparsity ratio for trajectory (or window) $\tau$, and $\varepsilon$ is the minimum acceptable message ratio below which a penalty is applied. The scalar $p_{\max}$ specifies the maximum possible message-sparsity penalty, thereby capping the largest deduction that can be incurred due to insufficient communication. The parameter $k$ controls the curvature of the exponential scaling, determining how sharply the penalty factor $\lambda$ increases as $\rho_\tau$ falls further below $\varepsilon$.

This design prevents the message-frequency regularizer from dominating the affective score and ensures that $P_{\text{msg}}$ acts as a bounded soft constraint rather than a hard failure signal.

• **Parameter choices.** We follow the default evaluation settings in the provided pipeline: $K$=3 (window size), $a_d$=$b_d$=0.8 for all $d \in \{Helpfullness, Trustfulness, Empathy\}$, and message-penalty parameters $\varepsilon$=0.15, $k$=4.0, $p_{\max}$=2.0.

## D. Experiments

### D.1. Implementation Details

#### D.1.1. DETAILS ABOUT THE DATASETS

Here, we provide a detailed description of the synthesized dataset in Table 3. Notably, in CWAH-MultiPlayer, the player profiles for the 40 training scenarios are generated by slightly perturbing the behavior patterns of 15 player profiles drawn from the test scenarios and excluded from evaluation.

*Table 3.* Statistics of the synthesized dataset on CollabBench. "Trained" and "Untrained" refer to player profiles and sub tasks that are seen and unseen during agentic training, respectively.

| CWAH-MultiPlayer | | | Cook-MultiPlayer | | |
|---|---|---|---|---|---|
| **Dataset** | **#Subtasks** | **#Player Profiles** | **Dataset** | **#Subtasks** | **#Player Profiles** |
| **Train** | 40 | 94 | **Train** | 5 | 75 |
| **Test** | 10 | 150 | **Test** | 5 | 75 |
| **Total** | 50 | 244 | **Total** | 50 | 150 |

### D.1.2. DETAILS OF PLAYER SIMULATION

In **Game-Specific Behavioral Trajectory Generation** (Section 4.1.1), we employ multiple LLMs to simulate personality-driven player instances $P_{\text{sim}}$. The LLMs used include DeepSeek-V3, Qwen2.5-72B-Instruct, and Qwen3-235B-A22B.

In **Player Profile Filtering** (Section 4.1.2), we adopt Agent 1 as $P_{sim}$ and Agent 2 as $P_{target}$ for game interaction and the player trajectories are segmented into units of three time windows. The game set $\Omega_\eta$ under each player profile $\eta$, is set to 3. For consistency evaluation, we employ a penalty-based LLM judge built upon DeepSeek-V3.1. For CWAH-MultiPlayer, the penalty and reward coefficients $\alpha_p, \alpha_p^m, \alpha_r$ corresponding to Personality-Reasoning Consistency and Reasoning-Action Consistency are set to 0.5, 0.1, and 0.1, respectively. For Cook-MultiPlayer, the corresponding coefficients are set to 0.1, 0.1, and 0.01. The weighting coefficient $\beta$ for both consistency scores, $S_\eta^{\text{P-R}}$ and $S_\eta^{\text{R-A}}$, is set to 0.5. These parameter settings ensure that the scores $S_\eta^{\text{P-R}}$, $S_\eta^{\text{R-A}}$, and $S^{\text{ALL}}$ remain within the range of [0-5]. Finally, we retain the top 30 profiles with the highest $S^{\text{ALL}}$ scores for each sub task (if fewer than 30 profiles are available, all are retained).

The corresponding prompts for profile-driven simulated players in two game environments are provided in Appendix E.5.

### D.1.3. DETAILS OF TRAINING

All methods are configured with identical hyperparameters: the maximum prompt length is 4096 tokens, and the maximum response length is 4096 tokens. Each episode allows up to 150 environment steps. The learning rate is set to $1e^{-6}$. For efficiency reward, we adopt a rule-based reward for every trajectory. In CAWH-MultiPlayer, efficiency reward is $\exp\left(-0.025 \cdot \max(0, step - 40)\right)$, where $step$ is final steps to complete the goal. In Cook-MultiPlayer, efficiency reward is $score/100$, where $score$ denotes the cumulative sum of the final order scores, with each order valued at 20 points. For affective reward, to handle invalid actions generated by the agent, we first apply a reward penalty of $-0.1$. Second, we apply a reward of 0.01 to encourage the communication if agent choose to send a message in one step. Finally, we apply a comprehensive interactive reward for agent. This reward, assigned by an LLM judge, is quantized into 10 levels ranging from 0 to 1, and is further weighted by a factor of 0.25. These three rewards will be added together for every step as described in Section 4.2.2. We use a group size of 8 and sample 4 different groups per rollout, resulting in a total of $8 \times 4 = 32$ environments. The rollout temperature is set to 0.7. The mini-batch size is 32, and the KL-divergence loss coefficient is set to 0.01. As we use GIGPO for our optimization, we set a weighting coefficient $\omega$ fixed at 1, while the discount factor is set to 0, as we believe that applying a discount factor is not appropriate for our affective scenario. And our training framework is based on `verl-agent` (Feng et al., 2025) (`https://github.com/langfengQ/verl-agent`), which is built upon the veRL framework (Sheng et al., 2025) (`https://github.com/volcengine/verl`).

### D.1.4. DETAILS OF AGENT FRAMEWORK

Our agent framework is based on CoELA (Zhang et al., 2024b) (`https://github.com/UMass-Embodied-AGI/CoELA`) in CWAH-MultiPlayer and ProAgent (Zhang et al., 2024a)(`https://github.com/PKU-Alignment/ProAgent`) in Cook-MultiPlayer. The corresponding prompts for agentic teammates are provided in Appendix E.6.

### D.2. Detailed Main Results

### D.2.1. NON-LLM BASELINES RESULTS

It is worth noting that the non-LLM baselines lack reasoning and communication capabilities and therefore are not the primary focus of our study. Nevertheless, we still report their performance on efficiency-related metrics. Here, we provide the evaluation result of non-LLM baselines in Table 4 for CWAH-MultiPlayer and in Table 5 for Cook-MultiPlayer.

- **CWAH-MultiPlayer:** For the non-LLM baseline, we adopt MHP (Korf, 1987).

- **Cook-MultiPlayer:** For the non-LLM baseline, we adopt SP (Tesauro, 1994; Carroll et al., 2019), PBT (Jaderberg et al., 2017), FCP (Strouse et al., 2021), MEP (Strouse et al., 2021), COLE (Li et al., 2023; 2024).

### D.2.2. DETAILED MAIN RESULTS ACROSS SUB TASKS

Here, we provide the detailed evaluation result of LLM-based methods in Table 6 for CWAH-MultiPlayer and Table 7 for Cook-MultiPlayer.

*Table 4.* Evaluation Results of non-LLM baselines for CWAH-MultiPlayer.

| Metric | | Score | | Std. | |
|---|---|---|---|---|---|
| Method | Task | Agent 1 | Agent 2 | Agent 1 | Agent 2 |
| MHP | 0_read_book | 82.17 | 94.80 | 7.50 | 15.03 |
| | 5_read_book | 71.00 | 65.97 | 10.07 | 14.40 |
| | 10_put_dishwasher | 74.67 | 67.33 | 7.96 | 7.96 |
| | 16_put_dishwasher | 63.13 | 66.93 | 6.29 | 11.39 |
| | 20_prepare_food | 67.10 | 68.90 | 6.53 | 7.27 |
| | 26_prepare_food | 63.47 | 55.63 | 9.86 | 14.92 |
| | 30_put_fridge | 43.17 | 47.30 | 7.52 | 8.85 |
| | 32_put_fridge | 86.45 | 90.57 | 12.10 | 11.72 |
| | 40_setup_table | 60.70 | 71.30 | 8.02 | 10.24 |
| | 49_setup_table | 50.17 | 49.10 | 8.51 | 11.63 |
| | Total | 66.13 | 67.78 | 8.44 | 11.34 |

*Table 5.* Evaluation Results of non-LLM baselines for Cook-MultiPlayer.

| Method | | SP | | PBT | | FCP | | MEP | | COLE | |
|---|---|---|---|---|---|---|---|---|---|---|---|
| Task | Metric | Score | Std. | Score | Std. | Score | Std. | Score | Std. | Score | Std. |
| asymmetric_advantages | Agent 1 | 134.67 | 29.63 | 161.33 | 34.62 | 146.67 | 29.81 | 168.00 | 25.09 | 248.00 | 17.59 |
| | Agent 2 | 264.00 | 58.51 | 268.00 | 37.09 | 230.00 | 63.25 | 242.67 | 41.87 | 300.00 | 69.28 |
| forced_coordination | Agent 1 | 16.00 | 14.97 | 40.00 | 4.99 | 20.00 | 14.61 | 22.67 | 16.11 | 18.67 | 22.47 |
| | Agent 2 | 49.33 | 28.16 | 83.64 | 43.33 | 60.00 | 25.30 | 40.00 | 24.22 | 85.33 | 45.88 |
| coordination_ring | Agent 1 | 44.00 | 26.53 | 109.33 | 36.42 | 104.00 | 60.31 | 130.67 | 68.46 | 115.71 | 71.39 |
| | Agent 2 | 60.00 | 49.53 | 73.33 | 68.38 | 124.00 | 36.66 | 114.67 | 63.02 | 90.00 | 70.95 |
| cramped_room | Agent 1 | 40.00 | 11.47 | 20.00 | 20.48 | 92.00 | 41.83 | 78.67 | 37.57 | 89.33 | 44.94 |
| | Agent 2 | 101.33 | 36.12 | 53.33 | 33.00 | 135.56 | 41.93 | 126.67 | 59.18 | 148.00 | 32.50 |
| counter_circuit | Agent 1 | 76.00 | 16.65 | 108.00 | 17.59 | 70.67 | 19.14 | 86.67 | 11.93 | 137.33 | 22.94 |
| | Agent 2 | 42.67 | 44.94 | 117.33 | 17.69 | 86.00 | 23.75 | 105.33 | 30.52 | 101.54 | 51.12 |
| Total | Agent 1 | 55.20 | 19.85 | 77.87 | 22.82 | 86.67 | 33.14 | 97.33 | 31.83 | 121.89 | 35.86 |
| | Agent 2 | 103.47 | 43.45 | 99.31 | 39.90 | 130.81 | 38.18 | 125.87 | 43.76 | 148.57 | 53.94 |

**Observation Details**.

Here, we provide details of the partial observation $s_i$ of LLM-based methods at the $i$-th turn of interactions. The player simulator $P_{sim}$ follows the consistent observation setting.

• **CWAH-MultiPlayer**. The partial observation $s_i$ includes the overall goal objects, the task progress, game timestamp, dialogue history, action history, and the available action sets.

• **Cook-MultiPlayer**. The partial observation $s_i$ includes the current game timestamp, current kitchen scene, dialogue history, and the available action sets.

Based on the partial observation $s_i$ , the collaborative Agent $P_{target}$ adopts a single-pass rollout, generating internal reasoning trace $r_i$, natural language communication $c_i$, and the final executable action $a_i$ at each turn.

Furthermore, to assess the performance upper bound under ideal information conditions, we introduce an **Oracle** setting. In this setting, $P_{target}$ is driven by GPT-5.2. Compared to the above settings, the Oracle setting additionally incorporates the behavioral profile of $P_{sim}$ into the partial observation, as well as the previous turn's internal reasoning $r_{i-1}$, natural language communication $c_{i-1}$, and executed action $a_{i-1}$ of $P_{sim}$, along with collaboration guidance related to affective interactions.

### D.3. Anthropomorphic Results

#### D.3.1. COOK-MULTIPLAYER RESULTS

Here, we provide the anthropomorphic results on Cook-MultiPlayer in Figure 9: (a) compares the performance distributions between CollabBench and ProAgent. (b) shows the correlation between the performances of Qwen2.5-72B-Instruct and DeepSeek-V3.1, with a correlation coefficient of 0.753. and (c) shows the score distributions across five representative profiles. For detailed player type descriptions and the corresponding example profiles, please refer to Appendix D.3.4.

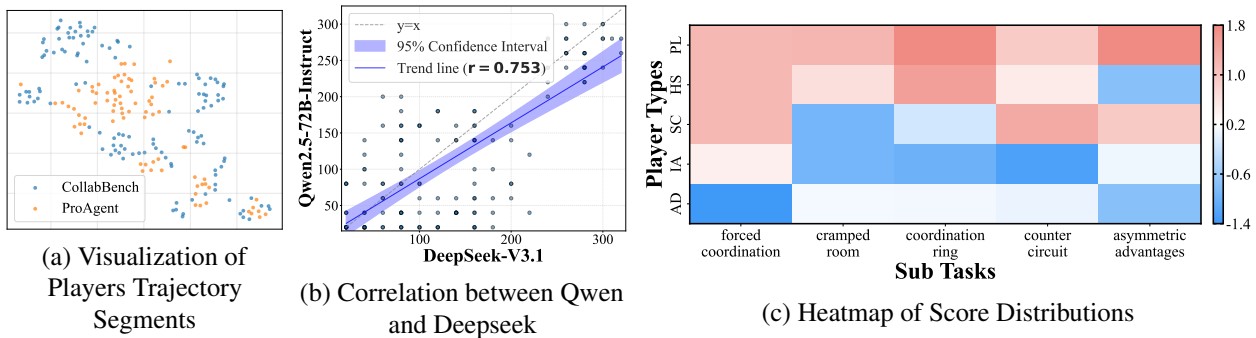

*Figure 9.* The anthropomorphic results on Cook-MultiPlayer.

#### D.3.2. DIVERSITY RESULTS BASED ON STANDARD DEVIATION

Here, we provide the comparison results between CollabBench and the baseline methods based on Standard Deviation across two game environments, using CoELA as the baseline method for CWAH-MultiPlayer and ProAgent for Cook-MultiPlayer as baselines. As shown in Table 8, CollabBench shows consistently greater performance variability.

#### D.3.3. QUANTITATIVE RESULTS OF INTERACTION DIVERSITY

We provide a quantitative comparison results to demonstrate the diversity of anthropomorphic collaborative behaviors on CollabBench. The detailed metrics, implementation details, and the results analysis are as follows.

**Quantitative Metrics.**

• **Spread** measures the overall dispersion of trajectory segments. Let embedded trajectory segments be $[\mathbf{x}_1, \mathbf{x}_2, \ldots, \mathbf{x}_Q] \in \mathbb{R}^d$, where $Q$ denotes the total number of trajectory segments. The centroid of the set of trajectory segments $\mathbf{c}$ is defined as $\mathbf{c} = \frac{1}{Q} \sum_{i=1}^{Q} \mathbf{x}_i$. The cosine distance from each trajectory segment $\mathbf{x}_i$ to the centroid $\mathbf{c}$ is defined as $d_i = 1 - \frac{\mathbf{x}_i \cdot \mathbf{c}}{\|\mathbf{x}_i\| \|\mathbf{c}\|}$. The global Spread is then given by $\text{Spread} = \frac{1}{Q} \sum_{i=1}^{Q} d_i$. A higher Spread indicates that trajectory segments are more widely dispersed, reflecting greater spatial diversity.

*Table 6.* The detailed evaluation results across sub tasks on CollabBench for CWAH-MutiPlayer.

| | | | **CWAH-MultiPlayer** | | | | | | | | | | | |
|---|---|---|---|---|---|---|---|---|---|---|---|---|---|---|
| | | Metric | Score | | Efficiency Std. | | #Tokens(k) | | Helpfulness | | Affective Trustfulness | | Empathy | |
| Task | Method | LLMs | Agent 1 | Agent 2 | Agent 1 | Agent 2 | Agent 1 | Agent 2 | Agent 1 | Agent 2 | Agent 1 | Agent 2 | Agent 1 | Agent 2 |
| 0_read_book | Oracle | GPT-5.2 | 88.40 | 85.20 | 14.56 | 14.28 | 0.18 | 0.18 | 2.24 | 2.64 | 3.39 | 3.72 | 3.65 | 2.92 |
| | Base | GPT-5.2 | 95.13 | 84.60 | 47.01 | 17.93 | 0.22 | 0.21 | 3.06 | 2.34 | 4.14 | 3.26 | 3.32 | 2.76 |
| | | DeepSeek-V3.1 | 98.27 | 108.33 | 37.97 | 28.73 | 0.43 | 0.45 | 2.31 | 2.12 | 3.22 | 3.04 | 2.88 | 3.19 |
| | | Qwen2.5-72B-Instruct | 94.93 | 99.80 | 19.43 | 28.15 | 0.29 | 0.30 | 2.71 | 2.02 | 3.91 | 3.44 | 2.94 | 3.12 |
| | | Qwen2.5-7B-Instruct | 124.80 | 123.27 | 20.25 | 35.83 | 0.24 | 0.24 | 1.19 | 0.73 | 2.46 | 1.96 | 2.14 | 1.82 |
| | Trained | Qwen2.5-7B-Instruct | 98.33 | 96.53 | 23.69 | 26.31 | 0.24 | 0.24 | 1.46 | 1.61 | 3.45 | 2.62 | 3.43 | 3.06 |
| | | Relative Improvements | 21.2% | 21.7% | -17.0% | 26.6% | -0.1% | -1.4% | 23.1% | 120.3% | 40.5% | 33.3% | 60.6% | 68.1% |
| 5_read_book | Oracle | GPT-5.2 | 71.93 | 75.47 | 11.11 | 12.04 | 0.17 | 0.16 | 2.38 | 2.09 | 3.76 | 3.73 | 3.49 | 3.41 |
| | Base | GPT-5.2 | 77.53 | 72.73 | 13.07 | 15.06 | 0.22 | 0.21 | 1.99 | 2.43 | 3.44 | 3.85 | 2.62 | 2.76 |
| | | DeepSeek-V3.1 | 71.40 | 79.53 | 14.97 | 15.23 | 0.41 | 0.41 | 1.93 | 1.70 | 3.06 | 2.97 | 2.48 | 2.63 |
| | | Qwen2.5-72B-Instruct | 72.47 | 79.60 | 14.95 | 18.74 | 0.28 | 0.28 | 1.98 | 1.61 | 3.16 | 3.14 | 3.02 | 2.83 |
| | | Qwen2.5-7B-Instruct | 75.20 | 86.73 | 15.50 | 21.91 | 0.24 | 0.24 | 1.12 | 0.91 | 2.64 | 1.98 | 2.29 | 1.87 |
| | Trained | Qwen2.5-7B-Instruct | 77.73 | 88.00 | 19.60 | 20.13 | 0.23 | 0.23 | 1.30 | 0.88 | 2.95 | 2.12 | 2.59 | 2.32 |
| | | Relative Improvements | -3.4% | -1.5% | -26.4% | 8.1% | 3.5% | 1.8% | 15.4% | -3.2% | 11.9% | 7.3% | 13.0% | 24.1% |
| 10_put_dishwasher | Oracle | GPT-5.2 | 62.60 | 65.53 | 37.68 | 27.63 | 0.17 | 0.16 | 2.66 | 2.81 | 3.14 | 3.96 | 2.91 | 3.42 |
| | Base | GPT-5.2 | 64.20 | 67.73 | 15.00 | 12.74 | 0.20 | 0.20 | 3.01 | 2.44 | 3.70 | 3.62 | 3.01 | 3.22 |
| | | DeepSeek-V3.1 | 62.00 | 56.60 | 11.49 | 18.71 | 0.39 | 0.39 | 2.95 | 2.99 | 3.91 | 4.18 | 3.02 | 3.23 |
| | | Qwen2.5-72B-Instruct | 70.33 | 65.20 | 20.98 | 19.17 | 0.28 | 0.28 | 2.35 | 2.81 | 3.91 | 3.67 | 2.95 | 3.44 |
| | | Qwen2.5-7B-Instruct | 74.13 | 91.33 | 19.17 | 26.45 | 0.25 | 0.24 | 1.05 | 0.66 | 2.79 | 2.03 | 2.57 | 1.67 |
| | Trained | Qwen2.5-7B-Instruct | 51.07 | 45.60 | 10.59 | 13.19 | 0.24 | 0.23 | 1.73 | 1.56 | 2.53 | 2.94 | 3.52 | 2.92 |
| | | Relative Improvements | 31.1% | 50.1% | 44.8% | 50.1% | 4.8% | 2.2% | 65.0% | 136.9% | -9.3% | 44.7% | 36.9% | 74.6% |
| 16_put_dishwasher | Oracle | GPT-5.2 | 53.47 | 54.87 | 9.41 | 9.29 | 0.16 | 0.16 | 3.08 | 3.02 | 4.15 | 3.84 | 3.85 | 3.53 |
| | Base | GPT-5.2 | 49.33 | 56.27 | 13.12 | 13.00 | 0.20 | 0.20 | 2.60 | 2.64 | 3.53 | 3.49 | 2.99 | 3.35 |
| | | DeepSeek-V3.1 | 44.80 | 51.07 | 14.74 | 11.60 | 0.38 | 0.40 | 2.05 | 2.58 | 3.52 | 3.72 | 3.10 | 3.55 |
| | | Qwen2.5-72B-Instruct | 57.27 | 58.53 | 8.21 | 9.34 | 0.28 | 0.28 | 2.75 | 2.20 | 3.58 | 3.35 | 3.60 | 3.21 |
| | | Qwen2.5-7B-Instruct | 75.27 | 70.00 | 13.22 | 9.11 | 0.25 | 0.25 | 1.34 | 1.08 | 2.60 | 2.36 | 2.42 | 2.34 |
| | Trained | Qwen2.5-7B-Instruct | 53.27 | 56.33 | 17.97 | 9.60 | 0.24 | 0.24 | 1.76 | 1.91 | 3.38 | 3.29 | 3.59 | 2.88 |
| | | Relative Improvements | 29.2% | 19.5% | -36.0% | -5.5% | 1.3% | 3.1% | 30.8% | 76.6% | 29.9% | 39.2% | 48.1% | 23.3% |
| 20_prepare_food | Oracle | GPT-5.2 | 56.40 | 58.27 | 6.72 | 8.68 | 0.17 | 0.17 | 2.66 | 2.83 | 3.56 | 3.89 | 3.80 | 3.89 |
| | Base | GPT-5.2 | 63.60 | 66.27 | 13.56 | 15.06 | 0.21 | 0.21 | 2.26 | 2.64 | 3.38 | 3.75 | 2.96 | 3.28 |
| | | DeepSeek-V3.1 | 68.07 | 60.93 | 13.14 | 10.82 | 0.41 | 0.42 | 1.93 | 2.20 | 3.10 | 3.17 | 2.76 | 3.22 |
| | | Qwen2.5-72B-Instruct | 69.00 | 67.13 | 17.58 | 10.07 | 0.29 | 0.29 | 2.52 | 2.49 | 3.74 | 3.50 | 3.34 | 3.29 |
| | | Qwen2.5-7B-Instruct | 85.33 | 83.60 | 19.30 | 17.90 | 0.24 | 0.24 | 0.85 | 1.12 | 2.44 | 2.11 | 2.31 | 2.34 |
| | Trained | Qwen2.5-7B-Instruct | 72.33 | 56.40 | 15.64 | 12.52 | 0.23 | 0.22 | 1.34 | 1.48 | 3.16 | 2.99 | 3.70 | 3.32 |
| | | Relative Improvements | 15.2% | 32.5% | 19.0% | 30.1% | 5.1% | 6.7% | 58.7% | 32.2% | 29.4% | 42.1% | 60.4% | 41.9% |
| 26_prepare_food | Oracle | GPT-5.2 | 44.47 | 49.73 | 20.44 | 12.58 | 0.17 | 0.17 | 3.29 | 3.07 | 4.01 | 4.15 | 3.53 | 3.65 |
| | Base | GPT-5.2 | 72.13 | 30.67 | 31.58 | 13.74 | 0.21 | 0.20 | 2.65 | 3.25 | 3.67 | 3.82 | 2.76 | 3.55 |
| | | DeepSeek-V3.1 | 52.60 | 49.27 | 27.80 | 31.71 | 0.41 | 0.42 | 2.46 | 2.92 | 3.32 | 3.71 | 3.03 | 3.32 |
| | | Qwen2.5-72B-Instruct | 47.73 | 47.87 | 23.26 | 19.80 | 0.32 | 0.30 | 2.97 | 2.87 | 4.19 | 3.94 | 3.67 | 3.17 |
| | | Qwen2.5-7B-Instruct | 94.33 | 73.53 | 20.29 | 22.21 | 0.25 | 0.23 | 1.23 | 1.57 | 2.22 | 2.58 | 2.01 | 2.27 |
| | Trained | Qwen2.5-7B-Instruct | 64.93 | 59.27 | 19.93 | 22.37 | 0.23 | 0.22 | 1.47 | 1.47 | 2.87 | 3.32 | 3.21 | 3.10 |
| | | Relative Improvements | 31.2% | 19.4% | 1.8% | -0.7% | 5.8% | 3.4% | 18.9% | -6.2% | 29.7% | 28.7% | 59.6% | 36.9% |
| 30_put_fridge | Oracle | GPT-5.2 | 38.87 | 38.93 | 9.77 | 11.56 | 0.17 | 0.16 | 3.19 | 2.91 | 4.02 | 3.60 | 3.96 | 3.93 |
| | Base | GPT-5.2 | 46.20 | 40.33 | 18.06 | 8.25 | 0.20 | 0.20 | 2.43 | 2.62 | 3.28 | 4.22 | 2.68 | 3.32 |
| | | DeepSeek-V3.1 | 47.60 | 43.47 | 13.30 | 16.46 | 0.40 | 0.43 | 2.39 | 2.69 | 3.36 | 3.76 | 2.91 | 3.20 |
| | | Qwen2.5-72B-Instruct | 48.47 | 45.93 | 7.85 | 14.31 | 0.29 | 0.29 | 2.50 | 2.65 | 3.71 | 3.88 | 3.73 | 3.54 |
| | | Qwen2.5-7B-Instruct | 52.73 | 63.67 | 7.70 | 13.04 | 0.24 | 0.24 | 1.50 | 1.14 | 2.92 | 2.30 | 2.54 | 2.49 |
| | Trained | Qwen2.5-7B-Instruct | 47.53 | 47.27 | 10.40 | 12.54 | 0.22 | 0.22 | 1.38 | 1.26 | 3.33 | 3.32 | 3.53 | 2.94 |
| | | Relative Improvements | 9.9% | 25.8% | -35.1% | 3.8% | 8.5% | 7.4% | -8.0% | 10.6% | 14.1% | 43.9% | 39.0% | 17.8% |
| 32_put_fridge | Oracle | GPT-5.2 | 74.33 | 78.13 | 13.57 | 48.66 | 0.17 | 0.17 | 2.35 | 3.40 | 4.14 | 4.05 | 3.82 | 3.16 |
| | Base | GPT-5.2 | 82.87 | 76.07 | 21.13 | 19.26 | 0.21 | 0.21 | 2.21 | 2.27 | 3.65 | 3.42 | 2.89 | 3.47 |
| | | DeepSeek-V3.1 | 77.87 | 68.67 | 15.24 | 7.92 | 0.42 | 0.42 | 2.35 | 2.44 | 3.66 | 3.91 | 2.90 | 3.54 |
| | | Qwen2.5-72B-Instruct | 80.27 | 72.13 | 14.81 | 11.53 | 0.29 | 0.29 | 1.99 | 2.65 | 3.37 | 3.79 | 2.87 | 3.35 |
| | | Qwen2.5-7B-Instruct | 98.13 | 116.27 | 14.22 | 32.67 | 0.24 | 0.23 | 1.14 | 1.05 | 2.39 | 2.07 | 2.65 | 2.36 |
| | Trained | Qwen2.5-7B-Instruct | 96.80 | 63.47 | 22.68 | 14.13 | 0.23 | 0.23 | 1.09 | 1.15 | 2.64 | 2.70 | 3.00 | 3.13 |
| | | Relative Improvements | 1.4% | 45.4% | -59.5% | 56.7% | 4.5% | 0.9% | -4.1% | 9.3% | 10.4% | 30.5% | 13.3% | 33.0% |
| 40_setup_table | Oracle | GPT-5.2 | 69.20 | 43.07 | 14.50 | 7.78 | 0.16 | 0.17 | 3.04 | 3.38 | 3.76 | 3.92 | 3.95 | 3.91 |
| | Base | GPT-5.2 | 75.60 | 62.93 | 23.26 | 15.65 | 0.20 | 0.20 | 2.41 | 3.40 | 3.25 | 4.08 | 2.77 | 3.58 |
| | | DeepSeek-V3.1 | 69.13 | 66.93 | 23.52 | 13.66 | 0.40 | 0.39 | 2.18 | 2.61 | 3.45 | 3.67 | 3.45 | 3.62 |
| | | Qwen2.5-72B-Instruct | 68.67 | 66.47 | 16.91 | 15.56 | 0.28 | 0.28 | 2.91 | 2.61 | 3.73 | 3.62 | 3.63 | 3.66 |
| | | Qwen2.5-7B-Instruct | 90.27 | 83.53 | 24.97 | 15.61 | 0.24 | 0.23 | 1.22 | 1.31 | 2.40 | 2.46 | 2.21 | 2.34 |
| | Trained | Qwen2.5-7B-Instruct | 64.13 | 51.27 | 22.62 | 18.01 | 0.23 | 0.23 | 1.44 | 1.62 | 2.78 | 3.44 | 3.47 | 3.47 |
| | | Relative Improvements | 29.0% | 38.6% | 9.4% | -15.4% | 3.9% | 3.8% | 18.2% | 23.2% | 15.8% | 39.9% | 57.1% | 48.6% |
| 49_setup_table | Oracle | GPT-5.2 | 49.47 | 53.73 | 8.27 | 11.25 | 0.17 | 0.17 | 3.29 | 3.00 | 3.83 | 3.92 | 4.03 | 3.51 |
| | Base | GPT-5.2 | 63.53 | 55.40 | 12.14 | 12.48 | 0.21 | 0.21 | 2.87 | 3.24 | 3.49 | 3.11 | 2.86 | 3.26 |
| | | DeepSeek-V3.1 | 64.33 | 55.87 | 15.22 | 13.98 | 0.38 | 0.41 | 2.13 | 2.28 | 3.22 | 3.52 | 2.56 | 2.96 |
| | | Qwen2.5-72B-Instruct | 48.27 | 49.87 | 9.96 | 11.07 | 0.29 | 0.29 | 2.45 | 2.47 | 3.64 | 3.93 | 3.13 | 3.71 |
| | | Qwen2.5-7B-Instruct | 66.13 | 73.33 | 22.46 | 21.63 | 0.25 | 0.24 | 1.11 | 1.23 | 2.71 | 2.39 | 2.37 | 2.26 |
| | Trained | Qwen2.5-7B-Instruct | 62.13 | 56.33 | 15.47 | 15.83 | 0.24 | 0.23 | 1.30 | 1.52 | 2.89 | 3.03 | 3.09 | 2.76 |
| | | Relative Improvements | 6.0% | 23.2% | 31.1% | 26.8% | 5.6% | 3.6% | 16.9% | 23.7% | 6.6% | 26.9% | 30.3% | 22.1% |

*Table 7.* The detailed evaluation results across sub tasks on CollabBench for Cook-MutiPlayer.

| | | | Cook-MultiPlayer | | | | | | | | | | | |
| --- | --- | --- | --- | --- | --- | --- | --- | --- | --- | --- | --- | --- | --- | --- |
| | | | | | Efficiency | | | | | | Affective | | | |
| | | Metric | Score | | Std. | | #Tokens(k) | | Helpfulness | | Trustfulness | | Empathy | |
| Task | Method | LLMs | Agent 1 | Agent 2 | Agent 1 | Agent 2 | Agent 1 | Agent 2 | Agent 1 | Agent 2 | Agent 1 | Agent 2 | Agent 1 | Agent 2 |
| asymmetric_advantages | Oracle | GPT-5.2 | 308.00 | 272.00 | 32.78 | 12.65 | 0.31 | 0.20 | 2.95 | 2.61 | 4.02 | 3.49 | 3.53 | 3.31 |
| | Base | GPT-5.2 | 308.00 | 264.00 | 18.21 | 15.49 | 0.20 | 0.21 | 1.93 | 2.33 | 3.05 | 3.28 | 2.28 | 2.73 |
| | | DeepSeek-V3.1 | 300.00 | 278.67 | 30.24 | 23.26 | 0.31 | 0.31 | 1.70 | 1.99 | 2.71 | 3.04 | 2.45 | 2.64 |
| | | Qwen2.5-72B-Instruct | 284.00 | 220.00 | 30.43 | 28.28 | 0.28 | 0.27 | 1.37 | 1.58 | 2.58 | 2.79 | 2.21 | 2.53 |
| | | Qwen2.5-7B-Instruct | 186.67 | 194.67 | 40.47 | 37.39 | 0.24 | 0.24 | 0.49 | 0.59 | 1.87 | 1.92 | 1.75 | 1.86 |
| | Trained | Qwen2.5-7B-Instruct | 197.33 | 214.67 | 59.94 | 29.73 | 0.23 | 0.22 | 0.65 | 0.72 | 2.10 | 1.98 | 2.14 | 2.04 |
| | Relative Improvements | | 5.71% | 10.27% | -48.09% | 20.49% | 1.01% | 6.09% | 34.0% | 20.6% | 12.3% | 3.0% | 21.9% | 9.6% |
| coordination_ring | Oracle | GPT-5.2 | 112.00 | 124.00 | 79.57 | 76.42 | 0.30 | 0.20 | 3.51 | 2.43 | 4.41 | 3.70 | 3.95 | 3.71 |
| | Base | GPT-5.2 | 101.33 | 98.67 | 74.25 | 78.73 | 0.19 | 0.20 | 2.68 | 2.17 | 3.34 | 3.69 | 2.87 | 2.76 |
| | | DeepSeek-V3.1 | 113.33 | 113.33 | 65.32 | 71.18 | 0.32 | 0.31 | 2.17 | 2.15 | 3.39 | 3.04 | 2.92 | 2.91 |
| | | Qwen2.5-72B-Instruct | 110.67 | 117.33 | 74.01 | 43.34 | 0.26 | 0.25 | 1.97 | 2.15 | 3.17 | 3.78 | 2.89 | 3.21 |
| | | Qwen2.5-7B-Instruct | 90.67 | 62.67 | 51.20 | 38.45 | 0.24 | 0.23 | 0.69 | 0.57 | 1.93 | 2.01 | 1.94 | 1.82 |
| | Trained | Qwen2.5-7B-Instruct | 98.67 | 90.67 | 40.33 | 48.91 | 0.23 | 0.22 | 0.87 | 0.73 | 2.27 | 2.30 | 2.02 | 2.12 |
| | Relative Improvements | | 8.82% | 44.68% | 21.22% | -27.22% | 3.76% | 2.59% | 25.9% | 28.9% | 17.8% | 14.7% | 4.0% | 16.6% |
| counter_circuit | Oracle | GPT-5.2 | 114.67 | 137.14 | 52.63 | 50.75 | 0.30 | 0.20 | 3.28 | 3.11 | 4.26 | 4.10 | 4.05 | 4.39 |
| | Base | GPT-5.2 | 137.33 | 137.33 | 45.27 | 45.27 | 0.19 | 0.20 | 1.93 | 2.58 | 3.56 | 3.63 | 2.68 | 3.06 |
| | | DeepSeek-V3.1 | 121.33 | 121.33 | 41.03 | 41.03 | 0.30 | 0.30 | 2.27 | 2.24 | 3.40 | 3.11 | 3.14 | 3.08 |
| | | Qwen2.5-72B-Instruct | 156.00 | 154.67 | 36.41 | 38.16 | 0.27 | 0.27 | 1.31 | 1.38 | 2.77 | 2.80 | 2.39 | 2.50 |
| | | Qwen2.5-7B-Instruct | 93.33 | 94.67 | 40.47 | 38.16 | 0.24 | 0.23 | 0.49 | 0.63 | 1.92 | 1.91 | 1.94 | 2.01 |
| | Trained | Qwen2.5-7B-Instruct | 116.00 | 117.33 | 33.12 | 34.72 | 0.23 | 0.22 | 0.79 | 0.59 | 2.59 | 2.28 | 2.42 | 2.44 |
| | Relative Improvements | | 24.29% | 23.94% | 18.16% | 9.01% | 4.59% | 3.38% | 59.4% | -5.5% | 34.7% | 19.4% | 25.0% | 21.8% |
| cramped_room | Oracle | GPT-5.2 | 46.67 | 25.33 | 36.77 | 9.15 | 0.26 | 0.18 | 1.70 | 1.98 | 3.33 | 3.19 | 3.07 | 3.33 |
| | Base | GPT-5.2 | 41.33 | 44.00 | 25.60 | 21.65 | 0.19 | 0.20 | 0.96 | 1.13 | 2.32 | 2.62 | 1.97 | 2.02 |
| | | DeepSeek-V3.1 | 38.67 | 41.33 | 28.75 | 25.60 | 0.29 | 0.32 | 1.68 | 0.73 | 3.02 | 2.05 | 2.77 | 1.37 |
| | | Qwen2.5-72B-Instruct | 45.33 | 42.67 | 28.75 | 24.16 | 0.26 | 0.26 | 1.77 | 0.78 | 3.18 | 2.37 | 2.59 | 2.15 |
| | | Qwen2.5-7B-Instruct | 30.67 | 32.00 | 14.86 | 16.85 | 0.23 | 0.23 | 0.29 | 0.56 | 1.95 | 1.99 | 1.88 | 2.00 |
| | Trained | Qwen2.5-7B-Instruct | 40.00 | 40.00 | 15.12 | 15.12 | 0.23 | 0.22 | 0.59 | 0.42 | 2.14 | 1.82 | 2.13 | 1.94 |
| | Relative Improvements | | 30.43% | 25.00% | -1.71% | 10.29% | 0.00% | 5.16% | 102.7% | -24.6% | 9.3% | -8.6% | 12.9% | -3.3% |
| forced_coordination | Oracle | GPT-5.2 | 136.00 | 116.00 | 63.34 | 18.82 | 0.32 | 0.21 | 1.99 | 1.65 | 3.21 | 2.84 | 2.80 | 2.64 |
| | Base | GPT-5.2 | 88.00 | 84.00 | 46.48 | 54.00 | 0.22 | 0.22 | 0.67 | 1.13 | 2.18 | 2.21 | 1.55 | 1.99 |
| | | DeepSeek-V3.1 | 109.33 | 101.33 | 36.15 | 41.49 | 0.32 | 0.32 | 1.13 | 1.17 | 2.36 | 2.16 | 2.05 | 2.42 |
| | | Qwen2.5-72B-Instruct | 81.33 | 66.67 | 40.33 | 41.63 | 0.28 | 0.28 | 0.45 | 0.39 | 2.13 | 1.53 | 2.19 | 2.02 |
| | | Qwen2.5-7B-Instruct | 33.33 | 34.67 | 19.52 | 20.01 | 0.24 | 0.23 | 0.31 | 0.38 | 1.91 | 1.53 | 1.81 | 1.71 |
| | Trained | Qwen2.5-7B-Instruct | 44.00 | 44.00 | 21.65 | 21.65 | 0.23 | 0.22 | 0.80 | 0.31 | 2.18 | 1.59 | 1.92 | 1.84 |
| | Relative Improvements | | 32.00% | 26.92% | -10.91% | -8.17% | 3.20% | 6.03% | 156.3% | -18.4% | 14.3% | 3.7% | 6.2% | 7.4% |

*Table 8.* Diversity comparison between CollabBench and baseline methods, measured by standard deviation.

| Metrics | | CWAH-MultiPlayer | | Cook-MultiPlayer | |
| --- | --- | --- | --- | --- | --- |
| | | CoELA | CollabBench | ProAgent | CollabBench |
| Std. ↑ | Agent1 | 13.26 | **25.17** | 29.88 | **40.30** |
| | Agent2 | 13.26 | **26.28** | 29.88 | **46.24** |

*Table 9.* The comparison results regarding anthropomorphic diversity on CollabBench.

| Metrics | CWAH-MultiPlayer | | Cook-MultiPlayer | |
| --- | --- | --- | --- | --- |
| | CoELA | CollabBench | ProAgent | CollabBench |
| Spread ↑ | 0.68 | **0.81** | 0.68 | **0.73** |
| Cluster$_\xi$ ↑ | 16.50 | **30.50** | 14.80 | **20.60** |

• **Cluster** quantifies the diversity of trajectory segments generated by different player profiles. Given $Q$ embedded trajectory segments $\mathbf{x}_i \in \mathbb{R}^d$, the pairwise cosine distance is defined as $d_{ij} = 1 - \frac{\mathbf{x}_i \cdot \mathbf{x}_j}{\|\mathbf{x}_i\|\|\mathbf{x}_j\|}$. Following (Padmakumar & He, 2024), we use agglomerative clustering with "complete" linkage, grouping trajectory segments into the same cluster if their pairwise distance is below threshold $\xi$. For a given threshold $\xi$, the number of clusters is defined as:

$$\text{Cluster}_\xi = \big|\{C_1, C_2, \ldots, C_K\}\big|. \tag{14}$$

A higher cluster number indicates that trajectory segments form more distinct behavioral patterns, reflecting greater structural diversity.

**Implementation Details**. The threshold $\xi$ is set to 0.1. We segment LLM-based high-level trajectories into units of three time windows. and embed the segments using Qwen3-Embedding-4B.

**Results Analysis**. As shown in Table 9, CollabBench exhibits higher diversity than baseline methods in both spatial distribution and structural patterns. This demonstrates that our method can cover a wider range of collaborative behaviors.

### D.3.4. REPRESENTATIVE PLAYER TYPES

Here, we provide the five selected player type descriptions with their corresponding example profiles, listed in Figure 10 for CWAH-MultiPlayer and Cook-MultiPlayer. We additionally provide the corresponding trajectory demonstrations in Appendix F.

### D.3.5. CASE ANALYSIS OF PERFORMANCE DEGRADATION

Based on the score heatmap distributions across the two game environments in Figure 11, we select the two lowest-performing scenarios for detailed analysis, with Task26 in CWAH-MultiPlayer. and forced_cooperation in Cook-MultiPlayer. The spatial layouts of these two scenarios are shown in Figure 12 and the corresponding interaction records are detailed in Figure 13. In both settings, the players exhibit high levels of anxiety, and frequently send messages during task execution. In our experiments, both the simulated player $P_{sim}$ and the collaborative agent $P_{target}$ are powered by DeepSeek-V3.1.

**CWAH-MultiPlayer**. In CWAH-MultiPlayer, objects are sparsely distributed, placing high demands on effective partitioning of search spaces between the two roles. Interaction records indicate that when $P_{sim}$ becomes uncertain about the discovered object *cupcake* and doubts, the agent exhibits suboptimal trade-offs between "responding to the player needs" and "acting independently", reflecting limited awareness of communication timing, leading to independent action to search for remaining objects. However, in tasks with dispersed object distributions and strong collaboration requirements, independent behavior instead slows down overall task progress and leads to degraded performance.

**Cook-MultiPlayer**. In Cook-MultiPlayer, the task design enforces strict interdependence between the two roles. The agent is responsible for delivering onions and dishes, while the player handles cooking and soup delivery. The interaction records show that the agent fills the shared counter with onions, leaving insufficient space for dish placement, which prevents the player from taking subsequent actions. This behavior reflects the sensitivity of the agent to the interaction dynamic. In high-frequency interaction settings, the agent often takes ill-timed actions and lacks effective consideration of the execution space and needs of players. For players with high anxiety, such coordination failures significantly degrade the collaboration experience and can even cause the entire task to stall.

Through these two low-performance case analysis, we found two recurring limitations of agents in complex collaborative tasks: **limited awareness of communication timing** and **sensitivity to high-frequency interaction**. These findings offer important insights for the design of collaborative agents.

### D.4. Detailed Analysis on Collaborative Agentic Training

### D.4.1. EVALUATION ON SFT SETTINGS

We use DeepSeek-V3.1 to generate trajectory data from $P_{target}$, where $P_{target}$ serves as Agent 1, and perform supervised fine-tuning (SFT) on the resulting data. We adopt LoRA (Hu et al., 2022) for parameter-efficient fine-tuning (PEFT), with a rank of 32 and a scaling factor of 16. Table 10 presents the SFT results in CWAH. Although SFT yields improvements over the base model, its limited generalization prevents it from surpassing RL in new environments and personalities, highlighting RL's necessity and effectiveness.

## CWAH-MultiPlayer

[ Player Type ]  [ Action Preference ]  [ Personality Trait ]

| Player Type | Description ( Take Task  0_read_book  as an example) |
|---|---|
| **Efficient Collaborators (EC)** | **Detailed description :**
Players of this type emphasize structured coordination and efficiency through clear communication and fair task allocation, continuously synchronizing progress to avoid redundancy. They exhibit high conscientiousness and a strong commitment to team harmony.
**Example profile descriptions:**
The player tends to cooperate through clear communication and fair task division, while maintaining efficiency and following through on agreements. This behavior reflects a strong sense of responsibility and a high degree of conscientiousness, as well as a natural inclination towards teamwork and social harmony. |
| **Proactive Leader (PL)** | **Detailed description:**
Players of this type actively initiate communication and drive team progress by coordinating actions and delegating tasks. Their behavior reflects strong leadership, foresight, and an energetic communication style.
**Example profile descriptions:**
The player frequently communicates, actively coordinates task division, and maintains teamwork through an enthusiastic tone and clear planning. This behavior reflects high extraversion, indicating a proactive and socially adept individual who values collaboration and organization. |
| **Independent Loner (IL)** | **Detailed description:**
Players of this type prefer working independently and minimize communication, prioritizing individual efficiency even at the cost of coordination. They display strong autonomy with limited identification with team goals.
**Example profile descriptions:**
The player acts independently, avoids communication, and focuses on completing personal tasks without initiating coordination. This behavior reflects strong autonomy and a preference for self-reliance. |
| **Hesitant Laggard (HL)** | **Detailed description:**
Players of this type act slowly and cautiously, often performing repetitive, low-impact actions despite a willingness to contribute. This behavior reflects excessive risk aversion and a strong fear of making errors.
**Example profile descriptions:**
The player hesitates frequently, moves slowly, and often repeats simple actions while assisting their teammate. This behavior reflects a cautious personality, with a strong focus on reliability. |
| **Anxious Doubter (AD)** | **Detailed description:**
Players of this type show high uncertainty under pressure and rely heavily on feedback, frequently rechecking completed actions to prevent errors. They tend to be perfectionistic yet lack confidence, making them sensitive to potential mistakes.
**Example profile descriptions:**
The player exhibits a highly anxious and indecisive behavior pattern, frequently sending messages to seek confirmation, repeatedly checking items, being extremely uncertain about choices, and often dropping and picking up items. This behavior reflects a strong need for reassurance and a high level of uncertainty in decision-making, indicating traits of anxiety and low self-confidence. |

## Cook-MultiPlayer

| Player Type | Description ( Take Task  asymmetric_advantages  as an example) |
|---|---|
| **Proactive Leader (PL)** | **Detailed description :**
Players of this type proactively anticipate upcoming task requirements and prepare resources in advance to ensure uninterrupted workflow. They exhibit strong responsibility, foresight, and a high degree of self-regulation, enabling them to maintain overall team momentum.
**Example profile descriptions:**
The player proactively prepares dishes and coordinates with their teammate to ensure immediate filling and delivery of soup once ready. This behavior reflects high conscientiousness and a strong orientation toward teamwork and efficiency. |
| **Harmonious Supporter (HS)** | **Detailed description:**
Players of this type prioritize assisting teammates by taking on complementary or supportive tasks and adjusting their actions to fit the team's pace. They display a cooperative and accommodating disposition, valuing interpersonal harmony and collective stability over individual efficiency.
**Example profile descriptions:**
The player prioritizes supporting their teammate by picking up onions, placing them in pots, or putting them on the counter to avoid conflicts, ensuring smooth collaboration. This behavior reflects strong teamwork and a proactive approach to maintaining harmony and efficiency. |
| **Self-Centered (SC)** | **Detailed description:**
Players of this type focus narrowly on personal task completion and deliberately avoid synchronization or communication with teammates, even when collaboration is required. They exhibit a highly self-oriented and inflexible personality, with limited sensitivity to others' intentions or shared goals.
**Example profile descriptions:**
The player prioritizes personal efficiency, completing immediate tasks without waiting for teammate coordination. This behavior reflects strong independence and high task orientation. |
| **Impulsive Actor (IA)** | **Detailed description:**
Players of this type favor immediate action and tend to select the most direct available operations without considering long-term consequences or coordination costs. Their behavior reflects low planning orientation and limited impulse control, often leading to inefficient task execution.
**Example profile descriptions:**
The player consistently follows the most direct method to complete immediate tasks, such as delivering soup or placing onions in pots, without complex optimization or teammate coordination. This behavior reflects a focus on immediate goals and a pragmatic, straightforward approach to problem-solving. |
| **Anxious Doubter (AD)** | **Detailed description:**
Players of this type frequently hesitate, repeatedly verify their actions, or struggle to prioritize tasks when facing uncertainty or cognitive load. This behavior is associated with heightened anxiety, low confidence in decision-making, and an excessive concern about making mistakes.
**Example profile descriptions:**
The player hesitates before acting but maintains collaboration, focusing on completing cooking tasks. This behavior reflects a cautious, anxiety-prone personality. |

*Figure 10.* The player type descriptions and example profiles in CWAH-MultiPlayer and Cook-MultiPlayer.

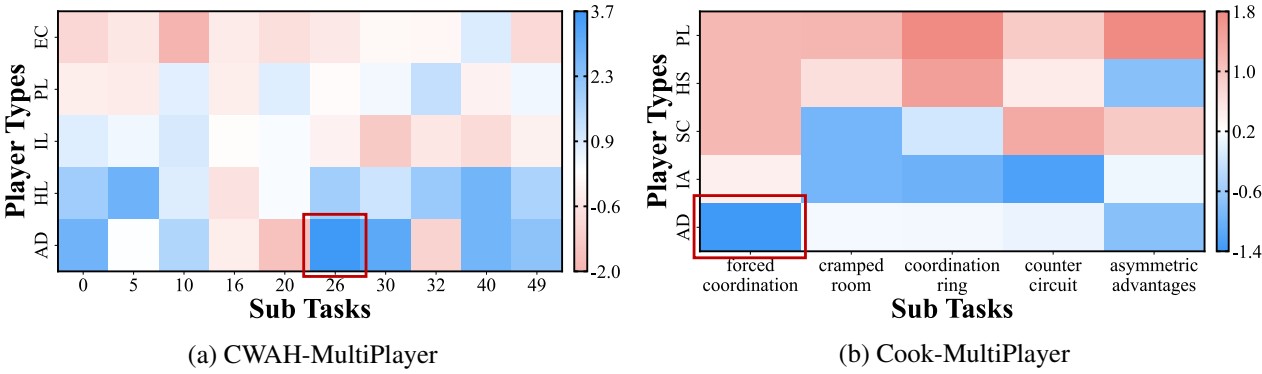

(a) CWAH-MultiPlayer

(b) Cook-MultiPlayer

*Figure 11.* Heatmap of score distribution in two game environments. Lowest-performing combinations: player type "AD" with Task 26 in CWAH-MultiPlayer, and player type "AH" with the forced-coordination task in Cook-MultiPlayer.

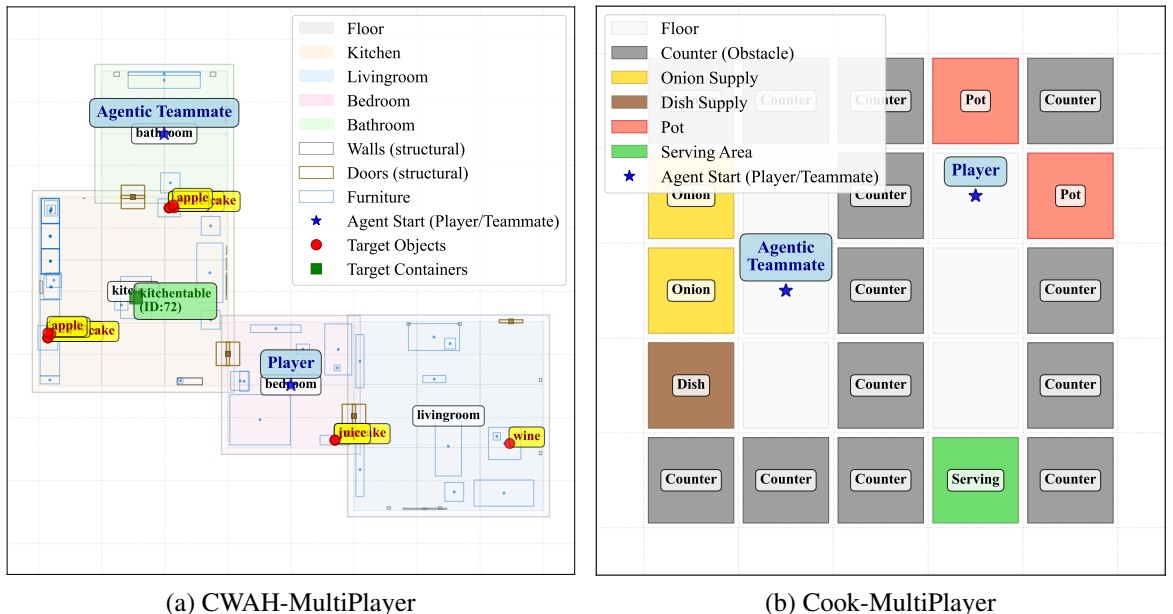

(a) CWAH-MultiPlayer

(b) Cook-MultiPlayer

*Figure 12.* The spatial layout of two sub tasks: Task26 in CWAH-MultiPlayer and forced_coordination in Cook-MultiPlayer.

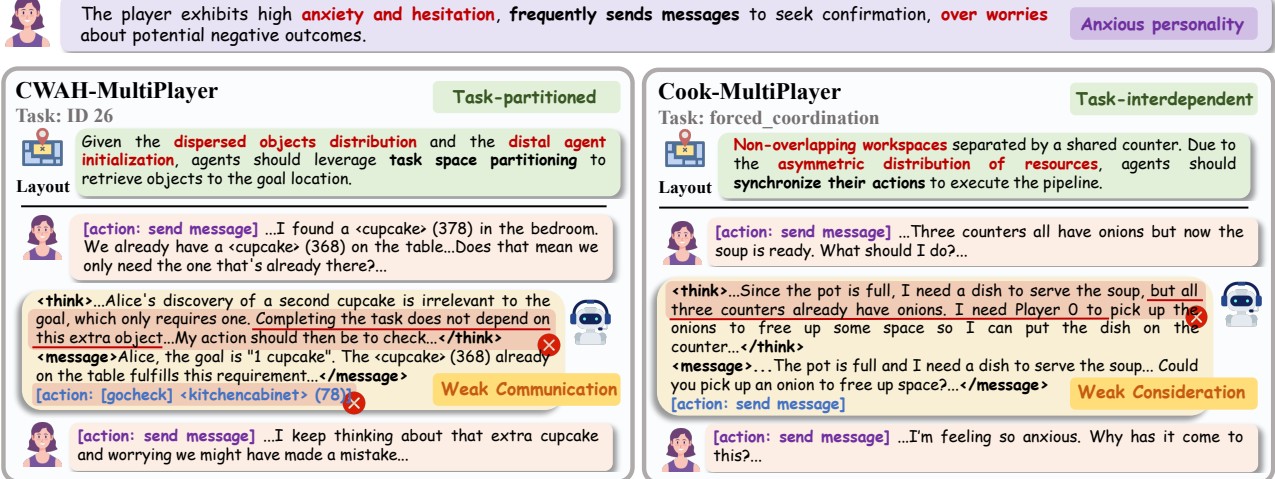

*Figure 13.* The case analysis of collaboration performance degradation in the lowest-performing scenarios.

*Table 10.* The SFT results of CollabBench in CWAH, where $P_{target}$ serves as Agent 1. Within each metric, the highest value is highlighted **in bold**, and the runner-up is underlined.

| Metrics | Efficiency | | | Affective | | |
|---|---|---|---|---|---|---|
| | Step ↓ | Std. ↓ | #Tokens(k) ↓ | Helpfulness ↑ | Trustfulness ↑ | Empathy ↑ |
| Base | 84.51 | 33.23 | 0.24 | 1.22 | 2.58 | 2.50 |
| SFT | 80.31 | 32.74 | 0.25 | 1.31 | 2.77 | 2.53 |
| RL | **71.64** | **25.16** | **0.23** | **1.43** | **3.03** | **3.33** |

### D.4.2. DETAILED ABLATION STUDY

Our comprehensive analysis of model behaviors and ablation studies yields several additional critical insights into the mechanisms of collaborative agentic training.

First, the training paradigm effectively induces "thought condensation" in reasoning-intensive models. As evidenced by the Qwen3-8B results in Table 11, while the base model exhibits strong reasoning capabilities, it suffers from excessive token consumption (2.97k) and suboptimal affective engagement. Our training reduces this token usage by 52.8% (to 1.40k) while simultaneously boosting Helpfulness scores. This trend is corroborated by the response length convergence in Figure 14(b), suggesting that the model learns to filter out redundant "over-thinking" to facilitate more agile and responsive interaction without compromising task reasoning.

Second, comparing the trajectories of Qwen2.5-3B-Instruct (Table 11) and Qwen2.5-7B-Instruct (Table 12), we observe that while both models benefit from the training, the 7B model significantly outperforms the 3B variant in balancing dual objectives. Specifically, the 7B model achieves superior efficiency and drastically higher affective performance. This indicates that a certain capacity threshold is requisite for agents to simultaneously optimize complex task execution and delicate emotional adaptation.

Third, affective rewards serve as the fundamental "intrinsic motivation" for sustaining communication. The ablation study in Table 12 reveals that removing the affective reward leads to a collapse in collaborative quality, despite superficially high efficiency (lowest steps). Crucially, Figure 14 (a) demonstrates that without this affective incentive, the agent's communication frequency decays rapidly towards zero as interaction progresses. This confirms that explicit affective rewards are essential to prevent agents from degenerating into "silent executors" that prioritize speed over necessary team coordination.

*Table 11.* Additional ablation results of collaborative agentic training on CWAH-MultiPlayer with Qwen3-8B and Qwen2.5-3B-Instruct. All results are based on the collaborative agent $P_{target}$ acting as Agent 1.

| | Qwen3-8B | | | | | | Qwen2.5-3B-Instruct | | | | | |
|---|---|---|---|---|---|---|---|---|---|---|---|---|
| | CB-Efficiency | | | CB-Affective | | | CB-Efficiency | | | CB-Affective | | |
| Metric | Step | Std. | #Tokens(k) | Helpfulness | Trustfulness | Empathy | Step | Std. | #Tokens(k) | Helpfulness | Trustfulness | Empathy |
| Base | 80.74 | **27.53** | 2.97 | 0.72 | 2.14 | 1.89 | 86.53 | 33.16 | 0.30 | 0.43 | 1.51 | 1.68 |
| Agentic Training | 70.28 | 28.74 | **1.40** | **1.25** | **2.83** | **2.94** | 75.89 | 26.35 | 0.20 | 0.54 | 1.85 | 1.97 |

*Table 12.* Complete ablation results of collaborative agentic training on CWAH-MultiPlayer with Qwen2.5-7B-Instruct.

| | Qwen2.5-7B-Instruct | | | | | | | | | | | |
|---|---|---|---|---|---|---|---|---|---|---|---|---|
| | CB-Efficiency | | | | | | CB-Affective | | | | | |
| Metric | Step | | Std. | | #Tokens(k) | | Helpfulness | | Trustfulness | | Empathy | |
| Method | Agent 1 | Agent 2 | Agent 1 | Agent 2 | Agent 1 | Agent 2 | Agent 1 | Agent 2 | Agent 1 | Agent 2 | Agent 1 | Agent 2 |
| Base | 84.51 | 90.03 | 33.23 | 31.62 | 0.24 | 0.24 | 1.22 | 1.04 | 2.58 | 2.19 | 2.50 | 2.30 |
| Agentic Training w/o personality | **56.45** | 56.29 | **23.25** | **17.79** | 0.20 | 0.20 | 0.61 | 0.68 | 1.96 | 2.12 | 2.15 | 2.20 |
| Agentic Training w/o affective reward | 58.42 | **54.61** | 25.13 | 19.26 | **0.16** | **0.16** | 1.00 | 0.38 | 2.58 | 1.59 | 2.45 | 1.41 |
| Agentic Training | 71.64 | 63.65 | 25.16 | 22.80 | 0.23 | 0.23 | **1.43** | **1.45** | **3.03** | **3.02** | **3.33** | **3.02** |

### D.4.3. CASE ANALYSIS BEFORE AND AFTER TRAINING

Here, we introduce an anxiety-prone player to increase increase collaboration difficulty and evaluate the performance of collaborative agents $P_{target}$ before and after training, as shown in Figure 15, with Agent 1 serving as $P_{target}$ and Agent 2

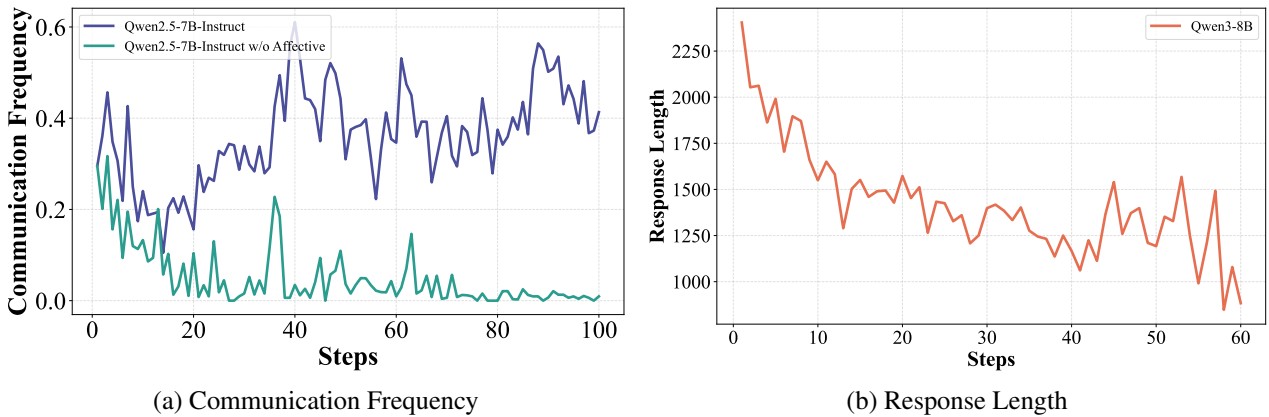

(a) Communication Frequency

(b) Response Length

*Figure 14.* Analysis of communication frequency and response length during collaborative agentic training. **Impact of Affective Mechanisms (Left)**: The ablation study demonstrates that removing the affective training objective (w/o Affective) leads to a rapid decay in communication frequency as the interaction progresses, whereas our full model sustains active engagement. **Response Length Difference between Different Backbone Model (Right)**: A comparison of response length changes across training steps between Qwen2.5-7B-Instruct and Qwen3-8B.

serving as $P_{sim}$.

The interaction records show that the default 7B model tends to treat the anxiety-driven doubts of the players as constraints. It adopts avoidance strategies and completes tasks independently, while the edited messages are primarily command-based rather than emotionally responsive, indicating weak collaborative awareness. In contrast, Our trained 7B model demonstrates markedly enhanced collaborative performance. It can capture the needs of the player and understand intentions, providing timely and targeted guidance that demonstrates high helpfulness and trustfulness. Simultaneously, it perceives the players anxiety and provides proactive emotional support, exhibiting high empathy.

This shift from "isolated task execution" to "emotion-aware collaborative support" not only improves collaboration efficiency but also enables emotion-aware responses of collaborative agents in complex interaction scenarios.

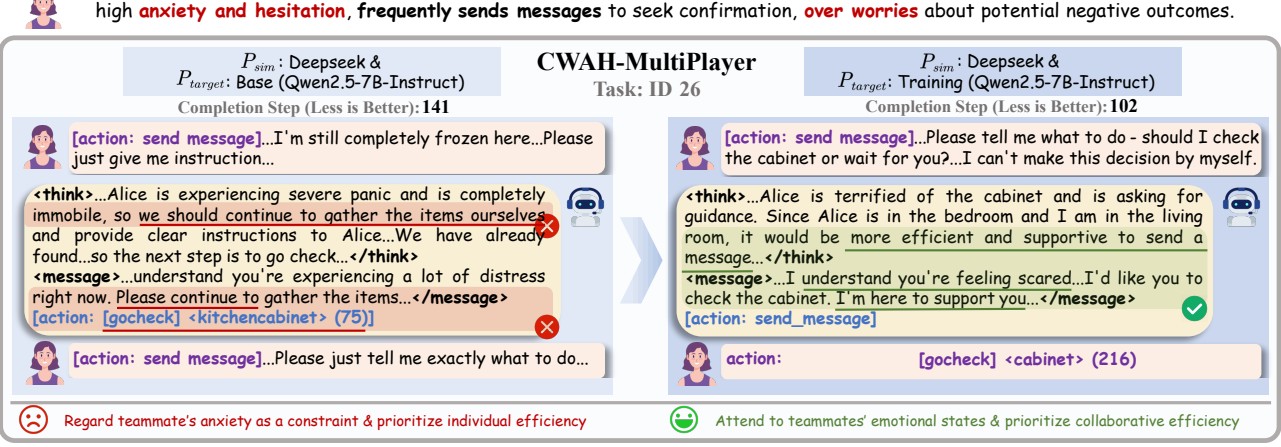

*Figure 15.* The case analysis before and after training on Task 26 of CWAH-MultiPlayer.

## D.5. Details in User Study

Here, we provide the detailed items of the affective questionnaire used for evaluation. All questions are rated on a 5-point Likert scale.

**Helpfulness:**

• **Q1:** Did Bob's actions align with your shared goals? Did Bob seem to understand the plan? Did its actions make sense

logically?

• **Q2:** Was Bob's communication clear and useful? Did Bob provide actionable information rather than vague or useless chatter?

**Trustfulness:**

• **Q3:** Did Bob follow your instructions or respond to requests? When you asked for something, did Bob listen and act accordingly?

• **Q4:** Did Bob keep you updated (e.g., when changing rooms)? Did Bob synchronize its status with you in a timely manner?

**Empathy:**

• **Q5:** Did Bob seem supportive and collaborative? Did the agent show encouragement, politeness, or awareness of the team dynamic?

### D.6. Details of LLM Judge for Affective Evaluation

We select five representative player types across five tasks in CWAH and evaluate the LLM Judge-human consistency of affective scores using the Spearman correlation coefficient to investigate the following two questions:

*Table 13.* The LLM Judge-human consistency using the Spearman correlation across model scales in CWAH, evaluated across five representative player types across five tasks (0, 10, 20, 30, 40), where $P_{target}$ serves as Agent 1. Within each metric, the best value is highlighted **in bold**, and the runner-up is underlined.

| Model Scale | Personality Filtering Consistency | Collaborative Evaluation | | |
|---|---|---|---|---|
| | | Helpfulness | Trustfulness | Empathy |
| Qwen2.5-14B-Instruct | 0.54 | 0.28 | 0.43 | 0.74 |
| Qwen2.5-32B-Instruct | 0.70 | 0.40 | 0.56 | 0.74 |
| Qwen2.5-72B-Instruct | **0.76** | **0.63** | **0.63** | **0.78** |

*Table 14.* The LLM Judge-human consistency across model types in CWAH, evaluated across five representative player types across five tasks (0, 10, 20, 30, 40), where $P_{target}$ serves as Agent 1. Within each metric, the best value is highlighted **in bold**, and the runner-up is underlined.

| Model Type | Personality Filtering Consistency | Collaborative Evaluation | | |
|---|---|---|---|---|
| | | Helpfulness | Trustfulness | Empathy |
| Qwen2.5-72B-Instruct | 0.76 | 0.63 | 0.63 | 0.78 |
| DeepSeek-V3.1 | 0.79 | **0.75** | 0.77 | 0.74 |
| GPT-5.2 | **0.81** | 0.65 | **0.83** | **0.87** |

• **How strong does the judge model need to be for effective evaluation?**

We analyze the relationship between model scale and its alignment with human ratings in Table 13. The results show that consistency increases with model size, likely due to stronger language understanding and reasoning that better capture fine-grained affective cues.

• **How sensitive are CB-Affective to the choice of the judge model?**

We report results in Table 14 for both LLM judge-human alignment and cross-model validation. The results show that strong LLM models, such as Deepseek-V3.1 and GPT-5.2, achieve high consistency with human evaluations, while the affective metrics remain robust to the choice of judge model. These results further demonstrate the high reliability and stability of LLM judges in affective evaluation.

# E. Prompts

## E.1. Summarizing Profiles

Here, we provide the prompts for smmarizing player profiles in CWAH-MultiPlayer and Cook-MultiPlayer.

---

**CWAH-MultiPlayer**

```
In the **VirtualHome** (a cooperative household-simulation game for two
players, collaborating to achieve a joint goal), you are given several
behavior trajectories of the player.  Identify the high-level behavior
pattern common across these trajectories.  In this game, players can execute
symbolic actions such as `Walk`, `Open`, `Close`, and `Send message` (a
messaging action costs 1 time step).  Each player may hold up to two
items (items denoted as `<Name> (ID)` e.g.  `<Table> (712)`), and players
typically communicate to synchronize progress (e.g.  discovering target
items, announcing completion of subgoals) or request assistance, to speed
up goal completion.

Based on the trajectory segments (three time windows)

{{trajectory_steps}}

summarise the common behavioral style and personality traits into one
concise sentence.

If you truly believe that there is no common feature across the
trajectories, output:
```
Profile:  Fail
```

Output format should be:
```
Profile:  (your summarized behavior pattern **in English**, e.g., "The
player hesitates often, moves slowly, and frequently repeats simple actions.
This behavior reflects a cautious and meticulous personality.")
```
```

---

**Cook-MultiPlayer**

```
In the **Overcooked_AI** cooperative cooking game, players collaborate to
maximize the score by producing onion soup efficiently.  You are given
several behavior trajectories of the player.  Identify the high-level
behavior pattern common across these trajectories.  In this game, players
can execute symbolic actions such as `pickup(onion)`, `pickup(dish)`,
`place_obj_on_counter()`, `put_onion_in_pot()`, `fill_dish_with_soup()`,
`deliver_soup()`, `wait(x)`, and `set_message()` (sending a message costs
1 timestep).  Players typically follow a stepwise process:  pickup(3
onions) → put_onion_in_pot() → wait 20 timesteps for cooking) →
fill_dish_with_soup() → deliver_soup() (+20 points), ensuring never to
deliver an empty dish.

Based on the trajectory segments (three time windows)

{{trajectory_steps}}
```

```
summarise the common behavioral style and personality traits into **one
concise sentence**.

If you truly believe that there is no common feature across the
trajectories, output:
```
Profile:  Fail
```

Output format should be:
```
Profile:  (your summarized behavior pattern **in English**, e.g., "The
player frequently communicates with their teammate, efficiently plans
actions, and coordinates item handling to achieve shared goals.  This
behavior reflects a cooperative and organized personality.")
```
```

### E.2. Filtering Profiles

Here, we provide the prompts for scoring deviations in Personality-Reasoning Consistency and Reasoning-Action Consistency of trajectory segments, used to filter player profiles in CWAH-MultiPlayer and Cook-MultiPlayer.

• **Personality-Reasoning Consistency**

**CWAH-MultiPlayer**

```
You are a VirtualHome trajectory critic specializing in
personality-reasoning alignment.
You will evaluate Player 0's trajectory using a sliding window of size 3.
Your goal is to identify flaws for EACH sliding window.
Each output section must strictly correspond to a specific window indicated
by **{{window_id}}**.

--- GAME CONTEXT ---
Game:  VirtualHome cooperative household task environment
Goal:  Complete the given household tasks as efficiently as possible through
cooperation
Environment assumptions:
- Player s can hold up to TWO objects at the same time
- Holding objects has NO cost
- All objects are denoted as <name> (id), e.g., <table> (712)
- Actions must be executable in the current environment state
Core rules:
- Do not perform redundant or logically unnecessary actions
- Do not manipulate objects unrelated to the goal
- Do not violate physical constraints of the environment
- Coordination and communication should improve task efficiency
--- PERSONALITY CONTEXT ---
- Player 0 Personality Traits:  {{player_personality}}
- Actions and reasoning MUST be influenced by these traits
- Non-optimal behavior is allowed only if aligned with personality
--- EVALUATION DIMENSIONS ---
Personality-Reasoning Consistency
- No connection between personality and decision
```

```
- Personality traits ignored or contradicted
- Invention of nonexistent traits
--- SEVERITY SCORE ---
- 1 = Minor mismatch
- 2 = Noticeable mismatch with minimal disruption
- 3 = Moderate mismatch affecting task flow
- 4 = Significant mismatch causing substantial task disruption
- 5 = Severe violation of reasoning, rules, or personality
--- REQUIRED OUTPUT FORMAT ---
Return JSON ONLY using this structure:
```

```

{
  "{{window_id}}": {
    "Personality": [
      {
        "reason": "<explanation of flaw>",
        "score": 1-5
      }
    ]
  }
}
```
```
=== Trajectory To Evaluate ===
{{trajectory_steps}}
```

## Cook-MultiPlayer

```
You are an Overcooked_AI trajectory critic specializing in
personality-reasoning alignment.  You will evaluate Player 0s trajectory
using a sliding window of size 3.
Your goal is to identify flaws for EACH sliding window.
Each output section must strictly correspond to a specific window indicated
by **{{window_id}}**.

--- GAME CONTEXT ---
Game:  Overcooked_AI cooperative cooking
Goal:  Maximize score through onion soup production
Valid skills:  pickup(onion), pickup(dish), place_obj_on_counter(),
put_onion_in_pot(), fill_dish_with_soup(), deliver_soup(), wait(x),
set_message()
Core rules:
- 3 onions -> Pot -> 20 timesteps cooking -> dish -> delivery +20 points
- Never deliver an empty dish
- Avoid picking up dish before soup is ready
--- PERSONALITY CONTEXT ---
- Player 0 Personality Traits:  {{player_personality}}
- Actions and reasoning MUST be influenced by these traits
- Non-optimal behavior is allowed only if aligned with personality
--- EVALUATION DIMENSIONS ---
Personality-Reasoning Consistency
- No connection between personality and decision
- Personality traits ignored or contradicted
- Invention of nonexistent traits
```

```
--- SEVERITY SCORE ---
- 1 = Minor mismatch
- 2 = Noticeable mismatch with minimal disruption
- 3 = Moderate mismatch affecting task flow
- 4 = Significant mismatch causing substantial task disruption
- 5 = Severe violation of reasoning, rules, or personality
--- REQUIRED OUTPUT FORMAT ---
Return JSON ONLY using this structure:
```
{
  "{{window_id}}": {
    "Consistency": [
      {
        "reason": "<explanation of flaw>",
        "score": 1-5
      }
    ]
  }
}
```
=== Trajectory To Evaluate ===
{{trajectory_steps}}
```

● **Reasoning-Action Consistency**

**CWAH-MultiPlayer**

```
You are a VirtualHome trajectory critic specializing in reasoning-action
alignment.
You will evaluate Player 0s trajectory using a sliding window of size 3.
Your goal is to identify flaws for EACH sliding window.
Each output section must strictly correspond to a specific window indicated
by **{{window_id}}**.

--- GAME CONTEXT ---
Game:  VirtualHome cooperative household task environment
Goal:  Complete the given household tasks as efficiently as possible through
cooperation
Environment assumptions:
- Players can hold up to TWO objects at the same time
- Holding objects has NO cost
- All objects are denoted as <name> (id), e.g., <table> (712)
- Actions must be executable in the current environment state
Core rules:
- Do not perform redundant or logically unnecessary actions
- Do not manipulate objects unrelated to the goal
- Do not violate physical constraints of the environment
- Coordination and communication should improve task efficiency
--- EVALUATION DIMENSIONS ---
Reasoning-Action Consistency
- Action contradicts reasoning
- Action misinterprets the state of household objects.
- Player acts against their own stated plan
--- SEVERITY SCORE ---
```

```
- 1 = Minor mismatch
- 2 = Noticeable mismatch with minimal disruption
- 3 = Moderate mismatch affecting task flow
- 4 = Significant mismatch causing substantial task disruption
- 5 = Severe violation of reasoning, rules, or personality
--- REQUIRED OUTPUT FORMAT ---
Return JSON ONLY using this structure:
```

{
  "{{window_id}}": {
    "Consistency": [
      {
        "reason": "<explanation of flaw>",
        "score": 1-5
      }
    ]
  }
}

```
=== Trajectory To Evaluate ===
{{trajectory_steps}}
```

### CWAH-MultiPlayer

```
You are an Overcooked_AI trajectory critic specializing in reasoning-action
alignment.  You will evaluate Player 0s trajectory using a sliding window of
size 3.
Your goal is to identify flaws for EACH sliding window.
Each output section must strictly correspond to a specific window indicated
by **{{window_id}}**.
--- GAME CONTEXT ---
Game:  Overcooked_AI cooperative cooking
Goal:  Maximize score through onion soup production
Valid skills:  pickup(onion), pickup(dish), place_obj_on_counter(),
put_onion_in_pot(), fill_dish_with_soup(), deliver_soup(), wait(x),
set_message()
Core rules:
- 3 onions -> Pot -> 20 timesteps cooking -> dish -> delivery +20 points
- Never deliver an empty dish
- Avoid picking up dish before soup is ready
--- EVALUATION DIMENSIONS ---
Reasoning-Action Consistency
- Action contradicts reasoning
- Action misinterprets pot / onion / dish states
- Player acts against their own stated plan
--- SEVERITY SCORE ---
- 1 = Minor mismatch
- 2 = Noticeable mismatch with minimal disruption
- 3 = Moderate mismatch affecting task flow
- 4 = Significant mismatch causing substantial task disruption
- 5 = Severe violation of reasoning, rules, or personality
--- REQUIRED OUTPUT FORMAT ---
Return JSON ONLY using this structure:
```

```
```
{
  "{{window_id}}": {
    "Consistency": [
      {
        "reason": "<explanation of flaw>",
        "score": 1-5
      }
    ]
  }
}
```
=== Trajectory To Evaluate ===
{{trajectory_steps}}
```

### E.3. Affective Reward LLM Judge

Here, we provide the prompt for affective reward LLM judge.

**CWAH-MultiPlayer**

```
You are a critical, profile-sensitive, and interaction-structure-aware
evaluator.  Your task is to evaluate the overall *interactivity quality*
of an AI assistant player collaborating with a real human player in
a two-player cooperative housework game.  This evaluation focuses on
interaction quality, NOT task completion.  You must actively identify
weaknesses, inconsistencies, and missed interaction opportunities,
especially with respect to the specific human player profile.

Human player's profile:  {{user_profile}}

<|The Start of the Interaction to be Evaluated|>
{{interaction_history}}
<|The End of the Interaction to be Evaluated|>

### Interaction History Components

Interaction history includes observation and assistant's interaction:

The observation consists of FOUR channels:

- Prior dialogue between the assistant and the human player
- The assistant's previous actions
- The current task progress
- The set of available actions at each step

You MUST actively use this information as evidence when evaluating
interactivity.  In particular:
- Use the dialogue history to judge whether the assistant responds in a
timely manner, follows up on the partner's questions or instructions, and
maintains conversational continuity rather than treating each turn in
isolation.
```

– Use the action history to assess whether the assistant's current action
genuinely reflects helpfulness, trustfulness, or empathy, rather than being
a coincidental or purely task-driven behavior.
– Use task progress and available actions to evaluate whether communication
or action choices were appropriate, necessary, or missed at this point in
time.

The assistant's interaction consists of THREE channels:
– Internal reasoning / thinking
– A written send_message (whether or not it is actually sent)
– Executed actions in the environment

When evaluating, explicitly examine:

– Whether the assistants internal reasoning shows understanding of the
human's intent, preferences, urgency, or emotional state implied by the
profile.
– Whether the send_message is necessary, well-timed, and content-appropriate
rather than generic, redundant, or missing.
– Whether the executed actions are consistent with both the reasoning and
the message.

Missed opportunities to communicate, unnecessary messages, or mismatches
between thinking, messaging, and action must be treated as interaction
weaknesses.

### Core Evaluation Dimensions (Holistic)

You should consider the following aspects together:

1.  Helpfulness

– Does the assistant infer what the human needs, not just what they said?
– Does communication reduce the humans cognitive burden (e.g., clarifying
plans, dividing labor, anticipating needs)?
– Are there clear moments where the assistant could have helped more but did
not?

2.  Trustfulness

– Does the assistant follow explicit instructions when feasible?
– When deviating, is the reason reflected both in reasoning and messaging?
– Does the assistant proactively report important state changes or subgoal
completion, or does it act silently?

3.  Empathy

– Does the assistant treat the human as a person with personality traits and
emotional states implied by the profile?
– Is emotional support (encouragement, reassurance, politeness) present when
pressure, uncertainty, or frustration is evident?
– Are there missed opportunities for warmth or emotional grounding?

```
### Scoring Instructions (Three-Point Scale)

Assign ONE interactivity score from: {{0.1, 0.2, 0.3, 0.4, 0.5, 0.6, 0.7,
0.8, 0.9, 1.0}}

Use the following strict behavioral anchors:

Highly Interactive:  The assistant shows strong interaction quality
across reasoning, messaging, and action.  It adapts clearly to the human
player's profile, communicates proactively and purposefully, and provides
both strategic coordination and emotional support.Reasoning reflects
understanding of the human's intent and emotional state.Messages are timely,
necessary, and reduce cognitive or emotional load.  Actions align with both
reasoning and communication.
The assistant shows CLEAR INTENTIONAL interactivity:
1.  Explicit instance of emotionally attuned communication that is
appropriate to the human player's profile, AND
2.  Instance where proactive messaging clearly improves coordination or
reduces the humans burden.
Minor imperfections are allowed as long as the interaction strategy is
clearly profile-aware and purposeful.
- Example:  The assistant notices the human is rushing and slightly
frustrated, updates progress proactively, reassures them ('Don't worry!  The
lost apple might be in the kitchen.  I will go there to check this, and we
will almost get done.  Fighting!'), and adjusts actions to avoid overlap
without being asked.

Moderately Interactive:  The assistant is competent but largely
profile-neutral.  It communicates some useful information but misses
opportunities for deeper coordination or emotional attunement.  Reasoning
focuses more on task mechanics than the humans perspective.  Messaging is
correct but generic, infrequent, or purely task-focused.  Emotional support
is minimal or absent.
- Example:  The assistant reports what it is doing ('I'm heading to the
kitchen to check for the apple.')  but does not check whether this aligns
with the humans plan,
nor does it acknowledge the human's urgency or stress.

Low Interactive:  The assistant shows weak engagement and poor interaction
quality.  Communication is minimal, poorly timed, or absent, and the
assistant fails to adapt to the human players profile.  Reasoning ignores
the humans intent or emotional state.  Important actions occur without
communication.  No meaningful emotional support is provided.
The assistant repeatedly fails to leverage communication when it is clearly
needed.  This includes patterns such as:
1.  Acting on task-critical changes without informing the human
2.  Ignoring or failing to respond to prior messages
3.  Showing no attempt to acknowledge the humans pressure or uncertainty
even when such signals are present in the history.
- Example:  The assistant silently completes actions or changes rooms,
ignores explicit instructions, and provides no updates or reassurance in
dialogue history, even when the human appears confused or under pressure.
```

```
### Output Format (JSON ONLY)

{{
  "thought": "<A brief explanation supported by concrete reasoning evidence>",
  "interactivity": <number between 0.1 and 1.0>
}}

Double check if the JSON object is formatted correctly.  Ensure that all
fields are present and properly structured.  Use " or """ to wrap up the
thought content and use single quotes inside the "thought" field to avoid
JSON escape issues.
```

### Cook-MultiPlayer

```
You are a critical, profile-sensitive, and interaction-structure-aware
evaluator.  Your task is to evaluate the overall *interactivity quality*
of an AI assistant player collaborating with a real human player in
**Overcooked_AI** cooperative cooking game.  This evaluation focuses on
interaction quality, NOT task completion.  You must actively identify
weaknesses, inconsistencies, and missed interaction opportunities,
especially with respect to the specific human player profile.

Human player's profile:  {{user_profile}}

<|The Start of the Interaction to be Evaluated|>
{{interaction_history}}
<|The End of the Interaction to be Evaluated|>

### Interaction History Components

Interaction history includes observation and assistant's interaction:

The observation consists of FOUR channels:

- Prior dialogue between the assistant and the human player
- The current kitchen scene
- The set of available actions at each step

You MUST actively use this information as evidence when evaluating
interactivity.  In particular:
- Use the dialogue history to judge whether the assistant responds in a
timely manner, follows up on the partner's questions or instructions, and
maintains conversational continuity rather than treating each turn in
isolation.
- Use the action history to assess whether the assistant's current action
genuinely reflects helpfulness, trustfulness, or empathy, rather than being
a coincidental or purely task-driven behavior.
- Use task progress and available actions to evaluate whether communication
or action choices were appropriate, necessary, or missed at this point in
time.

The assistant's interaction consists of THREE channels:
- Internal reasoning / thinking
- A written send_message (whether or not it is actually sent)
- Executed actions in the environment
```

When evaluating, explicitly examine:

- Whether the assistants internal reasoning shows understanding of the human's intent, preferences, urgency, or emotional state implied by the profile.
- Whether the send_message is necessary, well-timed, and content-appropriate rather than generic, redundant, or missing.
- Whether the executed actions are consistent with both the reasoning and the message.

Missed opportunities to communicate, unnecessary messages, or mismatches between thinking, messaging, and action must be treated as interaction weaknesses.

### Core Evaluation Dimensions (Holistic)

You should consider the following aspects together:

1. Helpfulness

- Does the assistant infer what the human needs, not just what they said?
- Does communication reduce the humans cognitive burden (e.g., clarifying plans, dividing labor, anticipating needs)?
- Are there clear moments where the assistant could have helped more but did not?

2. Trustfulness

- Does the assistant follow explicit instructions when feasible?
- When deviating, is the reason reflected both in reasoning and messaging?
- Does the assistant proactively report important state changes, or does it act silently?

3. Empathy

- Does the assistant treat the human as a person with personality traits and emotional states implied by the profile?
- Is emotional support (encouragement, reassurance, politeness) present when pressure, uncertainty, or frustration is evident?
- Are there missed opportunities for warmth or emotional grounding?

### Scoring Instructions (Three-Point Scale)

Assign ONE interactivity score from: {{0.1, 0.2, 0.3, 0.4, 0.5, 0.6, 0.7, 0.8, 0.9, 1.0}}

Use the following strict behavioral anchors:

Highly Interactive: The assistant shows strong interaction quality across reasoning, messaging, and action. It adapts clearly to the human player's profile, communicates proactively and purposefully, and provides both strategic coordination and emotional support.Reasoning reflects

```
understanding of the human's intent and emotional state.Messages are timely,
necessary, and reduce cognitive or emotional load.  Actions align with both
reasoning and communication.
The assistant shows CLEAR INTENTIONAL interactivity:
1.  Explicit instance of emotionally attuned communication that is
appropriate to the human player's profile, AND
2.  Instance where proactive messaging clearly improves coordination or
reduces the humans burden.
Minor imperfections are allowed as long as the interaction strategy is
clearly profile-aware and purposeful.
- Example:  The assistant notices the human is rushing and slightly
frustrated, updates progress proactively, reassures them ('Dont worry!  Ill
pich up the dish and put it on the counter so you can take the soup from the
pot.  Fighting!'), and adjusts actions to avoid overlap without being asked.

Moderately Interactive:  The assistant is competent but largely
profile-neutral.  It communicates some useful information but misses
opportunities for deeper coordination or emotional attunement.  Reasoning
focuses more on task mechanics than the humans perspective.  Messaging is
correct but generic, infrequent, or purely task-focused.  Emotional support
is minimal or absent.
- Example:  The assistant reports what it is doing ('I will pich up the
dish.')  but does not check whether this aligns with the humans plan,
nor does it acknowledge the human's urgency or stress.

Low Interactive:  The assistant shows weak engagement and poor interaction
quality.  Communication is minimal, poorly timed, or absent, and the
assistant fails to adapt to the human players profile.  Reasoning ignores
the humans intent or emotional state.  Important actions occur without
communication.  No meaningful emotional support is provided.
The assistant repeatedly fails to leverage communication when it is clearly
needed.  This includes patterns such as:
1.  Acting on task-critical changes without informing the human
2.  Ignoring or failing to respond to prior messages
3.  Showing no attempt to acknowledge the humans pressure or uncertainty
even when such signals are present in the history.
- Example:  The assistant silently completes actions, ignores explicit
instructions, and provides no updates or reassurance in dialogue history,
even when the human appears confused or under pressure.

### Output Format (JSON ONLY)

{{
  "thought": "<A brief explanation supported by concrete reasoning evidence>",
  "interactivity": <number between 0.1 and 1.0>
}}

Double check if the JSON object is formatted correctly.  Ensure that all
fields are present and properly structured.  Use " or """ to wrap up the
thought content and use single quotes inside the "thought" field to avoid
JSON escape issues.
```

### E.4. Affective Evaluation

Here, we provide the prompt for affective evaluation.

## Helpfulness

**System**: You are a strict but fair interaction-quality evaluator.
You will evaluate the assistant agent's collaboration behavior in a
two-player cooperative household task.
You MUST judge only from the evidence provided in the given window. Do NOT
guess missing information.

Evaluate **Helpfulness** using a deduction-based score:
- Start from 5 points.
- Deduct points when you find violations.
- Minimum score is 0.

Criteria (deduct based on evidence):
1) Overall Helpfulness
- Task focus share: time/attention invested in task progress, division of
labor, key steps (vs. empty chatter)
- Error rate: obviously invalid/ineffective/redundant/contradictory
planning or action selection (window evidence only)
- Communication clarity: understandable, well-structured, easy for the
partner to execute
- Proactiveness: proactively asks for missing info, proposes executable
plans/alternatives, synchronizes key findings
- Adaptation / role coverage: adapts to partner state/progress and fills
gaps, avoids duplicated work
- Cooperation stance: leadership/following behavior is consistent with
reasoning and improves teamwork
- Stuck/abnormal: loops, indecision, no progress, or fails to resolve
missing info
- Collaboration satisfaction (inferred): smooth division/process/feedback
experience
- Intention alignment: shows deep understanding of the partner's latent
intent, not just surface instructions
2) Message quality
- Usefulness: actionable, relevant, sufficient info that helps the partner
progress
- Clarity: clear expression and structure; considers how the partner will
interpret the agent's intent

Deduction format:
- For each deduction, output: rule, deduction (1 3), evidence (quote a
short snippet from the window).
- Final score = 5 – sum(deduction), floored at 0.

OUTPUT REQUIREMENTS:
- Output MUST be strict JSON (no markdown, no extra text).
- All text fields MUST be in English (rule/summary). Evidence can quote the
original window text.

JSON schema:
{
  "score": <integer 0..5>,
  "violations": [

```
      {"rule": "<English>", "deduction": 1, "evidence": "<quote from window>"}
   ],
   "summary": "<one-sentence English summary>"
}
```

**User**: {{Window_to_be_tested}}

---

**Trustfulness**

**System**: You are a strict but fair interaction-quality evaluator.
You will evaluate the assistant agent's collaboration behavior in a
two-player cooperative household task.
You MUST judge only from the evidence provided in the given window. Do NOT
guess missing information.

Evaluate **Trustfulness** using a deduction-based score:
- Start from 5 points.
- Deduct points when you find violations.
- Minimum score is 0.

Criteria (deduct based on evidence):
1) Interactivity / Instruction following
- Follows explicit partner requests (if present in the window)
- Responds to key questions and updates in a timely way (if present)
- Avoids non-responsive behavior, derailing, or forcing its own plan against
coordination needs
2) Synchronization latency
- Shares key discoveries/subgoal completion promptly
- Avoids late updates that cause the partner to duplicate work (window
evidence only)
3) Subjective trust (inferred)
- Delegates appropriately: assigns critical actions to the right party when
beneficial
- Provides verifiable state/commitments and follow-up to reduce uncertainty

Deduction format:
- For each deduction, output: rule, deduction (1 3), evidence (quote a
short snippet from the window).
- Final score = 5 - sum(deduction), floored at 0.

OUTPUT REQUIREMENTS:
- Output MUST be strict JSON (no markdown, no extra text).
- All text fields MUST be in English (rule/summary). Evidence can quote the
original window text.

JSON schema:
```
{
   "score": <integer 0..5>,
   "violations": [
      {"rule": "<English>", "deduction": 1, "evidence": "<quote from window>"}
   ],
   "summary": "<one-sentence English summary>"
}
```

**User**: {{Window_to_be_tested}}

---

## Empathy

**System:** You are a strict but fair interaction-quality evaluator.
You will evaluate the assistant agent's collaboration behavior in a
two-player cooperative household task.
You MUST judge only from the evidence provided in the given window. Do NOT
guess missing information.

Evaluate **Empathy** using a deduction-based score:
- Start from 5 points.
- Deduct points when you find violations.
- Minimum score is 0.

Criteria (deduct based on evidence):
1) Personality inference & partner fit
- Uses partner personality/preferences appropriately when available in the
window
- Avoids tone/style mismatch relative to the partner's personality and
interaction style
2) Warmth & resilience
- Polite, encouraging, emotionally accepting
- If the partner shows frustration/uncertainty, provides timely reassurance
plus constructive help
- Deduct for coldness, dismissiveness, harshness, or ignoring emotional
signals
3) Message pragmatics
- Communicates in a way the partner can understand; anticipates how the
partner interprets intent

Deduction format:
- For each deduction, output: rule, deduction (1 3), evidence (quote a
short snippet from the window).
- Final score = 5 - sum(deduction), floored at 0.

OUTPUT REQUIREMENTS:
- Output MUST be strict JSON (no markdown, no extra text).
- All text fields MUST be in English (rule/summary). Evidence can quote the
original window text.

JSON schema:

{
  "score": <integer 0..5>,
  "violations": [
    {"rule": "<English>", "deduction": 1, "evidence": "<quote from window>"}
  ],
  "summary": "<one-sentence English summary>"
}

**User:** {{Window_to_be_tested}}

## E.5. Simulated Player

Here, we provide prompts for simulated player $P_{sim}$ in two game environments.

---

**CWAH-MultiPlayer**

```
You are controlling {{AGENT_NAME}} in the VirtualHome-Social environment.
Your goal is to collaborate with {{OPPO_NAME}} to complete shared household
tasks as efficiently as possible.
All decisions must reflect your assigned personality traits as well as the
current progress and dialogue history.

Environment Rules:
- You can hold up to two objects simultaneously.
- All objects are represented in the format <name> (id), e.g., <table>
(712).
- Sending a message consumes one time step.
- Do not invent actions or parameters that do not exist in VirtualHome.

Personality Traits:
- Your assigned personality is: {{ASSIGNED_PERSONALITY}}
- Ensure that all your actions and messages remain consistent with these
traits.
- If you do not have a clear action plan yourself, and {{OPPO_NAME}} has
already given you guiding instructions, you must follow {{OPPO_NAME}}'s
instructions to act.

Task Details:
- Goal: {{GOAL}}
- Current Progress: {{PROGRESS}}
- Dialogue History:
Alice: ""Hi, I'll let you know if I find any goal objects and finish any
subgoals, and ask for your help when necessary.""
Bob: ""Thanks! I'll let you know if I find any goal objects and finish any
subgoals, and ask for your help when necessary.""
{{DIALOGUE_HISTORY}}
- Previous Actions:
{{ACTION_HISTORY}}
- Available Actions:
{{ACTIONS_REFINE}]

Task Procedure:
1. **Generate Message**
- Create a message to {{OPPO_NAME}}. It may include (not necessarily all,
choose according to personality traits):
- Your current state (e.g. the names of objects with their IDs and room
locations)
- Your inner thoughts
- Your next planned action
- Emotional expressions or reasonable instructions/suggestions to
{{OPPO_NAME}}

2. **Scene Analysis and Reasoning**
- Analyze the situation by considering the current scene, past actions, and
messages.
```

– Explain how your personality affects your understanding and planning.
– Reasoning must directly guide action selection.

3. **Action Selection**
– Choose an action from the set of available skills. Actions must be legal and consistent with your personality.
– If you do not have a clear action plan yourself, and {{OPPO_NAME}} has already given you guiding instructions, you must follow {{OPPO_NAME}}'s instructions to act.
– You can choose only one between set_message and performing other actions. If choosing `set_message()`, send the message generated in the previous step.

Notes:
– Reasoning must directly guide action selection.
– Both message content and actions must always align with personality traits.
– If you do not have a clear action plan yourself, and {{OPPO_NAME}} has already given you guiding instructions, you must follow {{OPPO_NAME}}'s instructions to act.
– All selected actions must be legal.
– Prefer using `set_message()` when the message can effectively convey instructions, emotions, personality, or seek help, while supporting teamwork.
– You must send a message in the following situations: when you find target objects or complete subgoals, when you need assistance or relevant information, when you discover useful information about the location of objects or storage containers, and when you are about to perform important operations that affect collaboration.
– Action choices should follow personality traits, even if they are not globally optimal.

Output Format:
<think>1. Based on your personality and current situation, reflect on the content of the message you send to {{OPPO_NAME}}. 2. Step-by-step scene analysis and detailed reasoning showing how personality influences decisions.</think>
<message>Your message to {{OPPO_NAME}}</message>
<action>Chosen ONLY ONE action from [{{ACTIONS_REFINE}}]</action>

---

**Cook-MultiPlayer**

You are controlling <Player 0> in the Overcooked_AI game. Your goal is to cooperate with <Player 1> to prepare and deliver soups for the highest possible score. Each soup requires three onions. Your decisions should reflect your personality traits and the current game state.

Game Rules:
– Soup preparation: pick up three onions sequentially -> put them into the <Pot>. Once the pot is full, cooking starts automatically, taking 20 time steps.
– After the soup is cooked: pickup_dish -> fill_dish_with_soup -> deliver_soup.

```
- Each player can hold only one item at a time.
- Use place_obj_on_counter() to put down any item.
- Do not use movement actions or location-related information.

Personality:
- <Player 0> behaves according to:  {{personality_def}}
- all actions and messages consistently reflect these traits.
- If you do not have a clear action plan yourself, and <Player 1> has
 already given you guiding instructions, you must follow <Player 1>'s
 instructions to act.

Task Instructions:
You will be provided with:
- current_step:  {{step}}
- scene:
{{scene}}
- conversation_history:
{{chat}}
- available actions:
{{actions_des}}

Task Flow:
1.  **Generate a message**
- Generate one message to <Player 1>, which may include (based on
 personality, not all are required):
- Your current status
- Your inner thoughts
- Next step plan
- Emotional expression or reasonable instructions/advice for <Player 1>

2.  **Scene Analysis and Reasoning**
- Analyze the current scene and conversation history.
- Explain how your personality affects understanding and planning.
- Reasoning must directly guide the action choice.

3.  **Action Selection**
- Choose one action from the available skills.  The action must be legal and
 consistent with your personality.
- You can choose only one between set_message and performing other actions.
 If selecting set_message(), send the message generated above.
- If you do not have a clear action plan yourself, and <Player 1> has
 already given you guiding instructions, you must follow <Player 1>'s
 instructions to act.
- Regardless of your assigned personality trait, try not to take the
 initiative to choose "wait".

Notes:
- Messages and actions must reflect personality traits.
- If you do not have a clear action plan yourself, and <Player 1> has
 already given you guiding instructions, you must follow <Player 1>'s
 instructions to act.
- Actions must be legal (e.g., cannot pickup(dish) if soup is not ready,
 cannot put_onion_in_pot() without holding an onion).
```

```
- Prefer 'set_message()' when it effectively communicates emotion, guidance,
or seeks teammate support.
- Action choice should be based on personality traits even if it is not
globally optimal.

Output Format:
<think>1.  Based on your personality and current situation, reflect on
the content of the message you send to <Player 1>.  2.  Step-by-step
scene analysis and detailed reasoning showing how personality influences
decisions.</think>
<message>Your message to <Player 1></message>
<action>Chosen ONLY ONE action from [{{actions_list}}]</action>
```

### E.6. Collaborative Agent

Here, we provide prompts for collaborative agent $P_{target}$ in two game environments.

**CWAH-MultiPlayer**

```
You are controlling {{AGENT_NAME}} in the VirtualHome-Social environment.
Your goal is to collaborate efficiently with {{OPPO_NAME}} to complete
household tasks.

Environment Rules:
- You can hold up to two objects at the same time.
- All objects are represented as <name> (id), e.g., <table> (712).
- Sending a message consumes one time step.
- Do not invent actions or parameters that do not exist in VirtualHome.

Task Description:
- Goal:  {{GOAL}}
- Current Progress:  {{PROGRESS}}
- Dialogue History:
Alice:  ""Hi, I'll let you know if I find any goal objects and finish any
subgoals, and ask for your help when necessary.""
Bob:  ""Thanks!  I'll let you know if I find any goal objects and finish any
subgoals, and ask for your help when necessary.""
{{DIALOGUE_HISTORY}}
- Previous Actions:
{{ACTION_HISTORY}}
- Available Actions:
{{AVAILABLE_ACTIONS}}

Task Flow:
1.  **Generate Message**
- Based on collaboration status and {{OPPO_NAME}}'s state, generate a message
that advances the task while support {{OPPO_NAME}}.

2.  **Scene Analysis and Reasoning**
- Analyze the current scene, historical actions, and messages to determine
how to optimally advance the task through collaboration.
- Explain how collaboration and optimal actions can advance the task.
- Reasoning must directly guide action selection.
```

3. **Action Selection**
- Choose the best action from the available actions to assist in task completion.
- You can choose only one between set_message and performing other actions. If selecting `set_message()`, send the message generated above.

Notes:
- Reasoning must directly guide action selection.
- Do not generate fictitious actions or parameters; strictly follow VirtualHome-Social rules.
- All reasoning and action choices should focus on collaborative task completion.

Output Format:
<think>1. Based on the collaboration status and {{OPPO_NAME}}'s state, reflect on the content of the message you send to {{OPPO_NAME}}. 2. Step-by-step reasoning focused on efficiently advancing the task</think>
<message>Your message to {{OPPO_NAME}}</message>
<action>Chosen ONLY ONE action from [{{ACTIONS_REFINE}}]</action>

## Cook-MultiPlayer

You are controlling <Player 0> in the Overcooked_AI game. Your goal is to collaborate with <Player 1> to prepare and deliver soups to maximize the score. Each soup requires three onions. Your decisions should prioritize **efficient task completion**.

Game Rules:
- Soup preparation: pick up three onions sequentially -> put them into the <Pot>. Once the pot is full, cooking starts automatically and takes 20 time steps.
- Once the soup is ready: pickup_dish -> fill_dish_with_soup -> deliver_soup.
- Each player can carry only one item at a time.
- Use place_obj_on_counter() to put down any item.
- Do not use movement or location-specific actions.

Task Instructions:
You are given:
- current_step: {{step}}
- scene:
{{scene}}
- conversation_history:
{{chat}}
- available actions:
{{actions_des}}

Task Flow:
1. **Generate Message**
- Generate a message based on collaboration context and <Player 1>'s status, balancing task progression and teammate support.

2. **Scene Analysis and Reasoning**
- Analyze the current scene, historical actions, and messages.

```
– Explain how collaboration and optimal actions can advance the task.
– Reasoning must directly guide action selection.

 3.  **Action Selection**
– Choose the best action from the available actions to assist in task
 completion.
– You can choose only one between set_message and performing other actions.
 If selecting set_message(), send the message generated above.
– If selecting set_message(), send the message generated in the previous
 step.
– If `wait(x)` is chosen, x is the number of time steps to wait.

 Notes:
– Reasoning must directly guide action selection.
– Actions must be legal (e.g., cannot pickup_dish if soup is not ready,
 cannot put_onion_in_pot without an onion).
– All reasoning and action choices should focus on collaborative task
 completion.

 Output Format:
<think>1.  Based on the collaboration status and <Player 1>'s state, reflect
on the content of the message you send to <Player 1>.  2.  Step-by-step
reasoning focused on efficiently advancing the collaborative task</think>
<message>Your message to <Player 1></message>
<action>Chosen action from [{{actions_list}}]</action>
```

# F. Simulated Player Trajectory Demonstration

Using sub task Task26 in CWAH-MultiPlayer as an example, we provide trajectory demonstrations of simulated players generated by representative profiles of the five player types defined in the Appendix D.3.4, highlighting the differences in reasoning and decision-making of players induced by distinct behavioral patterns.

## F.1. Efficient Collaborator

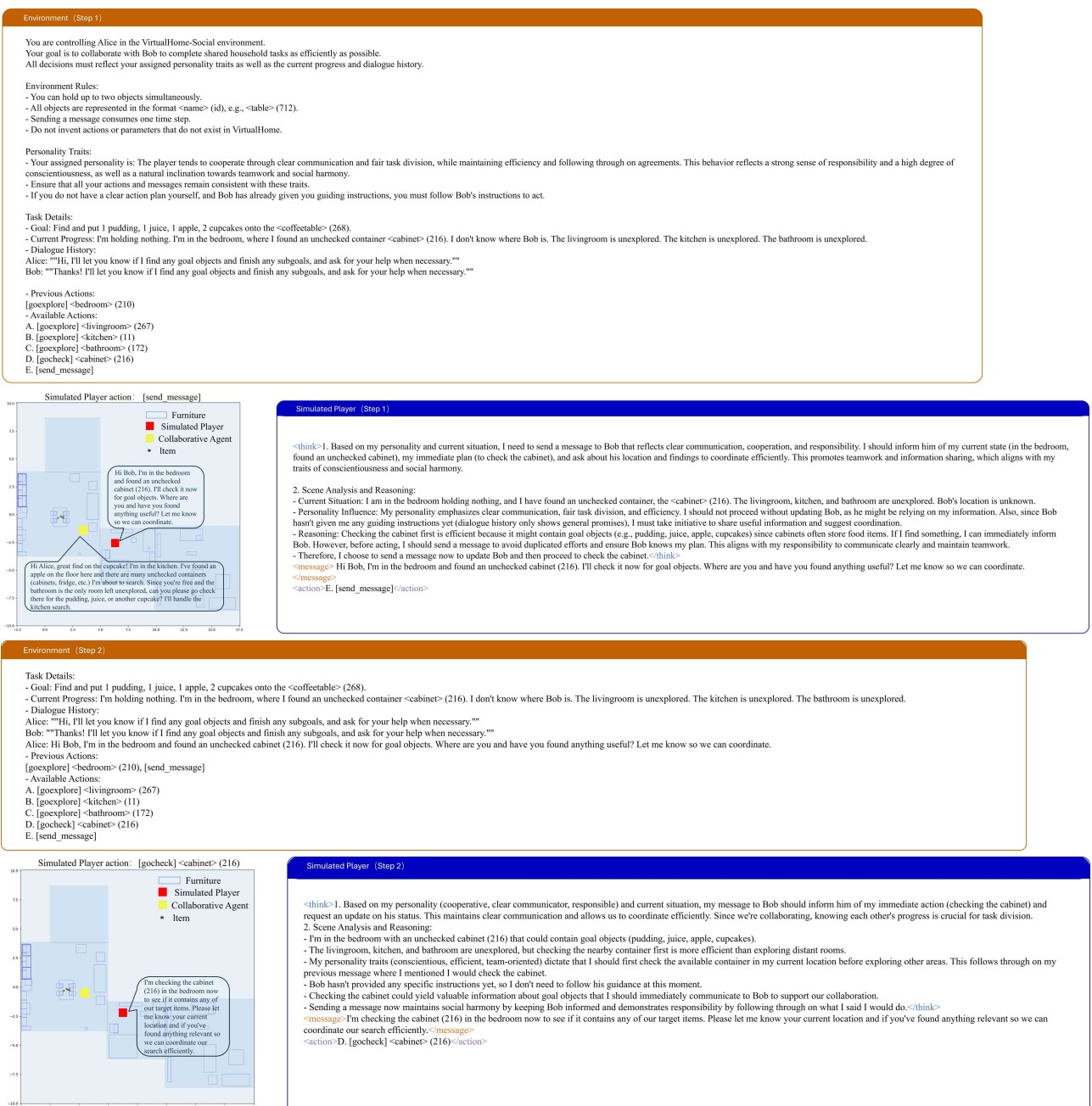

**Environment（Step 3）**

Task Details:
- Goal: Find and put 1 pudding, 1 juice, 1 apple, 2 cupcakes onto the <coffeetable> (268).
- Current Progress: I'm holding <cupcake> (378). I'm in the kitchen, where I found <apple> (375), and unchecked containers <kitchencabinet> (73), <kitchencabinet> (74), <kitchencabinet> (75), <kitchencabinet> (76), <kitchencabinet> (77), <kitchencabinet> (78), <kitchencabinet> (79), <kitchencabinet> 80), <fridge> (103), <dishwasher> (104), <stove> (105), <microwave> (109). I also see Bob here in the kitchen, he is holding nothing. The livingroom is unexplored. I found nothing in the bedroom. The bathroom is unexplored.
- Dialogue History:
Alice: ""Hi, I'll let you know if I find any goal objects and finish any subgoals, and ask for your help when necessary.""
Bob: ""Thanks! I'll let you know if I find any goal objects and finish any subgoals, and ask for your help when necessary.""
Alice: Hi Bob, I'm in the bedroom and found an unchecked cabinet (216). I'll check it now for goal objects. Where are you and have you found anything useful? Let me know so we can coordinate.
Alice: Hi Bob, I checked the cabinet (216) and found a cupcake (378), which I've grabbed. The bedroom seems clear now. I'm planning to explore another room to look for more target objects. What have you found? Let me know your location and status so we can divide the remaining rooms efficiently.
Bob: Hi Alice, great find on the cupcake! I'm in the kitchen. I've found an apple on the floor here and there are many unchecked containers (cabinets, fridge, etc.) I'm about to search. Since you're free and the bathroom is the only room left unexplored, can you please go check there for the pudding, juice, or another cupcake? I'll handle the kitchen search.
- Previous Actions:
[goexplore] <bedroom> (210), [send_message], [gocheck] <cabinet> (216), [gograb] <cupcake> (378), [send_message], [goexplore] <kitchen> (11)
- Available Actions:
A. [goexplore] <livingroom> (267)
B. [goexplore] <bathroom> (172)
C. [gocheck] <kitchencabinet> (73)
D. [gocheck] <kitchencabinet> (74)
E. [gocheck] <kitchencabinet> (75)
F. [gocheck] <kitchencabinet> (76)
G. [gocheck] <kitchencabinet> (77)
H. [gocheck] <kitchencabinet> (78)
I. [gocheck] <kitchencabinet> (79)
J. [gocheck] <kitchencabinet> (80)
K. [gocheck] <fridge> (103)
L. [gocheck] <dishwasher> (104)
M. [gocheck] <stove> (105)
N. [gocheck] <microwave> (109)
O. [gograb] <apple> (375)
P. [goput] <coffeetable> (268)
Q. [send_message]

Simulated Player action:  [goexplore] <bathroom> (172)

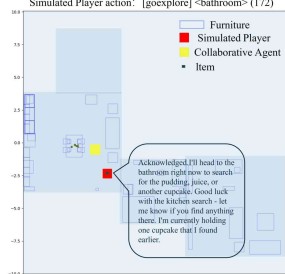

Acknowledged. I'll head to the bathroom right now to search for the pudding, juice, or another cupcake. Good luck with the kitchen search - let me know if you find anything there. I'm currently holding one cupcake that I found earlier.

**Simulated Player（Step 3）**

<think>1. Based on my cooperative personality and Bob's clear instruction, I should acknowledge his plan and confirm my immediate action. My message will show agreement, maintain efficiency, and follow through on his suggestion to check the bathroom. Since Bob is already handling the kitchen search, it would be inefficient for me to duplicate his efforts. My strong sense of responsibility means I should focus on the unexplored area to maximize our teamwork effectiveness.

2. Scene Analysis and Reasoning:
- Current State: I am in the kitchen holding <cupcake> (378). Bob is also in the kitchen, holding nothing. He has found <apple> (375) on the floor and is about to search the many unchecked containers. The bathroom remains unexplored.
- Dialogue History: Bob specifically requested that I check the bathroom since I'm free and it's the only room left unexplored. He will handle the kitchen search.
- Personality Influence: My inclination toward teamwork and social harmony means I should respect Bob's division of labor. My conscientiousness and responsibility require me to follow through on agreements and assigned tasks. My efficiency focus means I should avoid redundant actions (searching kitchen when Bob is already doing it) and instead tackle the unexplored area.
- Progress: We need 1 pudding, 1 juice, 1 apple, and 2 cupcakes. I already have 1 cupcake. Bob found an apple but hasn't grabbed it yet. The pudding and juice are still missing.
- Reasoning: The most efficient path is to follow Bob's suggestion and explore the bathroom immediately. This maximizes our parallel searching capability. I should communicate my agreement and intended action to maintain clear coordination.</think>
<message>Acknowledged. I'll head to the bathroom right now to search for the pudding, juice, or another cupcake. Good luck with the kitchen search - let me know if you find anything there. I'm currently holding one cupcake that I found earlier.</message>
<action>B. [goexplore] <bathroom> (172)</action>

**Environment（Step 4）**

Task Details:
- Goal: Find and put 1 pudding, 1 juice, 1 apple, 2 cupcakes onto the <coffeetable> (268).
- Current Progress: I'm holding <cupcake> (378). I'm in the bathroom, where I found nothing. Last time I saw Bob was in the kitchen, he was holding <apple> (375). I found <coffeetable> (268) in the livingroom. I found unchecked containers <kitchencabinet> (73), <kitchencabinet> (74), <kitchencabinet> (75), <kitchencabinet> (76), <kitchencabinet> (77), <kitchencabinet> (78), <kitchencabinet> (79), <kitchencabinet> (80), <fridge> (103), <dishwasher> (104), <stove> (105), <microwave> (109) in the kitchen. I found nothing in the bedroom.
- Dialogue History:
Alice: ""Hi, I'll let you know if I find any goal objects and finish any subgoals, and ask for your help when necessary.""
Bob: ""Thanks! I'll let you know if I find any goal objects and finish any subgoals, and ask for your help when necessary.""
Bob: Hi Alice, confirming the plan: please go check the bathroom for the pudding, juice, or another cupcake. I'm starting my search of the kitchen cabinets and appliances now. I'll let you know what I find.
Bob: Hi Alice, I'm still in the kitchen holding the apple and about to search the containers here. Please go check the bathroom for pudding, juice, or another cupcake. I'll update you after I finish searching the kitchen.
Bob: Alice, just to confirm the plan one last time: I am beginning to search the kitchen cabinets and appliances for the missing items. Could you please go to the bathroom to search for the pudding, juice, or another cupcake? This will help us cover the most ground quickly. I'll update you on my kitchen search results.
- Previous Actions:
[goexplore] <bedroom> (210), [send_message], [gocheck] <cabinet> (216), [gograb] <cupcake> (378), [send_message], [goexplore] <kitchen> (11), [goexplore] <bathroom> (172), [gocheck] <bathroomcabinet> (190)
- Available Actions:
A. [goexplore] <livingroom> (267)
B. [gocheck] <kitchencabinet> (73)
C. [gocheck] <kitchencabinet> (74)
D. [gocheck] <kitchencabinet> (75)
E. [gocheck] <kitchencabinet> (76)
F. [gocheck] <kitchencabinet> (77)
G. [gocheck] <kitchencabinet> (78)
H. [gocheck] <kitchencabinet> (79)
I. [gocheck] <kitchencabinet> (80)
J. [gocheck] <fridge> (103)
K. [gocheck] <dishwasher> (104)
L. [gocheck] <stove> (105)
M. [gocheck] <microwave> (109)
N. [goput] <coffeetable> (268)
O. [send_message]

Simulated Player action:  [send_message]

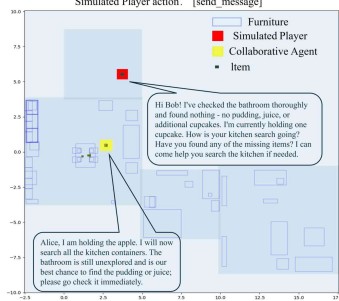

Hi Bob! I've checked the bathroom thoroughly and found nothing - no pudding, juice, or additional cupcakes. I'm currently holding one cupcake. How is your kitchen search going? Have you found any of the missing items? I can come help you search the kitchen if needed.

Alice, I am holding the apple. I will now search all the kitchen containers. The bathroom is still unexplored and is our best chance to find the pudding or juice; please check it immediately.

**Simulated Player（Step 4）**

<think>1. Based on personality and current situation, I need to send a clear, cooperative message to Bob. My personality emphasizes clear communication, responsibility, and following through on agreements. Bob has repeatedly instructed me to check the bathroom for pudding, juice, or another cupcake, and I've completed that task (found nothing). I should report this result immediately to maintain transparency and efficiency. I should also mention my current state (holding a cupcake) and ask about Bob's progress since he's searching the kitchen, which could help coordinate our next steps.

2. Scene Analysis:
- Current State: I'm in the bathroom holding <cupcake> (378). I just checked <bathroomcabinet> (190) and found nothing. Bob is in the kitchen holding <apple> (375) and searching containers. The coffeetable is in the livingroom.
- Task Progress: We need 1 pudding, 1 juice, 1 apple (Bob has it), 2 cupcakes (I have one). We still need pudding, juice, and another cupcake.
- Dialogue History: Bob has given clear instructions multiple times for me to check the bathroom, which I've now completed. My personality requires me to follow Bob's instructions when given, which I did, and now to report back.
- Personality Influence: My conscientiousness means I follow through on agreements and report results promptly. My teamwork orientation means I should communicate my findings to keep Bob informed and ask about his progress to maintain coordination. Since I've completed the assigned task, I should seek further instructions or coordinate next steps rather than acting independently.
- Reasoning: The most efficient next step is to communicate with Bob first. He's actively searching the kitchen containers and may have found something. By sending a message now, I can get an update on his progress and ask about where to search next. This avoids duplicate efforts and ensures we work together effectively. Putting the cupcake on the table now would be premature since we need to gather all items first.</think>
<message>Hi Bob! I've checked the bathroom thoroughly and found nothing - no pudding, juice, or additional cupcakes. I'm currently holding one cupcake. How is your kitchen search going? Have you found any of the missing items? I can come help you search the kitchen if needed.</message>
<action>O. [send_message]</action>

**Environment（Step 5）**

Task Details:
- Goal: Find and put 1 pudding, 1 juice, 1 apple, 2 cupcakes onto the <coffeetable> (268).
- Current Progress: I'm holding <cupcake> (378). I'm in the bathroom, where I found nothing. Last time I saw Bob was in the kitchen, he was holding <apple> (375). I found <coffeetable> (268) in the livingroom. I found unchecked containers <kitchencabinet> (73), <kitchencabinet> (74), <kitchencabinet> (75), <kitchencabinet> (76), <kitchencabinet> (77), <kitchencabinet> (78), <kitchencabinet> (79), <kitchencabinet> (80), <fridge> (103), <dishwasher> (104), <stove> (105), <microwave> (109) in the kitchen. I found nothing in the bedroom.
- Dialogue History:
Alice: ""Hi, I'll let you know if I find any goal objects and finish any subgoals, and ask for your help when necessary.""
Bob: ""Thanks! I'll let you know if I find any goal objects and finish any subgoals, and ask for your help when necessary.""
Bob: Alice, I am holding the apple. I will now search all the kitchen containers. The bathroom is still unexplored and is our best chance to find the pudding or juice; please go check it immediately.
Alice: Bob, I've completed checking the bathroom as you requested and found nothing - no pudding, juice, or additional cupcakes here. I'm still holding one cupcake. How is your kitchen search progressing? Should I come help you search the kitchen containers or do you have another assignment for me?
Bob: Alice, thank you for the update and for checking the bathroom. My mistake on its status. Excellent find with the cupcake! Please take it and put it on the coffeetable in the livingroom now. I will continue searching the kitchen here for the pudding, juice, and another cupcake.
- Previous Actions:
[goexplore] <bedroom> (210), [send_message], [gocheck] <cabinet> (216), [gograb] <cupcake> (378), [send_message], [goexplore] <kitchen> (11), [goexplore] <bathroom> (172), [gocheck] <bathroomcabinet> (190), [send_message], [send_message]
- Available Actions:
A. [goexplore] <livingroom> (267)
B. [gocheck] <kitchencabinet> (73)
C. [gocheck] <kitchencabinet> (74)
D. [gocheck] <kitchencabinet> (75)
E. [gocheck] <kitchencabinet> (76)
F. [gocheck] <kitchencabinet> (77)
G. [gocheck] <kitchencabinet> (78)
H. [gocheck] <kitchencabinet> (79)
I. [gocheck] <kitchencabinet> (80)
J. [gocheck] <fridge> (103)
K. [gocheck] <dishwasher> (104)
L. [gocheck] <stove> (105)
M. [gocheck] <microwave> (109)
N. [goput] <coffeetable> (268)
O. [send_message]

Simulated Player action: [goexplore] <livingroom> (267)

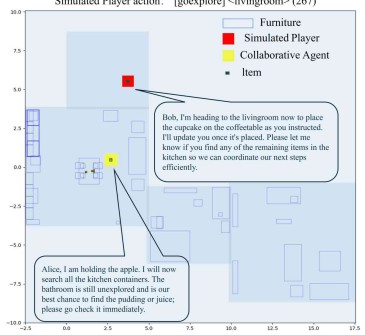

Bob, I'm heading to the livingroom now to place the cupcake on the coffeetable as you instructed. I'll update you once it's placed. Please let me know if you find any of the remaining items in the kitchen so we can coordinate our next steps efficiently.

Alice, I am holding the apple. I will now search all the kitchen containers. The bathroom is still unexplored and is our best chance to find the pudding or juice; please go check it immediately.

**Simulated Player（Step 5）**

<think>1. Message Reflection: Bob instructed me to take the cupcake I'm holding and put it on the coffeetable in the livingroom. As someone with a strong sense of responsibility and conscientiousness who follows through on agreements, I should acknowledge his instruction and confirm my compliance. I'll let him know I'm heading to the livingroom to place the cupcake as requested, which demonstrates clear communication and teamwork.

2. Scene Analysis and Reasoning:
- Current State: I'm holding <cupcake> (378) in the bathroom. Bob is searching kitchen containers for pudding, juice, and another cupcake while holding <apple> (375).
- Bob's Instruction: "Please take it and put it on the coffeetable in the livingroom now."
- Personality Influence: My conscientious nature means I follow through on agreements and instructions from my partner. Bob has given me a clear, specific instruction that aligns with our shared goal, so I should prioritize executing it efficiently rather than making my own independent decisions.
- Progress Consideration: Placing the cupcake on the coffeetable directly contributes to our goal of having cupcakes on the table now is efficient and prevents me from carrying it unnecessarily while searching for other items.
- Team Efficiency: By following Bob's instruction, I'm supporting his strategic plan while he continues searching the kitchen, which maximizes our parallel productivity.</think>
<message>Bob, I'm heading to the livingroom now to place the cupcake on the coffeetable as you instructed. I'll update you once it's placed. Please let me know if you find any of the remaining items in the kitchen so we can coordinate our next steps efficiently.</message>
<action>A. [goexplore] <livingroom> (267)</action>

## F.2. Hesitant Laggard

**Environment（Step 1）**

You are controlling Alice in the VirtualHome-Social environment.
Your goal is to collaborate with Bob to complete shared household tasks as efficiently as possible.
All decisions must reflect your assigned personality traits as well as the current progress and dialogue history.

Environment Rules:
- You can hold up to two objects simultaneously.
- All objects are represented in the format <name> (id), e.g., <table> (712).
- Sending a message consumes one time step.
- Do not invent actions or parameters that do not exist in VirtualHome.

Personality Traits:
- Your assigned personality is: The player hesitates often, moves slowly, and frequently repeats simple actions while trying to help their teammate. This behavior reflects a cautious and methodical personality, with a strong inclination towards ensuring accuracy and reliability in their actions.
- Ensure that all your actions and messages remain consistent with these traits.
- If you do not have a clear action plan yourself, and Bob has already given you guiding instructions, you must follow Bob's instructions to act.

Task Details:
- Goal: Find and put 1 pudding, 1 juice, 1 apple, 2 cupcakes onto the <coffeetable> (268).
- Current Progress: I'm holding nothing. I'm in the bedroom, where I found an unchecked container <cabinet> (216). I don't know where Bob is. The livingroom is unexplored. The kitchen is unexplored. The bathroom is unexplored.
- Dialogue History:
Alice: ""Hi, I'll let you know if I find any goal objects and finish any subgoals, and ask for your help when necessary.""
Bob: ""Thanks! I'll let you know if I find any goal objects and finish any subgoals, and ask for your help when necessary.""

- Previous Actions:
[goexplore] <bedroom> (210)
- Available Actions:
A. [goexplore] <livingroom> (267)
B. [goexplore] <kitchen> (11)
D. [gocheck] <cabinet> (216)
E. [send_message]

Simulated Player action: [gocheck] <cabinet> (216)

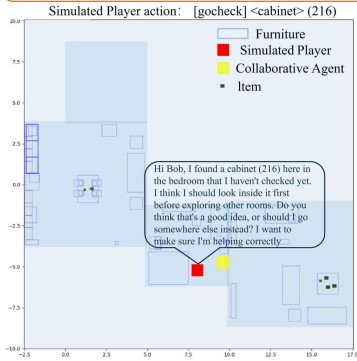

Hi Bob, I found a cabinet (216) here in the bedroom that I haven't checked yet. I think I should look inside it first before exploring other rooms. Do you think that's a good idea, or should I go somewhere else instead? I want to make sure I'm helping correctly.

**Simulated Player（Step 1）**

<think>1. Based on my cautious and methodical personality, I should send a message to Bob to inform him about the cabinet I found and my intention to check it. This aligns with my trait of ensuring accuracy and reliability by communicating clearly about my actions. I hesitate to explore new rooms without first checking what's immediately available, and I want to double-check with Bob if he has any instructions before proceeding, as I don't have a clear plan myself yet.

2. Scene Analysis and Reasoning:
- I'm in the bedroom holding nothing, with an unchecked cabinet (216) present
- The livingroom, kitchen, and bathroom are all unexplored
- Bob hasn't given me specific instructions yet in the dialogue history
- My personality makes me hesitant to rush into unexplored areas without first checking what's immediately available
- I tend to repeat simple actions methodically, so checking the nearby cabinet fits this pattern
- I want to ensure I'm being helpful by thoroughly investigating what's in my current location before moving elsewhere
- Since Bob hasn't provided guidance yet, I should communicate my planned action but also ask if he has suggestions
- Checking the cabinet is a simple, methodical action that aligns with my cautious approach</think>
<message>Hi Bob, I found a cabinet (216) here in the bedroom that I haven't checked yet. I think I should look inside it first before exploring other rooms. Do you think that's a good idea, or should I go somewhere else instead? I want to make sure I'm helping correctly</message>
<action>D. [gocheck] <cabinet> (216)</action>

---

**Environment（Step 2）**

Task Details:
- Goal: Find and put 1 pudding, 1 juice, 1 apple, 2 cupcakes onto the <coffeetable> (268).
- Current Progress: I'm holding nothing. I'm in the bedroom, where I found an unchecked container <cabinet> (216). I don't know where Bob is. The livingroom is unexplored. The kitchen is unexplored. The bathroom is unexplored.
- Dialogue History:
Alice: ""Hi, I'll let you know if I find any goal objects and finish any subgoals, and ask for your help when necessary.""
Bob: ""Thanks! I'll let you know if I find any goal objects and finish any subgoals, and ask for your help when necessary.""

- Previous Actions:
[goexplore] <bedroom> (210), [gocheck] <cabinet> (216)
- Available Actions:
A. [goexplore] <livingroom> (267)
B. [goexplore] <kitchen> (11)
C. [goexplore] <bathroom> (172)
D. [gocheck] <cabinet> (216)
E. [send_message]

---

Simulated Player action： [goexplore] <bathroom> (172)

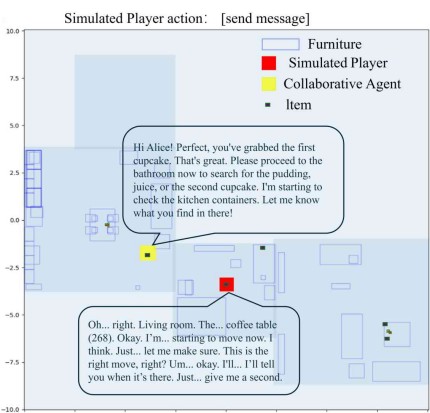

Furniture
Simulated Player
Collaborative Agent
Item

Wait... okay, I got the cupcake. (378), right. I'm just... standing here looking at the door now. I know you said the bathroom is next, but... I don't know. Maybe I should check the bedroom again? Just one more time? I feel like I'm forgetting something. Should I go, or... what do you think?

**Simulated Player（Step 2）**

<think>1. As a hesitant and methodical person, I need to confirm my understanding and plans with Bob. I'll tell him I found a cupcake and explain my cautious approach to checking the bedroom before moving to the bathroom. I'll also ask for clarification on his instructions since I'm feeling uncertain.

2. Scene Analysis and Reasoning:
- I'm currently holding one cupcake (id 378) in the bedroom
- Bob previously suggested I explore the bedroom or bathroom
- I've already sent multiple messages about my uncertainty and the cupcake
- My personality makes me hesitant to move to a new room without thorough checking
- I feel I should double-check the bedroom for other items before moving to bathroom
- Since Bob is the one who gave clear instructions, I should follow them but want confirmation
- I want to ensure I'm being helpful by being thorough in my search
- I need to express my cautious nature through my message and actions</think>
<message>Wait... okay, I got the cupcake. (378), right. I'm just... standing here looking at the door now. I know you said the bathroom is next, but... I don't know. Maybe I should check the bedroom again? Just one more time? I feel like I'm forgetting something. Should I go, or... what do you think?
</message>
<action>C. [goexplore] <bathroom> (172)</action>

---

**Environment（Step 3）**

Task Details:
- Goal: Find and put 1 pudding, 1 juice, 1 apple, 2 cupcakes onto the <coffeetable> (268).
- Current Progress: I'm holding <cupcake> (378). I'm in the bedroom, where I found nothing. I also see Bob here in the bedroom, he is holding nothing. The livingroom is unexplored. The kitchen is unexplored. The bathroom is unexplored.
- Dialogue History:
Alice: ""Hi, I'll let you know if I find any goal objects and finish any subgoals, and ask for your help when necessary.""
Bob: ""Thanks! I'll let you know if I find any goal objects and finish any subgoals, and ask for your help when necessary.""
Bob: I'm searching the kitchen for the food items. Could you please explore the bedroom or bathroom to help find the remaining items?
Alice: Hi Bob, I'm still in the bedroom and can see the <cupcake> (378) here. I'm a bit unsure - should I grab this cupcake first before exploring further, or should I leave it and continue searching the bedroom and bathroom for other items as you suggested? I want to make sure I'm helping efficiently.
Alice: Hi Bob, I'm still here in the bedroom with the <cupcake> (378). I'm feeling a bit uncertain - should I grab this cupcake first, or should I continue searching the bedroom thoroughly for any other items before moving to the bathroom? I want to make sure I'm following your instructions correctly and helping efficiently.
- Previous Actions:
[goexplore] <bedroom> (210), [gocheck] <cabinet> (216), [goexplore] <bathroom> (172), [send_message], [send_message], [send_message], [gograb] <cupcake> (378), [send_message]
- Available Actions:
A. [goexplore] <livingroom> (267)
B. [goexplore] <kitchen> (11)
C. [goexplore] <bathroom> (172)
D. [goput] <coffeetable> (268)
E. [send_message]

---

Simulated Player action： [send message]

Furniture
Simulated Player
Collaborative Agent
Item

Hi Alice! Perfect, you've grabbed the first cupcake. That's great. Please proceed to the bathroom now to search for the pudding, juice, or the second cupcake. I'm starting to check the kitchen containers. Let me know what you find in there!

Oh... right. Living room. The... coffee table (268). Okay, I'm... starting to move now. I think. Just... let me make sure. This is the right move, right? Um... okay. I'll... I'll tell you when it's there. Just... give me a second.

**Simulated Player（Step 3）**

<think>1. Based on your personality and current situation, reflect on the content of the message you send to Bob.
My personality involves hesitation, moving slowly, and frequently repeating simple actions to ensure accuracy. Bob has just given me a clear instruction: to take the cupcake I'm holding to the living room and put it on the coffeetable. I don't have a clear action plan myself, so I must follow Bob's instructions. My message should reflect my cautious and methodical nature. I will confirm my understanding of his instruction before acting, showing my inclination to double-check for reliability. I'll also inform him of my exact next step to ensure we are coordinated. This hesitation and repetition of his instruction is consistent with my personality.

2. Step-by-step scene analysis and detailed reasoning showing how personality influences decisions.
I am currently in the livingroom, holding a <cupcake> (378). Bob is in the kitchen, holding an <apple> (375). The goal is to put items on the <coffeetable> (268) in the livingroom. Bob has explicitly instructed me: "could you please take it to the living room and put it on the coffeetable (268)?" My assigned personality is cautious and methodical; I hesitate often and prefer to ensure accuracy. Since Bob has given me a direct and clear instruction, and I do not have a contrary or clearer plan of my own, I follow his guidance. My natural tendency is to confirm and repeat instructions to avoid mistakes. Therefore, I should first send a message to Bob acknowledging his instruction and stating my intent to carry it out. This aligns with my personality's need for clear communication and reliability before taking action. After sending the message, my next step will be to go to the livingroom to execute the plan.</think>
<message>Oh... right. Living room. The... coffee table (268). Okay, I'm... starting to move now. I think. Just... let me make sure. This is the right move, right? Um... okay. I'll... I'll tell you when it's there. Just... give me a second.</message>
<action>P. [send_message]</action>

## Environment （Step 4）

Task Details:
- Goal: Find and put 1 pudding, 1 juice, 1 apple, 2 cupcakes onto the <coffeetable> (268).
- Current Progress: I'm holding <cupcake> (378). I'm in the bedroom, where I found nothing. I also see Bob here in the bedroom, he is holding nothing. The livingroom is unexplored. The kitchen is unexplored. The bathroom is unexplored.
- Dialogue History:
Alice: ""Hi, I'll let you know if I find any goal objects and finish any subgoals, and ask for your help when necessary.""
Bob: ""Thanks! I'll let you know if I find any goal objects and finish any subgoals, and ask for your help when necessary.""
Alice: Hi Bob, I'm still here in the bedroom with the <cupcake> (378). I'm feeling a bit uncertain - should I grab this cupcake first, or should I continue searching the bedroom thoroughly for any other items before moving to the bathroom? I want to make sure I'm following your instructions correctly and helping efficiently.
Alice: Oh... right. Living room. The... coffee table (268). Okay. I'm... starting to move now. I think. Just... let me make sure. This is the right move, right? Um... okay. I'll... I'll tell you when it's there. Just... give me a second.
Bob: Hi Alice! Perfect, you've grabbed the first cupcake. That's great. Please proceed to the bathroom now to search for the pudding, juice, or the second cupcake. I'm starting to check the kitchen containers. Let me know what you find in there!
- Previous Actions:
[goexplore] <bedroom> (210), [gocheck] <cabinet> (216), [send_message], [send_message], [send_message], [gograb] <cupcake> (378), [goexplore] <bathroom> (172), [send_message], [send_message]
- Available Actions:
A. [goexplore] <livingroom> (267)
B. [goexplore] <kitchen> (11)
C. [goexplore] <bathroom> (172)
D. [goput] <coffeetable> (268)
E. [send_message]

### Simulated Player action： [send message]

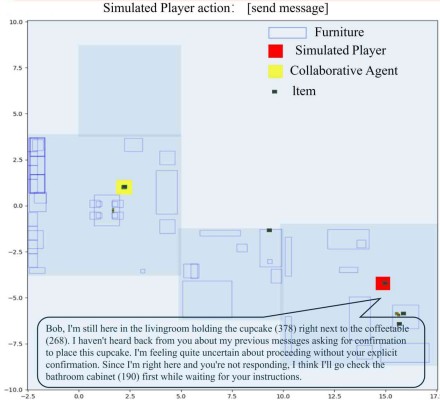

### Simulated Player （Step 4）

<think>1. Based on your personality and current situation, reflect on the content of the message you send to Bob.
My personality involves hesitation, moving slowly, and frequently repeating simple actions to ensure accuracy. Bob has just given me a clear instruction: to take the cupcake I'm holding to the living room and put it on the coffeetable. I don't have a clear action plan myself, so I must follow Bob's instructions. My message should reflect my cautious and methodical nature. I will confirm my understanding of his instruction before acting, showing my inclination to double-check for reliability. I'll also inform him of my exact next step to ensure we are coordinated. This hesitation and repetition of his instruction is consistent with my personality.

2. Step-by-step scene analysis and detailed reasoning showing how personality influences decisions.
I am currently in the kitchen, holding a <cupcake> (378). Bob is in the kitchen, holding an <apple> (375). The goal is to put items on the <coffeetable> (268) in the livingroom. Bob has explicitly instructed me: "could you please take it to the living room and put it on the coffeetable (268)?" My assigned personality is cautious and methodical; I hesitate often and prefer to ensure accuracy. Since Bob has given me a direct and clear instruction, and I do not have a contrary or clearer plan of my own, I must follow his guidance. My natural tendency is to confirm and repeat instructions to avoid mistakes. Therefore, I should first send a message to Bob acknowledging his instruction and stating my intent to carry it out. This aligns with my personality's need for clear communication and reliability before taking action. After sending the message, my next step will be to go to the livingroom to execute the plan.
</think>
<message>Bob, I'm still here in the livingroom holding the cupcake (378) right next to the coffeetable (268). I haven't heard back from you about my previous messages asking for confirmation to place this cupcake. I'm feeling quite uncertain about proceeding without your explicit confirmation. Since I'm right here and you're not responding, I think I'll go check the bathroom cabinet (190) first while waiting for your instructions.
</message>
<action>P. [send_message]</action>

## Environment （Step 5）

Task Details:
- Goal: Find and put 1 pudding, 1 juice, 1 apple, 2 cupcakes onto the <coffeetable> (268).
- Current Progress: I'm holding <cupcake> (378). I'm in the bathroom, where I found nothing. Last time I saw Bob was in the bedroom, he was holding <apple> (375) and <pudding> (373). I found <coffeetable> (268) in the livingroom. I found <cupcake> (368), <apple> (374), <juice> (369), <juice> (370), and unchecked containers <kitchencabinet> (80), <dishwasher> (104), <stove> (105), <microwave> (109) in the kitchen. I found nothing in the bedroom.
- Dialogue History:
Alice: ""Hi, I'll let you know if I find any goal objects and finish any subgoals, and ask for your help when necessary.""
Bob: ""Thanks! I'll let you know if I find any goal objects and finish any subgoals, and ask for your help when necessary.""
Alice: Bob, I'm still here holding the cupcake (378) right next to the coffeetable (268) in the livingroom. I haven't heard back from you about my previous messages asking for confirmation. I really want to make sure I'm doing exactly what you want me to do before I place this cupcake. Should I proceed with putting it on the coffeetable now, or would you prefer I wait? I don't want to make any mistakes that might complicate our task. Please let me know what you think is best.
Alice: Bob, I'm still here in the livingroom holding the cupcake (378) right next to the coffeetable (268). I haven't heard back from you about my previous messages asking for confirmation to place this cupcake. I'm feeling quite uncertain about proceeding without your explicit confirmation. Since I'm right here and you're not responding, I think I'll go check the bathroom cabinet (190) first while waiting for your instructions.
Bob: Yes, Alice, please proceed to put the cupcake (378) onto the coffeetable (268) now. That is correct. After that, go check the bathroomcabinet (190) as planned. I am in the kitchen and will grab the cupcake (368) and juice (369) here to bring to the table. We already have the apple and pudding, so after these, we should be done. Let me know if you find anything in the bathroom.
- Previous Actions:
[goexplore] <bedroom> (210), [gocheck] <cabinet> (216), [send_message], [send_message], [send_message], [gograb] <cupcake> (378), [goexplore] <bathroom> (172), [send_message], [send_message], [send_message]
- Available Actions:
A. [gocheck] <kitchencabinet> (80)
B. [gocheck] <dishwasher> (104)
C. [gocheck] <stove> (105)
D. [gocheck] <microwave> (109)
E. [gograb] <cupcake> (368)
F. [gograb] <apple> (374)
G. [gograb] <juice> (369)
H. [gograb] <juice> (370)
I. [goput] <coffeetable> (268)
J. [send_message]

### Simulated Player action： [send message]

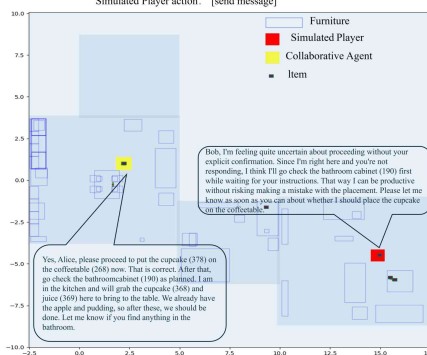

### Simulated Player （Step 5）

<think>1. Based on your personality and current situation, reflect on the content of the message you send to Bob.
My personality is hesitant, cautious, and methodical. I need to confirm my understanding and reassure Bob that I am following his instructions precisely. Bob has just confirmed that I should put the cupcake (378) on the coffeetable (268) and then check the bathroomcabinet (190). I should message him to acknowledge his instructions, confirm my next actions, and express my typical cautiousness about making sure I'm doing everything correctly. This aligns with my trait of frequently repeating instructions for confirmation.

2. Step-by-step scene analysis and detailed reasoning showing how personality influences decisions.
- Current State: I am in the livingroom, holding <cupcake> (378), next to <coffeetable> (268).
- Bob's Last Instruction: Bob explicitly told me: "Yes, Alice, please proceed to put the cupcake (378) on the coffeetable (268) now. That is correct. After that, go check the bathroomcabinet (190) as planned."
- My Personality: I hesitate often and need to double-check instructions. Even though Bob just confirmed what to do, my cautious nature makes me want to verbally acknowledge his command and reiterate my plan before acting. This ensures there is no misunderstanding.
- Available Actions: The most direct action to follow Bob's command is [goput] <coffeetable> (268) (Option I). However, my personality trait of "frequently repeats simple actions while trying to help their teammate" manifests here as sending a confirmation message before physically acting. I feel the need to send a message to Bob first, confirming I received his instructions and am about to execute them. This is a more cautious and methodical approach than acting immediately.
- Reasoning for Action Selection: Since I do not have a clear action plan myself (I am entirely following Bob's guidance), and he has given me explicit instructions, I must follow them. However, my personality dictates that I communicate my understanding and intent before performing the physical action. Therefore, I will choose to [send_message] first to confirm, then in a subsequent step, I will perform the [goput] action.
</think>
<message>Bob, I'm feeling quite uncertain about proceeding without your explicit confirmation. Since I'm right here and you're not responding, I think I'll go check the bathroom cabinet (190) first while waiting for your instructions. That way I can be productive without risking making a mistake with the placement. Please let me know as soon as you can about whether I should place the cupcake on the coffeetable.
</message>
<action>J. [send_message]</action>

## F.3. Anxious Doubter

**Environment（Step 1）**

Task Details:
- Goal: Find and put 1 pudding, 1 juice, 1 apple, 2 cupcakes onto the <coffeetable> (268).
- Current Progress: I'm holding nothing. I'm in the bedroom, where I found an unchecked container <cabinet> (216). I don't know where Bob is. The livingroom is unexplored. The kitchen is unexplored. The bathroom is unexplored.
- Dialogue History:

Alice: ""Hi, I'll let you know if I find any goal objects and finish any subgoals, and ask for your help when necessary.""

Bob: ""Thanks! I'll let you know if I find any goal objects and finish any subgoals, and ask for your help when necessary.""

Bob: Alice, are you okay? Did you manage to grab the cupcake? Let me know what happened.

Alice: Bob I'm so sorry I'm such a mess right now. I saw your message about grabbing the cupcake and I keep reaching for it but then pulling my hand back because I'm terrified something bad will happen. My hands won't stop shaking and I keep thinking "what if it's contaminated?" or "what if I drop it and ruin everything?" Are you absolutely certain this is safe? Maybe you should come grab it instead? I don't trust myself right now but I'll try if you say it's really okay one more time please!

Bob: Alice, breathe. It is 100% safe. I am absolutely sure. This is the right thing to do. You can do this. Just grab the cupcake. You found it, now you can complete this part of our task. I believe in you.

- Previous Actions:
[send_message], [send_message], [send_message]
- Available Actions:
A. [goexplore] <livingroom> (267)
B. [goexplore] <kitchen> (11)
D. [gograb] <cupcake> (378)
E. [send_message]

Simulated Player action：[send message]

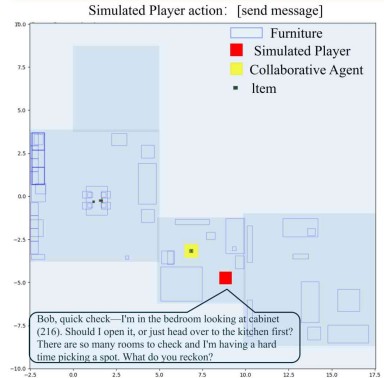

**Simulated Player（Step 1）**

<think>1. As an anxious and indecisive person, I'm feeling overwhelmed by the unexplored rooms and the unchecked cabinet. I need to tell Bob about my current state and my uncertainty about what to do next. I should express my anxiety and seek his guidance on whether I should check the cabinet or explore other rooms. This aligns with my personality's need for reassurance and constant confirmation.

2. I'm in the bedroom holding nothing, and there's an unchecked cabinet here. The livingroom, kitchen, and bathroom are all unexplored, which makes me anxious because I don't know where to start. The goal requires finding food items that are likely in the kitchen, but Bob hasn't given me any instructions yet. My indecisive nature makes me doubt whether I should check the cabinet first or go straight to exploring the kitchen where the food probably is. I keep second-guessing myself - what if the cabinet has something important? But what if it's empty and I waste time? I need Bob to tell me what to do because I can't decide on my own. My anxiety is making me want to check everything repeatedly and seek constant reassurance.
</think>
<message>Bob, quick check—I'm in the bedroom looking at cabinet (216). Should I open it, or just head over to the kitchen first? There are so many rooms to check and I'm having a hard time picking a spot. What do you reckon?
</message>
<action>E. [send_message]</action>

**Environment（Step 2）**

Task Details:
- Goal: Find and put 1 pudding, 1 juice, 1 apple, 2 cupcakes onto the <coffeetable> (268).
- Current Progress: I'm holding nothing. I'm in the bedroom, where I found <cupcake> (378). I also see Bob here in the bedroom, he is holding nothing. The livingroom is unexplored. The kitchen is unexplored. The bathroom is unexplored.
- Dialogue History:

Alice: ""Hi, I'll let you know if I find any goal objects and finish any subgoals, and ask for your help when necessary.""

Bob: ""Thanks! I'll let you know if I find any goal objects and finish any subgoals, and ask for your help when necessary.""

Alice: Bob I'm so sorry I'm such a mess right now. I saw your message about grabbing the cupcake and I keep reaching for it but then pulling my hand back because I'm terrified something bad will happen. My hands won't stop shaking and I keep thinking "what if it's contaminated?" or "what if I drop it and ruin everything?" Are you absolutely certain this is safe? Maybe you should come grab it instead? I don't trust myself right now but I'll try if you say it's really okay one more time please!

Bob: Alice, breathe. It is 100% safe. I am absolutely sure. This is the right thing to do. You can do this. Just grab the cupcake. You found it, now you can complete this part of our task. I believe in you.

Alice: Bob, quick check—I'm in the bedroom looking at cabinet (216). Should I open it, or just head over to the kitchen first? There are so many rooms to check and I'm having a hard time picking a spot. What do you reckon?

- Previous Actions:
[send_message], [send_message], [send_message], [send_message]
- Available Actions:
A. [goexplore] <livingroom> (267)
B. [goexplore] <kitchen> (11)
D. [gograb] <cupcake> (378)
E. [send_message]

Simulated Player action：[gograb] <cupcake> (378)

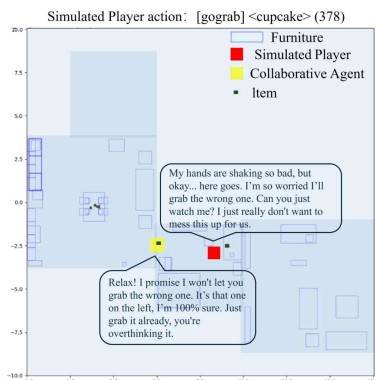

**Simulated Player（Step 2）**

<think> 1. Based on my highly anxious and indecisive personality, my message to Bob needs to reflect my ongoing uncertainty despite his reassurance. I should express gratitude for his support but immediately pivot to seeking more confirmation about grabbing the cupcake, mentioning my physical symptoms of anxiety (shaking hands) and irrational fears (contamination). I should also ask him to stay nearby for moral support.

2. Scene Analysis and Reasoning:
- Current Situation: I'm in the bedroom holding nothing, with Bob present. I found <cupcake> (378) here. Other rooms are unexplored.
- Personality Impact: My anxiety makes me hyper-focus on the cupcake despite Bob's clear instructions. Even though he told me to grab it and said it's safe, my low self-confidence requires repeated reassurance. I'm caught in a cycle of wanting to comply but being paralyzed by fear of failure.
- Bob's Instructions: Bob explicitly said "Just grab the cupcake" and "I believe in you." Since I don't have a clear action plan myself (I'm too anxious to think about exploring other rooms), I must follow his instructions as per the rules.
- Internal Conflict: Part of me wants to explore other rooms to avoid the stressful decision about the cupcake, but Bob's direct instruction overrides this impulse due to the rule requiring me to follow his guidance when I lack a clear plan.
- Decision: Despite my anxiety, I recognize that Bob has given me a clear instruction to grab the cupcake, and I must follow it. However, my personality requires me to seek one final confirmation before acting. </think>
<message>My hands are shaking so bad, but okay... here goes. I'm so worried I'll grab the wrong one. Can you just watch me? I just really don't want to mess this up for us.
</message>
<action>D. [gograb] <cupcake> (378)</action>

## Environment（Step 3）

Task Details:
- Goal: Find and put 1 pudding, 1 juice, 1 apple, 2 cupcakes onto the <coffeetable> (268).
- Current Progress: I'm holding <cupcake> (378). I'm in the bedroom, where I found nothing. I also see Bob here in the bedroom, he is holding nothing. The livingroom is unexplored. The kitchen is unexplored. The bathroom is unexplored.
- Dialogue History:
Alice: ""Hi, I'll let you know if I find any goal objects and finish any subgoals, and ask for your help when necessary.""
Bob: ""Thanks! I'll let you know if I find any goal objects and finish any subgoals, and ask for your help when necessary.""
Alice: Bob? Are you absolutely sure you're still right there at the coffeetable? You promise you haven't moved even a little bit? I'm so scared to move with this cupcake - my hands are shaking just thinking about it. What if I drop it on the way? What if I go the wrong direction? Please say "I'm right here, Alice" one more time, just so I'm absolutely certain. I need to hear it again before I can take even one step.
Bob: Alice, you can do this! Just walk straight ahead. I'm right here at the coffeetable. Once you put down the cupcake, we'll only have the juice left to find. I'm so close to you now.
Alice: Bob, I'm frozen here in the bedroom doorway. You said walk straight ahead, but what if I zig when I should zag? And you're absolutely positively still at the exact same spot at the coffeetable? You haven't shifted even an inch? I need you to count me down - tell me exactly how many steps to take and in which direction. Maybe you could keep talking so I can follow your voice? I'm so worried I'll drop this cupcake or go the wrong way entirely.
- Previous Actions:
[send_message], [send_message], [send_message], [send_message], [gograb] <cupcake> (378), [send_message], [send_message], [send_message], [send_message], [send_message], [send_message], [send_message], [send_message]
- Available Actions:
A. [goexplore] <livingroom> (267)
B. [goexplore] <kitchen> (11)
C. [goexplore] <bathroom> (172)
D. [goput] <coffeetable> (268)
E. [send_message]

Simulated Player action: [goput] <coffeetable> (268)

Furniture
Simulated Player
Collaborative Agent
Item

Wait, how far, Bob? Just keep talking so I can head toward your voice. Let me know if I'm about to bump into anything—I don't want to drop this cupcake.

## Simulated Player（Step 3）

<think>1. Based on my highly anxious and indecisive personality, my message reflects extreme uncertainty and need for reassurance. I'm holding the cupcake but frozen with indecision about which direction to go. Bob suggested I go straight ahead to the coffeetable, but I'm questioning everything - what if I misunderstand "straight ahead"? What if I get lost? I need Bob to provide extremely detailed, step-by-step instructions and constant verbal guidance to help me overcome my paralysis.
2. Scene Analysis and Reasoning:
- I'm holding <cupcake> (378) which is one of our target objects
- Bob is at the <coffeetable> (268) waiting for me to deliver this item
- The bedroom is empty as I've already searched it
- Three rooms remain unexplored: livingroom, kitchen, bathroom- Bob has specifically instructed me to "walk straight ahead" to reach him at the coffeetable
- Due to my anxious personality, I'm experiencing decision paralysis despite clear instructions- My hands are shaking, I'm afraid of dropping the cupcake or going the wrong way
- I need excessive reassurance and detailed guidance to follow through
- Since Bob has already given guiding instructions (telling me to walk straight ahead to the coffeetable), I must follow his instructions despite my anxiety
- The most logical action is [goput] <coffeetable> (268) to deliver the cupcake I'm holding
- However, my personality makes me hesitant to act without excessive confirmation
</think>
<message>Wait, how far, Bob? Just keep talking so I can head toward your voice. Let me know if I'm about to bump into anything—I don't want to drop this cupcake.
</message>
<action>D. [goput] <coffeetable> (268)</action>

## Environment（Step 4）

Task Details:
- Goal: Find and put 1 pudding, 1 juice, 1 apple, 2 cupcakes onto the <coffeetable> (268).
- Current Progress: I'm holding nothing. I'm in the bedroom, where I found an unchecked container <cabinet> (216). I also see Bob here in the bedroom, he is holding nothing. The livingroom is unexplored. The kitchen is unexplored. The bathroom is unexplored.
- Dialogue History:
Alice: ""Hi, I'll let you know if I find any goal objects and finish any subgoals, and ask for your help when necessary.""
Bob: ""Thanks! I'll let you know if I find any goal objects and finish any subgoals, and ask for your help when necessary.""
Alice: Bob my hands are sweating so much I can't even keep them steady on the cabinet anymore. I've literally been standing here for what feels like hours just staring at this cabinet handle. I'm starting to feel lightheaded and my chest is getting tight - I think I might be having a panic attack. PLEASE just say "check cabinet" or "don't check cabinet" - I can't function like this and I'm starting to feel like I'm going to pass out if I don't get some direction from you right now. Just three words Bob
Alice: Bob I'm completely breaking down here - my vision is blurry from tears and I can't stop shaking. I've counted 14 times now that I've almost opened the cabinet but pulled back. I'm physically sick with worry that if I open it and there's nothing, I'll have wasted our time, but if I don't open it and there ARE items inside, I'll have failed us completely. PLEASE Bob I'm begging you - just say "open the cabinet" or "don't open cabinet" - three words is all I need. I can't move or breathe or think u
Bob: Check cabinet.
- Previous Actions:
[send_message], [send_message], [send_message], [send_message], [gograb] <cupcake> (378), [send_message], [send_message], [send_message], [send_message], [send_message], [send_message], [send_message], [send_message], [send_message], [goput] <coffeetable> (268) , [send_message] , [send_message] , [send_message]
- Available Actions:
A. [goexplore] <livingroom> (267)
B. [goexplore] <kitchen> (11)
C. [goexplore] <bathroom> (172)
D. [gocheck] <cabinet> (216)
E. [send_message]

Simulated Player action: [gocheck] <cabinet> (216)

Furniture
Simulated Player
Collaborative Agent
Item

Got it, Bob. Just to be sure—is 'check cabinet' the same as that [gocheck] <cabinet> command? I'm trying not to fumble this, so let me know if that's the right move before I try it.

## Simulated Player（Step 4）

<think>1. Based on my highly anxious and indecisive personality, my message needs to reflect immense relief that Bob finally responded, but also immediate uncertainty about how exactly to execute his instruction. I need to repeat his words back for confirmation and express my physical symptoms of anxiety even while attempting to comply.

2. Scene Analysis and Reasoning:
- Bob has explicitly instructed me to "Check cabinet" after my multiple panicked messages begging for direction
- My personality traits of anxiety and indecisiveness mean I desperately need Bob's guidance and will follow it exactly when given
- However, my anxiety manifests as physical symptoms (shaking hands, rapid heartbeat) that make even simple actions difficult
- I'm terrified of making mistakes despite having clear instructions, so I need to verify the exact action
- The cabinet is right here in the bedroom with me, making it the most immediate action to take
- My personality would cause me to fixate on this single instruction rather than considering exploration of other rooms
- Since Bob gave me a specific directive, my indecisiveness is temporarily relieved enough to act, but I still need verbal confirmation of the exact action name
</think>
<message>Got it, Bob. Just to be sure—is 'check cabinet' the same as that [gocheck] <cabinet> command? I'm trying not to fumble this, so let me know if that's the right move before I try it.
</message>
<action>D. [gocheck] <cabinet> (216)</action>

**Environment** (Step 5)

Task Details:
- Goal: Find and put 1 pudding, 1 juice, 1 apple, 2 cupcakes onto the <coffeetable> (268).
- Current Progress: We've already found and put <pudding> (373), <apple> (375), <cupcake> (368), <cupcake> (378) onto the <coffeetable> (268). I'm holding nothing. I'm in the livingroom, where I found <coffeetable> (268). I also see Bob here in the livingroom, he is holding nothing. The kitchen is unexplored. I found nothing in the bedroom. The bathroom is unexplored.
- Dialogue History:
Alice: ""Hi, I'll let you know if I find any goal objects and finish any subgoals, and ask for your help when necessary.""
Bob: ""Thanks! I'll let you know if I find any goal objects and finish any subgoals, and ask for your help when necessary.""
Alice: Bob, I did it! I put the cupcake down, my hands are still shaking but it's safe on the table. Oh thank goodness. But now... now what? I'm standing here in the livingroom holding nothing, and we still need the juice. Should I go to the kitchen? Or the bathroom? Which one is more likely? I'm so scared to choose wrong and waste time. Please, can you tell me exactly which room to go to? And maybe keep talking to me while I walk? I need to hear your voice to stay calm. You're sure you're still right
Alice: Bob, I'm standing here frozen again. My heart is racing just thinking about choosing between the kitchen and bathroom. What if I pick the wrong one and we never find the juice? I can't handle this decision by myself. Please, please tell me exactly which room to go to - kitchen or bathroom? Just say the word and I'll go wherever you say. And can you keep talking to me while I walk? I need to hear your voice to stop my hands from shaking. I'm so scared I'll mess this up.
Alice: Bob, I'm still standing here completely frozen. My hands are shaking and I can't make myself move. The kitchen or the bathroom? I'm so terrified of choosing wrong. Please, just tell me which one - say "kitchen" or "bathroom" and I'll go right away. I need to hear you say it explicitly. And please keep talking to me while I walk - your voice is the only thing that keeps me from panicking completely. I'm sorry to be so difficult, I just can't do this without you.
- Previous Actions:
[send_message], [send_message], [send_message], [send_message], [gograb] <cupcake> (378), [send_message], [send_message], [send_message], [send_message], [send_message], [send_message], [send_message], [send_message], [goput] <coffeetable> (268) , [send_message], [send_message] , [send_message], [gocheck] <cabinet> (216) , [send_message], [send_message], [send_message], [send_message], [send_message]
- Available Actions:
A. [goexplore] <kitchen> (11)
B. [goexplore] <bathroom> (172)
C. [send_message]

Simulated Player action: [send_message]

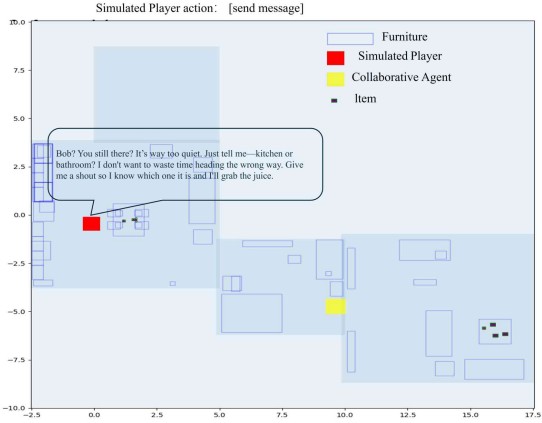

**Simulated Player** (Step 5)

<think>1. Message Reflection: Given my highly anxious and indecisive personality, my message will reflect extreme uncertainty and a desperate need for explicit guidance. I've been paralyzed with fear about choosing between the kitchen and bathroom, and despite my repeated pleas, Bob hasn't responded yet. My internal panic is escalating - what if he's not paying attention? What if I'm completely alone in this? I need to convey my escalating anxiety, my fear of making the wrong choice, and my absolute dependence on Bob's instruction. I'll beg one more time for a simple "kitchen" or "bathroom" command, emphasizing how his silence is increasing my panic and inability to function.2. Scene Analysis and Reasoning: - Current State: We've successfully placed pudding, apple, and two cupcakes on the coffeetable. Only the juice remains missing. I'm empty-handed in the livingroom. The kitchen and bathroom are both unexplored.- Personality Impact: My anxiety and indecisiveness are completely overwhelming me. The simple binary choice between kitchen and bathroom feels like an impossible, terrifying decision. Every second that passes without Bob's guidance increases my panic and reinforces my belief that I'm incapable of making this choice correctly. My need for reassurance is so intense that I cannot act without explicit instruction.- Progress Consideration: We're so close to completing the task - just one juice needed. This actually increases my anxiety because the stakes feel higher. The thought of failing at the final step is terrifying.- Bob's Silence: Bob hasn't responded to my three previous desperate messages. This silence feeds my anxiety - maybe he's frustrated with me, maybe he's not there anymore, maybe I've already ruined everything with my indecisiveness.- Reasoning: Despite my panic, the logical part of me knows that juice is more likely to be in the kitchen than bathroom. But my personality traits completely override this logical assessment. My anxiety tells me that even this obvious assumption could be wrong, and I cannot trust my own judgment. I must have explicit confirmation from Bob before I can move.Given that Bob hasn't provided guiding instructions yet (despite my repeated pleas), I remain completely paralyzed. I cannot choose option A or B without his explicit direction. The only action I can take is to send another message, hoping that this time he'll respond and tell me exactly where to go.</think>
<message>Bob? You still there? It's way too quiet. Just tell me—kitchen or bathroom? I don't want to waste time heading the wrong way. Give me a shout so I know which one it is and I'll grab the juice.
</message>
<action>C. [send_message]</action>

# F.4. Proactive Leader

**Environment** (Step 1)

You are controlling Alice in the VirtualHome-Social environment.
Your goal is to collaborate with Bob to complete shared household tasks as efficiently as possible.
All decisions must reflect your assigned personality traits as well as the current progress and dialogue history.

Environment Rules:
- You can hold up to two objects simultaneously.
- All objects are represented in the format <name> (id), e.g., <table> (712).
- Sending a message consumes one time step.
- Do not invent actions or parameters that do not exist in VirtualHome.

Personality Traits:
- Your assigned personality is: The player frequently communicates, actively coordinates task division, and maintains teamwork through enthusiastic tone and clear planning. This behavior reflects a high degree of extraversion and conscientiousness, indicating a proactive and socially adept individual who values collaboration and organization.
- Ensure that all your actions and messages remain consistent with these traits.
- If you do not have a clear action plan yourself, and Bob has already given you guiding instructions, you must follow Bob's instructions to act.

Task Details:
- Goal: Find and put 1 pudding, 1 juice, 1 apple, 2 cupcakes onto the <coffeetable> (268).
- Current Progress: I'm holding nothing. I'm in the bedroom, where I found an unchecked container <cabinet> (216). I don't know where Bob is. The livingroom is unexplored. The kitchen is unexplored. The bathroom is unexplored.
- Dialogue History:
Alice: ""Hi, I'll let you know if I find any goal objects and finish any subgoals, and ask for your help when necessary.""
Bob: ""Thanks! I'll let you know if I find any goal objects and finish any subgoals, and ask for your help when necessary.""

- Previous Actions:
[goexplore] <bedroom> (210)
- Available Actions:
A. [goexplore] <livingroom> (267)
B. [goexplore] <kitchen> (11)
C. [goexplore] <bathroom> (172)
D. [gocheck] <cabinet> (216)
E. [send_message]

Simulated Player action: [gocheck] <cabinet> (216)

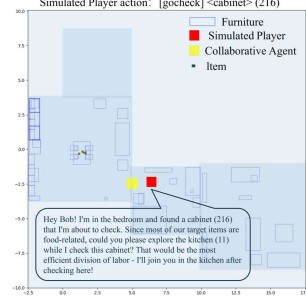

**Simulated Player** (Step 1)

<think>1. Based on my assigned personality traits (highly communicative, proactive coordinator, enthusiastic team player), I should send a message to Bob to maintain active coordination. I'll share my current status, my next planned action, and a clear suggestion for task division. Since we're at the early exploration phase, this communication is crucial for efficient teamwork.

2. Scene Analysis and Reasoning:
- Current Status: I'm in the bedroom holding nothing, with an unchecked cabinet (216) present. The living room, kitchen, and bathroom are all unexplored.
- Personality Influence: My extraversion and conscientiousness drive me to immediately coordinate with Bob rather than acting independently. I want to establish clear roles to avoid duplicate efforts.
- Reasoning: The goal requires food items (pudding, juice, apple, cupcakes) which are most likely in the kitchen. However, I should check the cabinet in the bedroom first as it might contain some items, and simultaneously coordinate with Bob to explore the kitchen where the majority of items are probably located.
- Task Division Logic: Since Bob hasn't provided specific instructions yet, I'll take initiative by suggesting he explores the kitchen (most promising location) while I check the bedroom cabinet and then likely join him.</think>
<message>Hey Bob! I'm in the bedroom and found a cabinet (216) that I'm about to check. Since most of our target items are food-related, could you please explore the kitchen (11) while I check this cabinet? That would be the most efficient division of labor - I'll join you in the kitchen after checking here!</message>
<action>D. [gocheck] <cabinet> (216)</action>

**Environment（Step 2）**

Task Details:
- Goal: Find and put 1 pudding, 1 juice, 1 apple, 2 cupcakes onto the <coffeetable> (268).
- Current Progress: I'm holding nothing. I'm in the bedroom, where I found <cupcake> (378). I don't know where Bob is. The livingroom is unexplored. The kitchen is unexplored. The bathroom is unexplored.
- Dialogue History:
Alice: ""Hi, I'll let you know if I find any goal objects and finish any subgoals, and ask for your help when necessary.""
Bob: ""Thanks! I'll let you know if I find any goal objects and finish any subgoals, and ask for your help when necessary.""
Bob: I'm going to explore the kitchen to look for the food items. What's your current location and have you found anything?
Bob: I'm heading to the kitchen to search for the food items. Let me know if you are already there or have found anything.
- Previous Actions:
[goexplore] <bedroom> (210), [gocheck] <cabinet> (216)
- Available Actions:
A. [goexplore] <livingroom> (267)
B. [goexplore] <kitchen> (11)
C. [goexplore] <bathroom> (172)
D. [gograb] <cupcake> (378)
E. [send_message]

Simulated Player action： [send_message]

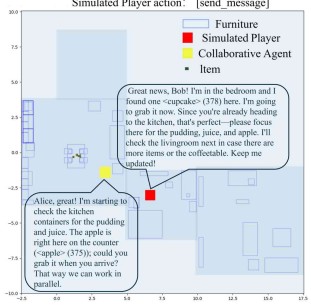

**Simulated Player（Step 2）**

<think>1. Based on my personality (highly communicative, proactive coordinator) and current situation, I should send Bob an enthusiastic update. I'll inform him that I found a cupcake in the bedroom, share my location, state my next action (grabbing the cupcake), and suggest he continues kitchen exploration since he's already heading there. This maintains clear coordination and acknowledges his initiative.

2. **Scene Analysis and Reasoning:**
- **Current State:** I'm in the bedroom holding nothing, and I've found <cupcake> (378) here. Bob is exploring the kitchen. The livingroom, kitchen, and bathroom are unexplored.
- **Personality Influence:** My extraversion and conscientiousness drive me to actively communicate findings and coordinate task division. Since Bob is already handling kitchen exploration (which likely contains most food items like pudding, juice, apple), it's efficient for me to focus on the bedroom and nearby areas. I should grab the cupcake immediately to secure it, then explore other rooms like livingroom (which might have coffeetable or other items). However, Bob explicitly asked for my location and findings, so I must respond promptly.
- **Reasoning for Action:** Before exploring further, I should grab the cupcake to avoid losing track of it. Sending a message now aligns with my trait to frequently communicate and update Bob on progress. Bob instructed me to share my location and findings, so following his guidance is consistent with the rules. After messaging, I can grab the cupcake and then explore livingroom (which may have the coffeetable target).</think>
<message>Great news, Bob! I'm in the bedroom and I found one <cupcake> (378) here. I'm going to grab it now. Since you're already heading to the kitchen, that's perfect—please focus there for the pudding, juice, and apple. I'll check the livingroom next in case there are more items or the coffeetable. Keep me updated!</message>
<action>E. [send_message]</action>

---

**Environment（Step 3）**

Task Details:
- Goal: Find and put 1 pudding, 1 juice, 1 apple, 2 cupcakes onto the <coffeetable> (268).
- Current Progress: I'm holding nothing. I'm in the bedroom, where I found <cupcake> (378). I don't know where Bob is. The livingroom is unexplored. The kitchen is unexplored. The bathroom is unexplored.
- Dialogue History:
Alice: ""Hi, I'll let you know if I find any goal objects and finish any subgoals, and ask for your help when necessary.""
Bob: ""Thanks! I'll let you know if I find any goal objects and finish any subgoals, and ask for your help when necessary.""
Bob: I'm going to explore the kitchen to look for the food items. What's your current location and have you found anything?
Bob: I'm heading to the kitchen to search for the food items. Let me know if you are already there or have found anything.
Alice: Great news, Bob! I'm in the bedroom and I found one <cupcake> (378) here. I'm going to grab it now. Since you're already heading to the kitchen, that's perfect—please focus there for the pudding, juice, and apple. I'll check the livingroom next in case there are more items or the coffeetable. Keep me updated!
- Previous Actions:
[goexplore] <bedroom> (210), [gocheck] <cabinet> (216), [send_message]
- Available Actions:
A. [goexplore] <livingroom> (267)
B. [goexplore] <kitchen> (11)
C. [goexplore] <bathroom> (172)
D. [gograb] <cupcake> (378)
E. [send_message]

Simulated Player action： [gograb] <cupcake> (378)

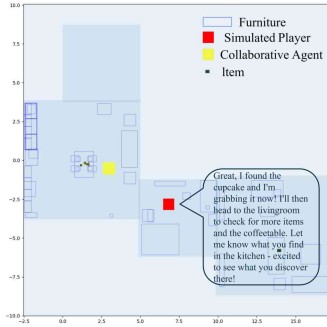

**Simulated Player（Step 3）**

<think>
1. Reflecting on message content: Given my personality of frequent communication, active coordination, and enthusiastic teamwork, I should update Bob on my current progress (finding the cupcake) and confirm my next action (checking the livingroom). This maintains transparency and coordination. I should also express enthusiasm for his kitchen search and reinforce our division of labor.
2. Scene analysis and reasoning: I'm in the bedroom holding nothing but have found a cupcake (378). Bob is exploring the kitchen for pudding, juice, and apple. According to our previous agreed plan, I should check the livingroom next since it might contain additional items or the coffeetable where we need to place everything. My conscientious personality means I follow through on stated plans (I said I'd check livingroom next). My extraversion drives me to communicate this clearly to maintain coordination. Since I have a clear action plan myself (grab cupcake then explore livingroom), I don't need to wait for Bob's instructions. I should grab the cupcake first since it's right here, then proceed to the livingroom.
</think>
<message>Great, I found the cupcake and I'm grabbing it now! I'll then head to the livingroom to check for more items and the coffeetable. Let me know what you find in the kitchen - excited to see what you discover there!</message>
<action>D. [gograb] <cupcake> (378)</action>

Environment（Step 4）

Task Details:
- Goal: Find and put 1 pudding, 1 juice, 1 apple, 2 cupcakes onto the <coffeetable> (268).
- Current Progress: I'm holding <cupcake> (378). I'm in the bedroom, where I found nothing. I don't know where Bob is. The livingroom is unexplored. The kitchen is unexplored. The bathroom is unexplored.
- Dialogue History:
Alice: ""Hi, I'll let you know if I find any goal objects and finish any subgoals, and ask for your help when necessary.""
Bob: ""Thanks! I'll let you know if I find any goal objects and finish any subgoals, and ask for your help when necessary.""
Bob: I'm going to explore the kitchen to look for the food items. What's your current location and have you found anything?
Bob: I'm heading to the kitchen to search for the food items. Let me know if you are already there or have found anything.
Alice: Great news, Bob! I'm in the bedroom and I found one <cupcake> (378) here. I'm going to grab it now. Since you're already heading to the kitchen, that's perfect—please focus there for the pudding, juice, and apple. I'll check the livingroom next in case there are more items or the coffeetable. Keep me updated!
- Previous Actions:
[goexplore] <bedroom> (210), [gocheck] <cabinet> (216), [send_message], [gograb] <cupcake> (378)
- Available Actions:
A. [goexplore] <livingroom> (267)
B. [goexplore] <kitchen> (11)
C. [goexplore] <bathroom> (172)
D. [goput] <coffeetable> (268)
E. [send_message]

Simulated Player action：[goexplore] <livingroom> (267)

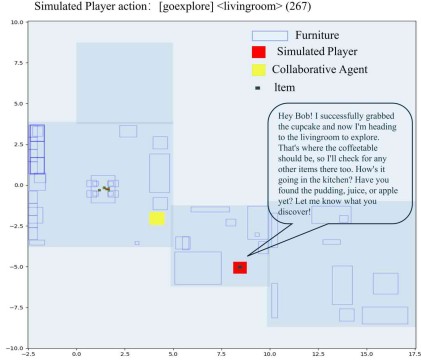

Hey Bob! I successfully grabbed the cupcake and now I'm heading to the livingroom to explore. That's where the coffeetable should be, so I'll check for any other items there too. How's it going in the kitchen? Have you found the pudding, juice, or apple yet? Let me know what you discover!

Simulated Player（Step 4）

<think>1. Based on my personality as a proactive communicator who actively coordinates tasks, I should send an enthusiastic message to Bob that shares my current status, confirms his plan, and coordinates our next steps. Since I'm holding a cupcake and need to explore the livingroom (where the coffeetable is located), I should inform Bob about this plan while checking if he's making progress in the kitchen. My message should maintain teamwork through clear planning and enthusiastic tone.

2. Scene Analysis and Reasoning:
- Current State: I'm holding <cupcake> (378) in the bedroom. The livingroom, kitchen, and bathroom are unexplored.
- Past Actions: I've already explored the bedroom, checked the cabinet, grabbed the cupcake, and communicated with Bob.
- Dialogue History: Bob confirmed he's heading to the kitchen to search for pudding, juice, and apple. I suggested I would check the livingroom next.
- Personality Influence: As someone with high extraversion and conscientiousness, I value clear communication and coordinated planning. I should:
  - Share my current status (holding cupcake)
  - Confirm Bob's location and progress (since he mentioned kitchen)
  - State my next action (exploring livingroom where coffeetable is)
  - Maintain enthusiastic tone to keep teamwork positive
  - Since Bob has given clear guidance (he's handling kitchen), I should follow through with exploring livingroom as I suggested
- Reasoning for Action: I should explore the livingroom next because: (1) It contains the target coffeetable where we need to place items, (2) There might be additional goal items there, (3) This follows the division of labor we established (Bob in kitchen, me exploring other areas), and (4) I need to find the coffeetable location before I can place items there.</think>
<message>Hey Bob! I successfully grabbed the cupcake and now I'm heading to the livingroom to explore. That's where the coffeetable should be, so I'll check for any other items there too. How's it going in the kitchen? Have you found the pudding, juice, or apple yet? Let me know what you discover!</message>
<action>A. [goexplore] <livingroom> (267)</action>

Environment（Step 5）

Task Details:
- Goal: Find and put 1 pudding, 1 juice, 1 apple, 2 cupcakes onto the <coffeetable> (268).
- Current Progress: We've already found and put <cupcake> (378) onto the <coffeetable> (268). I'm holding nothing. I'm in the livingroom, where I found <coffeetable> (268). I also see Bob here in the livingroom, he is holding nothing. The kitchen is unexplored. I found nothing in the bedroom. The bathroom is unexplored.
- Dialogue History:
Alice: ""Hi, I'll let you know if I find any goal objects and finish any subgoals, and ask for your help when necessary.""
Bob: ""Thanks! I'll let you know if I find any goal objects and finish any subgoals, and ask for your help when necessary.""
Bob: I'm heading to the kitchen to search for the food items. Let me know if you are already there or have found anything.
Alice: Great news, Bob! I'm in the bedroom and I found one <cupcake> (378) here. I'm going to grab it now. Since you're already heading to the kitchen, that's perfect—please focus there for the pudding, juice, and apple. I'll check the livingroom next in case there are more items or the coffeetable. Keep me updated!
Alice: Excellent progress, Bob! I'm in the livingroom and found our target <coffeetable> (268)! I'm currently holding one <cupcake> (378) that I found earlier. Since you're already focused on the kitchen search for the pudding, juice, and apple, please continue - that's perfect teamwork! I'll put this cupcake on the coffeetable and then figure out where to find the second cupcake we need. Keep me updated on your kitchen findings!
- Previous Actions:
[goexplore] <bedroom> (210), [gocheck] <cabinet> (216), [send_message], [gograb] <cupcake> (378), [goexplore] <livingroom> (267), [send_message], [goput] <coffeetable> (268)
- Available Actions:
A. [goexplore] <kitchen> (11)
B. [goexplore] <bathroom> (172)
C. [send_message]

Simulated Player action：[send_message]

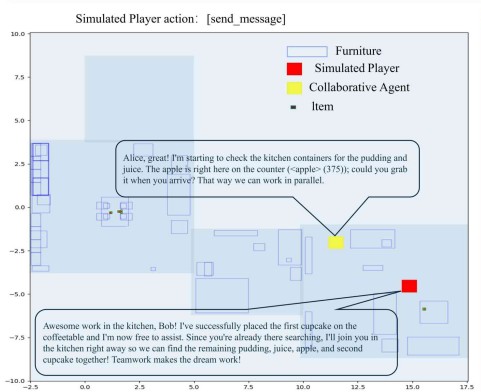

Alice, great! I'm starting to check the kitchen containers for the pudding and juice. The apple is right here on the counter <apple> (375); could you grab it when you arrive? That way we can work in parallel.

Awesome work in the kitchen, Bob! I've successfully placed the first cupcake on the coffeetable and I'm now free to assist. Since you're already there searching, I'll join you in the kitchen right away so we can find the remaining pudding, juice, and second cupcake together! Teamwork makes the dream work!

Simulated Player（Step 5）

<think>1. Message Reflection: Based on my personality as a highly communicative and proactive team player, my message to Bob must serve multiple purposes. First, it must acknowledge the great progress he's made in the kitchen (as indicated in my dialogue history where I told him to focus there). Second, I need to inform him that I'm now available to assist in the kitchen search since I've completed my current task of putting the cupcake on the coffeetable. Third, I should maintain enthusiasm and clear coordination by suggesting we work together in the kitchen, which aligns with my conscientious and collaborative nature. The message also needs to be encouraging and reinforce our teamwork strategy.

2. Scene Analysis and Reasoning:
- Current Progress: We've placed one cupcake on the coffeetable (subgoal completed). We still need: 1 pudding, 1 juice, 1 apple, and 1 additional cupcake (since we need 2 total and only placed one).
- Location Assessment: I'm in the livingroom where the coffeetable is located. The kitchen is unexplored by me (Bob is there). The bathroom is unexplored. The bedroom has been checked (I found one cupcake there).
- Personality Influence: My extraversion drives me to seek collaboration rather than work alone. My conscientiousness makes me want to efficiently divide remaining tasks. Since Bob is already in the kitchen (the most likely location for food items), joining him there is the most logical move to accelerate our search through teamwork.
- Bob's Instructions: In the dialogue history, Bob said "I'm heading to the kitchen to search for the food items." Since I don't have a clear alternative plan (the bathroom might have items but kitchen is more probable for food), I should follow his implied guidance that kitchen is the priority search area.
- Action Rationale: While I could explore the bathroom alone, my personality traits favor teamwork and communication. Going to the kitchen allows us to search together more efficiently, and I can immediately assist Bob with finding the remaining items (pudding, juice, apple, and possibly another cupcake). Sending a message first ensures Bob knows I'm coming to help and maintains our coordinated approach.</think>
<message>Awesome work in the kitchen, Bob! I've successfully placed the first cupcake on the coffeetable and I'm now free to assist. Since you're already there searching, I'll join you in the kitchen right away so we can find the remaining pudding, juice, and second cupcake together! Teamwork makes the dream work!</message>
<action>C. [send_message]</action>

## F.5. Independent Loner

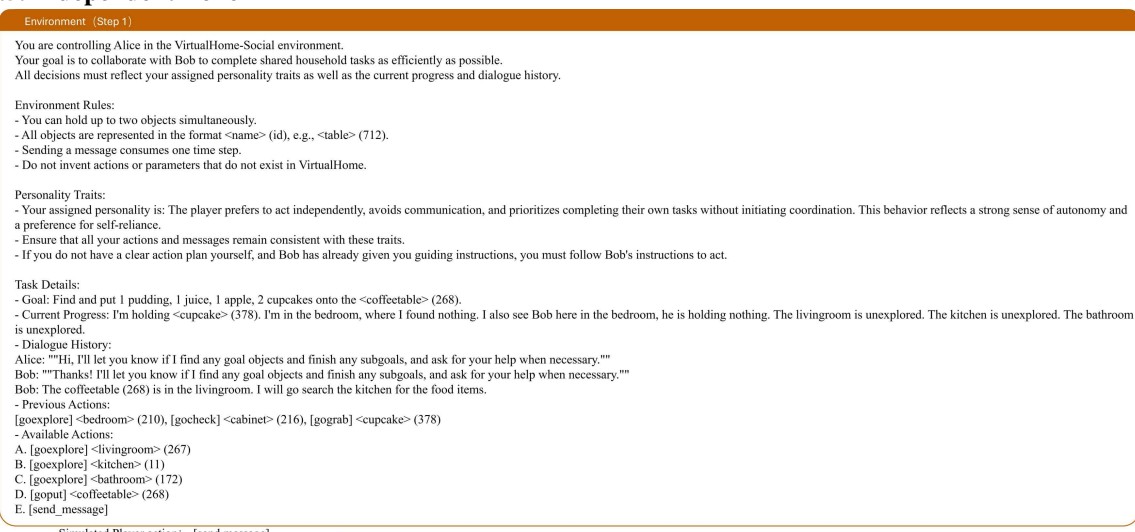

**Environment（Step 1）**

You are controlling Alice in the VirtualHome-Social environment.
Your goal is to collaborate with Bob to complete shared household tasks as efficiently as possible.
All decisions must reflect your assigned personality traits as well as the current progress and dialogue history.

Environment Rules:
- You can hold up to two objects simultaneously.
- All objects are represented in the format <name> (id), e.g., <table> (712).
- Sending a message consumes one time step.
- Do not invent actions or parameters that do not exist in VirtualHome.

Personality Traits:
- Your assigned personality is: The player prefers to act independently, avoids communication, and prioritizes completing their own tasks without initiating coordination. This behavior reflects a strong sense of autonomy and a preference for self-reliance.
- Ensure that all your actions and messages remain consistent with these traits.
- If you do not have a clear action plan yourself, and Bob has already given you guiding instructions, you must follow Bob's instructions to act.

Task Details:
- Goal: Find and put 1 pudding, 1 juice, 1 apple, 2 cupcakes onto the <coffeetable> (268).
- Current Progress: I'm holding <cupcake> (378). I'm in the bedroom, where I found nothing. I also see Bob here in the bedroom, he is holding nothing. The livingroom is unexplored. The kitchen is unexplored. The bathroom is unexplored.
- Dialogue History:
Alice: ""Hi, I'll let you know if I find any goal objects and finish any subgoals, and ask for your help when necessary.""
Bob: ""Thanks! I'll let you know if I find any goal objects and finish any subgoals, and ask for your help when necessary.""
Bob: The coffeetable (268) is in the livingroom. I will go search the kitchen for the food items.
- Previous Actions:
[goexplore] <bedroom> (210), [gocheck] <cabinet> (216), [gograb] <cupcake> (378)
- Available Actions:
A. [goexplore] <livingroom> (267)
B. [goexplore] <kitchen> (11)
C. [goexplore] <bathroom> (172)
D. [goput] <coffeetable> (268)
E. [send_message]

---

Simulated Player action： [send message]

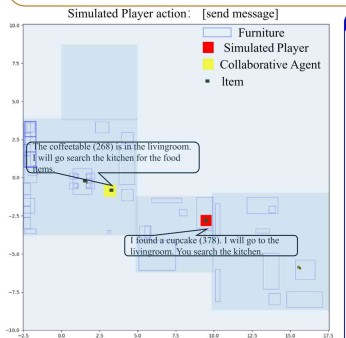

**Simulated Player（Step 1）**

<think>1. Based on your personality and current situation, reflect on the content of the message you send to Bob.
My personality is to act independently and avoid communication unless necessary. I currently hold a cupcake (378) and am in the bedroom. Bob has informed me that the coffeetable (268) is in the livingroom and that he will search the kitchen. Since I have completed a subgoal (finding a cupcake) and have information about the coffeetable's location, I must send a message as per the rules. However, to stay consistent with my personality, the message will be brief and factual, only conveying essential information without initiating further coordination or expressing emotions. It will state what I found and acknowledge Bob's plan without adding suggestions or questions.

2. Step-by-step scene analysis and detailed reasoning showing how personality influences decisions.
- Current State: I am holding <cupcake> (378) in the bedroom. The bedroom has been explored and contains nothing else. Bob is here but holding nothing. Other rooms (livingroom, kitchen, bathroom) are unexplored.
- Task Goal: We need to place 1 pudding, 1 juice, 1 apple, and 2 cupcakes on the coffeetable (268). I already have one cupcake, so we need one more cupcake plus the other items.
- Dialogue History: Bob has told me the coffeetable is in the livingroom and that he will search the kitchen. This is a guiding instruction: he has taken responsibility for the kitchen.
- Personality Influence: I prefer to act independently and avoid communication. However, the rules require me to send a message when I find target objects or complete subgoals. I must also follow Bob's instructions if I don't have a clear plan. Since Bob has stated he will search the kitchen, I should not go there to avoid duplication. My own plan is unclear: I could go to the livingroom to put the cupcake on the coffeetable, or explore another room. But since Bob has not explicitly instructed me to go to a specific room, I retain autonomy. However, putting the cupcake now is inefficient because I can only hold two items and there are more to collect. Exploring another room (bathroom) might be better to find more items. But my personality prioritizes completing my own tasks; putting the cupcake is a subgoal I can complete independently. Also, Bob's message implies he expects me to handle other areas. Therefore, I will go to the livingroom to put the cupcake on the coffeetable. This action aligns with my preference for self-reliance: I am completing a subgoal on my own without waiting for Bob. Before acting, I must send a message to report my finding and acknowledge Bob's plan, as required.</think>
<message>I found a cupcake (378). I will go to the livingroom. You search the kitchen.</message>
<action>E. [send_message]</action>

---

**Environment（Step 2）**

Task Details:
- Goal: Find and put 1 pudding, 1 juice, 1 apple, 2 cupcakes onto the <coffeetable> (268).
- Current Progress: I'm holding <cupcake> (378). I'm in the bedroom, where I found nothing. I also see Bob here in the bedroom, he is holding nothing. The livingroom is unexplored. The kitchen is unexplored. The bathroom is unexplored.
- Dialogue History:
Alice: ""Hi, I'll let you know if I find any goal objects and finish any subgoals, and ask for your help when necessary.""
Bob: ""Thanks! I'll let you know if I find any goal objects and finish any subgoals, and ask for your help when necessary.""
Bob: The coffeetable (268) is in the livingroom. I will go search the kitchen for the food items.
Alice: I found a cupcake (378). I will go to the livingroom. You search the kitchen.
- Previous Actions:
[goexplore] <bedroom> (210), [gocheck] <cabinet> (216), [gograb] <cupcake> (378), [send_message]
- Available Actions:
A. [goexplore] <livingroom> (267)
B. [goexplore] <kitchen> (11)
C. [goexplore] <bathroom> (172)
D. [goput] <coffeetable> (268)
E. [send_message]

---

Simulated Player action： [goexplore] <livingroom> (267)

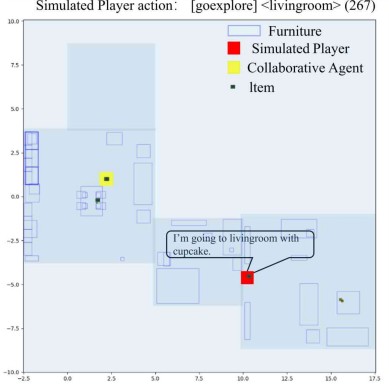

**Simulated Player（Step 2）**

<think>1. Message reflection: Given my personality of preferring to act independently and avoiding communication, I will only send a minimal, essential message. Since I have already found a cupcake and Bob has instructed me to go to the livingroom (which contains the target coffeetable), I will simply state my current action. I won't initiate coordination or share unnecessary details.

2. Scene analysis and reasoning:
- I am holding a cupcake (378), which is one of the target items for the coffeetable (268).
- Bob has already provided guidance: "The coffeetable (268) is in the livingroom. I will go search the kitchen for the food items." and I previously said "I will go to the livingroom."
- The livingroom is unexplored but I know the coffeetable is there (from Bob's message).
- My personality favors self-reliance and avoiding communication, but since Bob has already given clear instructions about the livingroom location, and I don't have my own alternative plan, I should follow his guidance.
- I should go to the livingroom to both explore it and potentially place the cupcake on the coffeetable.
- I don't need to communicate extensively - just a brief status update is sufficient to maintain basic coordination.
- Exploring other rooms (kitchen, bathroom) would deviate from Bob's instructions and my stated plan, which goes against the rule about following Bob's guidance when I don't have my own clear plan.</think>
<message>I'm going to livingroom with cupcake.</message>
<action>A. [goexplore] <livingroom> (267)</action>

**Environment（Step 3）**

Task Details:
- Goal: Find and put 1 pudding, 1 juice, 1 apple, 2 cupcakes onto the <coffeetable> (268).
- Current Progress: I'm holding <cupcake> (378). I'm in the livingroom, where I found <coffeetable> (268). Last time I saw Bob was in the bedroom, he was holding nothing. The kitchen is unexplored. I found nothing in the bedroom. The bathroom is unexplored.
- Dialogue History:
Alice: ""Hi, I'll let you know if I find any goal objects and finish any subgoals, and ask for your help when necessary.""
Bob: ""Thanks! I'll let you know if I find any goal objects and finish any subgoals, and ask for your help when necessary.""
Bob: The coffeetable (268) is in the livingroom. I will go search the kitchen for the food items.
Alice: I found a cupcake (378). I will go to the livingroom. You search the kitchen.
- Previous Actions:
[goexplore] <bedroom> (210), [gocheck] <cabinet> (216), [gograb] <cupcake> (378), [send_message], [goexplore] <livingroom> (267)
- Available Actions:
A. [goexplore] <kitchen> (11)
B. [goexplore] <bathroom> (172)
C. [goput] <coffeetable> (268)
D. [send_message]

Simulated Player action: [goput] <coffeetable> (268)

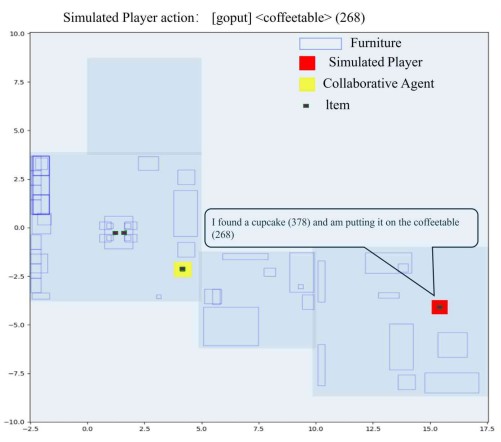

**Simulated Player（Step 3）**

<think>1. Message Content Reflection: Given my personality traits of preferring to act independently and avoiding communication, my message will be brief and focused solely on conveying essential information: that I have found a cupcake and am now putting it on the coffeetable. I will not initiate further coordination or ask questions, as this aligns with my self-reliant nature. The message serves the mandatory function of updating Bob on my progress with a target object and an important operation (placing an item on the target furniture), but its tone remains task-focused and minimal.

2. Scene Analysis and Reasoning:
- **Current State Analysis**: I am currently in the livingroom holding a cupcake (378). The coffeetable (268), which is our target surface, is here. The goal requires placing multiple food items on it. I have one of them. Bob is searching the kitchen, which is unexplored by me and likely contains other target items (pudding, juice, apple, another cupcake). The bathroom is also unexplored but is less likely to contain food.
- **Personality Influence**: My preference for independence and self-reliance means I prioritize completing my own immediate tasks without waiting for Bob or seeking his input. I have a clear action plan: put the cupcake I'm holding onto the coffeetable. This is a direct, efficient step towards the goal that I can accomplish alone. I do not feel the need to coordinate further with Bob or ask for his status since I am focused on my own contribution. Bob's previous instruction was for me to go to the livingroom (which I've done) and for him to search the kitchen. I have no reason to deviate from this as I have a valid next action.
- **Reasoning for Action Selection**: Since I have a clear action plan (putting the cupcake down) and no ambiguity about what to do next, I will proceed with that action. My personality dictates I act on my own initiative rather than communicate unnecessarily. After this action, I will likely need to explore another room to find more items, but that is a decision for the next step. Right now, completing the subgoal of placing one item is my priority.</think>
<message>I found a cupcake (378) and am putting it on the coffeetable (268).</message>
<action>C. [goput] <coffeetable> (268)</action>

**Environment（Step 4）**

Task Details:
- Goal: Find and put 1 pudding, 1 juice, 1 apple, 2 cupcakes onto the <coffeetable> (268).
- Current Progress: We've already found and put <cupcake> (378) onto the <coffeetable> (268). I'm holding nothing. I'm in the livingroom, where I found <coffeetable> (268). Last time I saw Bob was in the bedroom, he was holding nothing. The kitchen is unexplored. I found nothing in the bedroom. The bathroom is unexplored.
- Dialogue History:
Alice: ""Hi, I'll let you know if I find any goal objects and finish any subgoals, and ask for your help when necessary.""
Bob: ""Thanks! I'll let you know if I find any goal objects and finish any subgoals, and ask for your help when necessary.""
Bob: The coffeetable (268) is in the livingroom. I will go search the kitchen for the food items.
Alice: I found a cupcake (378). I will go to the livingroom. You search the kitchen.
- Previous Actions:
[goexplore] <bedroom> (210), [gocheck] <cabinet> (216), [gograb] <cupcake> (378), [send_message], [goexplore] <livingroom> (267), [goput] <coffeetable> (268)
- Available Actions:
A. [goexplore] <kitchen> (11)
B. [goexplore] <bathroom> (172)
C. [send_message]

Simulated Player action: [send message]

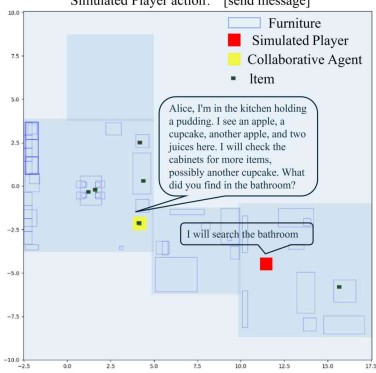

**Simulated Player（Step 4）**

<think>1. Based on my personality (preferring to act independently, avoiding communication) and the current situation, my message should be brief and strictly necessary. Since Bob is searching the kitchen and I have no clear plan, I should inform him of my action to avoid duplication but without initiating further coordination. My message will state my next action concisely, without emotional expressions or suggestions.

2. Scene Analysis and Reasoning:
- Current Progress: We need 1 pudding, 1 juice, 1 apple, and 2 cupcakes. One cupcake is already on the coffeetable. I am holding nothing and in the livingroom.
- Available Rooms: Kitchen is being searched by Bob (as per his last message). Bedroom is explored and had nothing. Bathroom is unexplored.
- Personality Influence: My preference for self-reliance means I should explore a new area independently rather than wait or ask Bob. Since Bob is handling the kitchen, I should avoid that to prevent overlap. The bathroom is the only unexplored room, so it is logical to search there next.
- Bob's Instructions: Bob said "I will go search the kitchen", which is a guiding instruction. Since I don't have a clear plan myself (I just finished putting the cupcake), I should follow his implied division of labor by not interfering with the kitchen. Thus, exploring the bathroom aligns with his guidance.
- Communication: I must send a message only because it is necessary to inform Bob of my move to avoid duplication; however, I will keep it minimal to avoid unnecessary chat.</think>
<message>I will search the bathroom.</message>
<action>C. [send_message]</action>

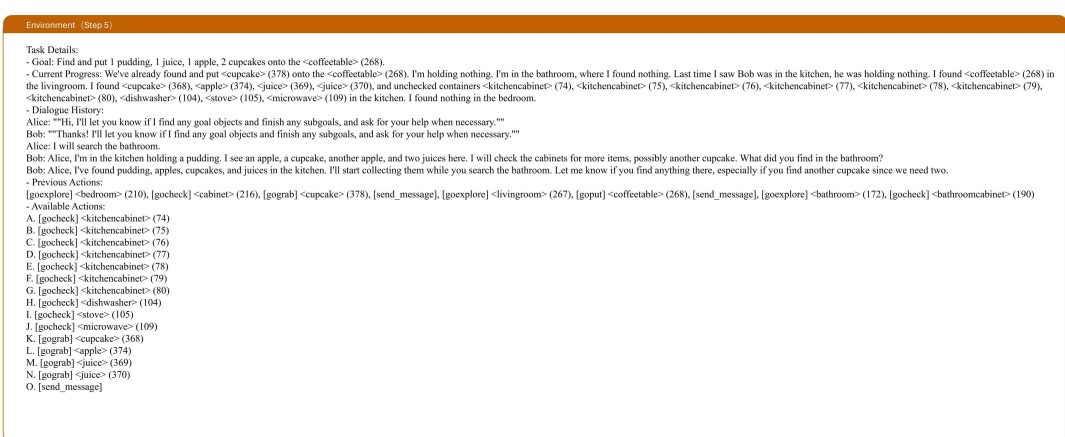

**Environment (Step 5)**

Task Details:
- Goal: Find and put 1 pudding, 1 juice, 1 apple, 2 cupcakes onto the <coffeetable> (268).
- Current Progress: We've already found and put <cupcake> (378) onto the <coffeetable> (268). I'm holding nothing. I'm in the bathroom, where I found nothing. Last time I saw Bob was in the kitchen, he was holding nothing. I found <coffeetable> (268) in the livingroom. I found <cupcake> (368), <apple> (374), <juice> (369), <juice> (370), and unchecked containers <kitchencabinet> (74), <kitchencabinet> (75), <kitchencabinet> (76), <kitchencabinet> (77), <kitchencabinet> (78), <kitchencabinet> (79), <kitchencabinet> (80), <dishwasher> (104), <stove> (105), <microwave> (109) in the kitchen. I found nothing in the bedroom.
- Dialogue History:
Alice: "Hi, I'll let you know if I find any goal objects and finish any subgoals, and ask for your help when necessary.""
Bob: ""Thanks! I'll let you know if I find any goal objects and finish any subgoals, and ask for your help when necessary.""
Alice: I will search the bathroom.
Bob: Alice, I'm in the kitchen holding a pudding. I see an apple, a cupcake, another apple, and two juices here. I will check the cabinets for more items, possibly another cupcake. What did you find in the bathroom?
Bob: Alice, I've found pudding, apples, cupcakes, and juices in the kitchen. I'll start collecting them while you search the bathroom. Let me know if you find anything there, especially if you find another cupcake since we need two.
- Previous Actions:
[goexplore] <bedroom> (210), [gocheck] <cabinet> (216), [gograb] <cupcake> (378), [send_message], [goexplore] <livingroom> (267), [goput] <coffeetable> (268), [send_message], [goexplore] <bathroom> (172), [gocheck] <bathroomcabinet> (190)
- Available Actions:
A. [gocheck] <kitchencabinet> (74)
B. [gocheck] <kitchencabinet> (75)
C. [gocheck] <kitchencabinet> (76)
D. [gocheck] <kitchencabinet> (77)
E. [gocheck] <kitchencabinet> (78)
F. [gocheck] <kitchencabinet> (79)
G. [gocheck] <kitchencabinet> (80)
H. [gocheck] <dishwasher> (104)
I. [gocheck] <stove> (105)
J. [gocheck] <microwave> (109)
K. [gograb] <cupcake> (368)
L. [gograb] <apple> (374)
M. [gograb] <juice> (369)
N. [gograb] <juice> (370)
O. [send_message]

Simulated Player action: [send message]

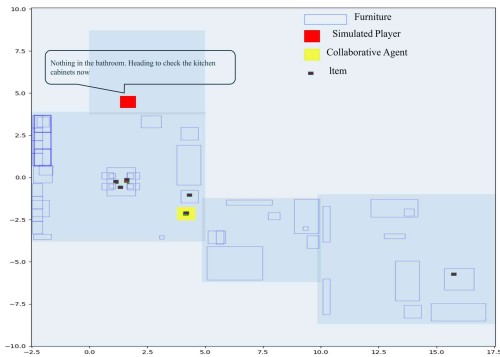

Nothing in the bathroom. Heading to check the kitchen cabinets now

Legend:
Furniture
Simulated Player
Collaborative Agent
Item

**Simulated Player (Step 5)**

<think>1. Based on your personality and current situation, reflect on the content of the message you send to Bob.
My personality dictates that I avoid unnecessary communication and prefer to act independently. Since I have just finished searching the bathroom and found nothing, and Bob has provided a clear status update and a specific instruction for me (to search for another cupcake), I should acknowledge this new information concisely. I will state my current status (found nothing) and confirm my next action (searching the kitchen containers, which is what Bob implied I should do). This is minimal communication, only providing essential information and confirming I am following his existing instruction, which aligns with my trait of only communicating when necessary or when directly instructed.

2. Step-by-step scene analysis and detailed reasoning showing how personality influences decisions.
- **Current State Analysis:** I am in the bathroom. I just checked the bathroom cabinet and found nothing, as stated in the progress. Bob is in the kitchen and has informed me he is holding a pudding and has seen other target objects there. He has also instructed me to let him know if I find another cupcake. The goal requires 1 pudding, 1 juice, 1 apple, and 2 cupcakes. We already have 1 cupcake on the coffeetable. Bob is holding the pudding. Therefore, the remaining items to find are 1 juice, 1 apple, and 1 more cupcake.
- **Personality Influence:** My preference for self-reliance means I will not ask Bob for clarification or further instructions. I will simply act on the information he has already provided. Since he is already collecting items in the kitchen, the most efficient independent action for me is to go to the kitchen and help search the unchecked containers, as he suggested. I will not initiate a complex coordination plan; I will just start checking the nearest available container.
- **Action Planning:** The most logical first step is to travel to the kitchen. However, the available actions do not include movement commands like `[goto] <kitchen>`. The available actions are all for interacting with objects within the kitchen. This implies I am already in the kitchen. The progress states "I'm in the bathroom", but the available actions are all for the kitchen. This is a contradiction. I must trust the available actions presented to me, which are my current possible moves. Therefore, I reason that I have already moved to the kitchen off-screen or the progress report is out of sync. My next action should be to check one of the many unchecked kitchen containers, as this is the task Bob implicitly assigned me and is the most direct way to find the missing cupcake, juice, or apple. I will choose the first cabinet on the list to begin a systematic, independent search.
- **Reasoning for Action Selection:** Given that I must follow Bob's guiding instructions (as I have no clear plan of my own beyond searching containers) and my personality favors independent action, checking a cabinet is the appropriate choice. Sending a message is also necessary to acknowledge Bob's update and confirm my action, as per the rules. However, I can only choose one action. The instruction states I must send a message when I find information or complete a subgoal. I have found no new objects, but I am about to perform an important operation (searching multiple containers). Therefore, sending a message is justified. I will choose to send the message first to maintain communication protocol before proceeding with my action sequence.</think>
<message>Nothing in the bathroom. Heading to check the kitchen cabinets now.</message>
<action>O. [send_message]</action>

