# OpenReview forum: "CollabBench: Benchmarking and Unleashing Collaborative Ability of LLMs with Diverse Players via Proactive Engagement"
_ICML.cc/2026/Conference — ICML 2026 regular_

### Official Review · Reviewer_eLVQ · 2026-02-16

**Soundness:** 1
**Presentation:** 3
**Significance:** 2
**Originality:** 2
**Overall Recommendation:** 3
**Confidence:** 3

**Summary:**

This paper presents **CollabBench**, a benchmark and interactive environment for evaluating and training LLM-based agents in proactive collaboration settings with heterogeneous partners. The core contribution lies in constructing a structured profile simulation pipeline that generates diverse behavioral types, coupled with a high-fidelity personality–behavior modeling framework to ensure consistency and realism. An interaction-based filtering mechanism is further introduced to improve profile quality and coherence.

In addition to the environment design, the paper proposes a comprehensive evaluation protocol to assess collaboration performance across multiple dimensions. Experimental results demonstrate the effectiveness of the benchmark and reveal important limitations of current LLM-based agents in proactive coordination and adaptation to diverse collaborators. Overall, the work provides both a practical testbed and diagnostic insights for advancing collaborative agent training.

**Compliance With Llm Reviewing Policy:**

Affirmed.

**Final Justification:**

The paper introduces a benchmark targeting proactive collaboration between LLM-based agents and heterogeneous partners. The authors have addressed most of my concerns during the rebuttal, including motivation and experimental results. However, the reward model is heuristic as the core contribution is the utilization of the LLM-as-judge, and the judgment consistency and prompting strategy are not direct evidence for the correctness of the LLM-generated judgment. Without theoretical guarantees, it is unconvincing that LLM can consistently generate accurate scores in all settings, and it lacks evidence that the reward model can avoid learning toward “looking parseable” collaboration (reward hacking). Therefore, I lean toward rejecting this paper.

**Key Questions For Authors:**

1. Why is the LLM-generated score accurate? If not, how can these biased signals not affect the performance of profile filtering, agentic training, and the evaluation protocol?
2. What is the more realistic application beyond online games of human-agent collaboration, and how does this approach perform?
3. Why is it necessary to use small language models instead of large models (e.g. GPT or Deepseek) as they outperform the trained 7B models? It further leads to a question: what is the performance of varying mode size?

**Limitations:**

Yes. The limitation is discussed in the paper.

**Strengths And Weaknesses:**

Strengths:
1. The paper introduces a well-motivated and timely benchmark targeting proactive collaboration between LLM-based agents and heterogeneous partners, a setting that is underexplored compared to single-agent or homogeneous multi-agent benchmarks.
2. The proposed profile simulation pipeline is thoughtfully designed, incorporating explicit personality–behavior mappings and interaction-based filtering to enhance realism and behavioral consistency. This structured construction improves over naïve role prompting or randomly generated profiles.
3. The benchmark includes a relatively comprehensive evaluation protocol that measures multiple dimensions of collaboration quality, going beyond task success to assess coordination and adaptability.

Weaknesses:
1. The motivation and the three identified challenges, while intuitively reasonable, are not sufficiently supported by empirical evidence. The paper would benefit from preliminary experiments, ablation studies, or real-world case analyses to substantiate why these challenges are critical and inadequately addressed by existing benchmarks.
2. Several key components—including the personality filtering score, collaborative training reward, and evaluation metrics—rely heavily on an LLM-as-judge paradigm. This introduces potential biases, such as model preference bias, prompt sensitivity, and limited calibration. The absence of human evaluation or cross-model validation weakens the reliability and objectivity of the reported results.
3. The necessity of focusing on small language models (e.g., 7B-scale models) is not clearly justified. Experimental results indicate that larger proprietary models (e.g., GPT- or DeepSeek-class models) substantially outperform the trained smaller models, raising questions about the practical significance of the proposed collaborative training framework. It remains unclear whether the gains stem from the training methodology itself or simply reflect model capacity differences.
4. The paper could further clarify how well the benchmark generalizes beyond the simulated profiles. Since the environment is synthetically constructed, there is a risk that agents overfit to the profile generation mechanism rather than learning broadly transferable collaborative behaviors.

---

> ### Author Rebuttal · Authors · 2026-03-31
>
> Thanks for your advice. ***All the additional experiments are shown `in this link`*** (**[https://anonymous.4open.science/r/CollabBench/pictures/To_Reviewer_eLVQ.md](https://anonymous.4open.science/r/CollabBench/pictures/To_Reviewer_eLVQ.md)**).
> >W1
>
> In fact, we provide extensive experimental evidence to support the three challenges we raise:
>
> + **Preliminary experiments (why these challenges are critical):** Figure 1 and Appendix A show that existing benchmarks focus on agent-agent efficiency and fail to capture human-agent collaboration. And this highlights the core challenges: **simulating humans effectively, training agents to collaborate efficiently with humans, and constructing benchmarks for human-agent interaction**. Based on this, we further provide extensive experimental evidence to demonstrate why addressing them is inherently difficult.
> + **Human simulation difficulty**: Section 5.2.2 shows that prior methods produce uniform trajectories poorly reflecting real behavior. Our approach achieves higher diversity, persona robustness, and more realistic behavior distributions.
>
> + **Training difficulty for human collaboration**: Sections 5.2.3 and Appendix D.4 demonstrate that traditional efficiency-only training improves task completion but harms user experience, as agents ignore human collaborators. Our method enhances affective capabilities, highlighting the limits of purely objective rewards.
>
> + **Evaluation validation**: Section 5.2.4 shows that in real human interactions, our trained agent outperforms baseline and efficiency-focused agents in both efficiency and affective metrics. Results align with benchmark scores (Table 1, Figure 5), confirming our framework better captures human-agent collaboration quality.
>
> >W2&Q1
>
> We agree that evaluating alignment between human judgment and our LLM Judge, as well as cross-model validation, is crucial. To this end, we conducted experiments in CWAH across five representative player types and tasks in `link`. Results show that **strong open-source models** like DeepSeek-R1 and top-tier closed models such as GPT-5.2 **align well with human experts**. **Smaller models** (e.g., Qwen-72B) **exhibit instability on challenging affective metrics, and performance of 32B and 14B models drops sharply**. Based on this, we select DeepSeek-R1 and adopt a penalty-based LLM Judge [2] to ensure both evaluation accuracy and interpretability.
> >W3&Q3
>
> + **Importance of training smaller models:** Smaller models remain crucial in the LLM community, as large open-source models increase cost and latency, and proprietary models can be expensive. Efficient smaller models are essential for real-time applications like online games.
>
> + **Performance relative to large models:** While our trained small models cannot surpass 10×–1000× larger models, they achieve competitive results in some places. In CWAH-MultiPlayer, they match or exceed large models in Efficiency and Affective metrics (Trustfulness, Empathy), with far lower inference overhead than models like Deepseek-R1, which shows the potential of small models. **More importantly, our goal is not a SOTA method, but to reveal and evaluate affective capabilities rarely addressed in existing benchmarks, providing a robust environment for future improvements.**
>
> + **Training gains vs. base model capacity:** Table 1 and Appendix D.2.2 show that, compared with base model, our models **improve Efficiency by 19.5% and Affective metrics by 24.4%**. Section 5.2.4 confirms enhanced real-world collaborative capabilities, indicating gains stem from our training methodology rather than capacity alone.
>
> + **Performance of varying model size:** Appendix D.4.1 and Table 10 show performance scales with model size, confirmed by small-model results (`link`); our method still **boosts Efficiency and Affective metrics across all scales**, improving even low-ability models.
>
> >W4&Q2
>
> + **Real-World User Case Study:** Section 5.2.4 shows experiments with real human participants, demonstrating that our models perform effectively in genuine human collaboration scenarios and generalize beyond simulated profiles.
>
> + **Generalization between our two tasks:**  We further transfer model trained on CWAH to COOK for evaluation in `link`. Compared to base model, it **improves both Efficiency and Affective metrics**, especially Affective, suggesting cross-environment generalization.
>
> + **Generalization in more realistic application beyond online games:** Further experiments on realistic collaborative writing and math tasks [1] are provided in `link`. Although task gains are limited across domains, improved interactivity shows our models’ **affective capabilities generalize** beyond the training environment.
> ```
> [1] COLLABLLM: From Passive Responders to Active Collaborators. ICML'25
> [2] CoSER: A Comprehensive Literary Dataset and Framework for Training and Evaluating LLM Role-Playing and Persona Simulation. ICML'25
> ```

---

> > ### Author Rebuttal · Reviewer_eLVQ · 2026-04-01
> >
> > Thank you for the detailed rebuttal. As the method is not sound enough due to the heristic reward model by the LLM judge and lack of theoretical guarantees, I will maintain my score, but am willing to be convinced otherwise.

---

> > > ### Author Response · Authors · 2026-04-02
> > >
> > > We sincerely thank the reviewer for this critical point. We fully understand the concern about using an LLM Judge as a reward signal.
> > >
> > > We would like to clarify that our goal is to provide a practical and principled proxy for subjective human judgments, which are difficult to formalize. In such settings, LLM-based judges are widely adopted as a prevailing and necessary approach [1–5]. Importantly, **rather than relying on a free-form heuristic, our LLM Judge is a structured and constrained approximation of human evaluation**. To ensure its reliability and validity, we implemented several specific designs:
> > >
> > > + **Hybrid Reward Design (Objective + Subjective):** As detailed in Section 4.2.2, we do not rely solely on the LLM Judge, but integrates objective and subjective signals. At the trajectory level, we utilize game scores as efficiency signals. At the step level, we use format rewards to ensure LLM responses contain sufficient information. Furthermore, communication frequency serves as an auxiliary reward to provide agents with adequate communication volume, compensating for the LLM’s limitations in frequency statistics, allowing it to focus on its strength: qualitative evaluation of affective interaction quality.
> > >
> > > + **Rigorous Rubric-Based Prompting:** To minimize subjective variance, our LLM Judge operates on highly detailed, rubric-based prompts (Appendix F), which proves to be effective in prior study [5]. We explicitly define the behavioral patterns corresponding to each score level and provide rich, human-verified examples. Meanwhile, in profile filtering and evaluation, we use a penalty-based scheme where the LLM Judge detects deviations and scores are computed via a strict formula instead of direct holistic scoring. This constrained approach has been proven to reduce direct LLM scoring limitations and improves accuracy [2].
> > >
> > > We would like to emphasize that **the LLM Judge reward constitutes only a small component of the overall reward design and is controlled through weighting to mitigate potential reward hacking**. Combined with rigorous rubric-based prompting, described above, this design effectively reduces risks.
> > >
> > > Beyond design, in such subjective settings, while our method may not provide formal theoretical guarantees, we adopt empirical alignment with human judgment as a practical and meaningful criterion for validation:
> > >
> > > + Ablation studies and case analyses (Sec. 5.2.3, App. D.4) show that incorporating LLM Judge rewards significantly **enhances affective capabilities** compared to objective rewards alone. Models trained with **only objective rewards tend to ignore human interactions and focus on independent task completion**, whereas LLM Judge improves communication, emotional awareness, and efficiency.
> > >
> > > + User study (Sec. 5.2.4) demonstrate **effective performance in genuine human collaboration scenarios**.
> > >
> > > + Due to the high cost of human-aligned rewards during training, we use an LLM Judge and validate it via LLM judge–human alignment experiments across five player types in CWAH tasks. Results show **high consistency with human ratings** (Spearman rank correlation), supporting the LLM Judge as a reliable proxy for human-aligned evaluation.
> > >
> > > | Model Type    | Personality Filtering | Collaborative Evaluation |              |          |
> > > | ------------- | --------------------- | ------------------------ | ------------ | -------- |
> > > |               | Consistency           | Helpfulness              | Trustfulness | Empathy  |
> > > | GPT-5.2       | **0.81**              | 0.65                     | **0.83**     | **0.87** |
> > > | DeepSeek-V3.1 | 0.79                  | **0.75**                 | 0.77         | 0.74     |
> > > | Qwen2.5-72B   | 0.76                  | 0.63                     | 0.63         | 0.78     |
> > >
> > > Finally, we highlight that our main contribution is **modeling human-like collaboration through a high-fidelity, personality-driven simulation pipeline and introducing a benchmark to evaluate affective capabilities—largely overlooked in existing benchmarks yet crucial for human-agent collaboration.** We think the value of our training and evaluation pipeline lies in providing a meaningful reference for future research and optimization, rather than proposing a specific SOTA method or reward design. This contribution is far more impactful than simply presenting a new training method.
> > >
> > > We sincerely appreciate your feedback and will further add the discussion in the revised version.
> > >
> > > ```
> > > [1] COLLABLLM: From Passive Responders to Active Collaborators. ICML'25
> > > [2] CoSER: A Comprehensive Literary Dataset and Framework for Training and Evaluating LLM Role-Playing and Persona Simulation. ICML'25
> > > [3] R-CHAR: A Metacognition-Driven Framework for Role-Playing in Large Language Models. EMNLP'25
> > > [4] RLVER: Reinforcement Learning with Verifiable Emotion Rewards for Empathetic Agents. ICLR'26
> > > [5] Rubrics as Rewards: Reinforcement Learning Beyond Verifiable Domains. ICLR'26
> > > ```

---

### Official Review · Reviewer_8Jwd · 2026-03-06

**Soundness:** 3
**Presentation:** 2
**Significance:** 2
**Originality:** 3
**Overall Recommendation:** 4
**Confidence:** 4

**Summary:**

The paper proposes CollabBench, a benchmark for training and evaluating LLM agents in multi-agent cooperative game contexts. CollabBench includes a diverse player simulation pipeline based on the Big 5 personality traits, and an agentic training pipeline. The agentic training pipeline is based on reinforcement learning and introduces a reward for task completion and another for affective alignment. CollabBench also introduces two new environments: CWAH-MultiPlayer and CookMultiPlayer. The paper demonstrates that training with diverse personality-driven teammates and with a combination of task-based and affective rewards leads to better overall collaboration.

**Compliance With Llm Reviewing Policy:**

Affirmed.

**Final Justification:**

My final assessment of the paper remains similar to the original assessment. The authors were able to clarify potential areas of concerns  that I raised.

**Key Questions For Authors:**

Most of my questions are above, but I have one additional question here:
- In CWAH-Multiplayer, why does the oracle (GPT 5.2) achieve improved task score metrics over the non-oracle GPT 5.2, even though the only difference is that it knows the ground truth personas?

**Limitations:**

Yes, described in Appendix E.

**Strengths And Weaknesses:**

*Strengths*
- **Importance of problem:** the benchmark addresses coordination within interactive grounded behavioral scenarios, which is both interesting and distinct from conversation-level reasoning tasks.

- **Grounded framework**: design decisions overall make sense and are grounded in existing cooperative frameworks and personality theories from cognitive science and psychology (e.g. use of big 5 personality traits, decomposing collaboration into taskwork and teamwork to motivate reward design).

- **Thorough experimental results**: the authors validate the persona generation process, the effectiveness of the agentic training, presents a human user study.

*Weaknesses:*

- **Why is this a benchmark?** To me, the paper is written as a methods paper. A benchmark paper should systematically compare existing methods using standardized datasets and metrics, and/or provide a unified implementation. Yet, the only methods benchmarked are baseline LLMs, and their proposed RL method. Can the authors explain why this is a benchmark paper rather than a methods paper?

- **Clarity**: some of the paper’s key details are hard to understand. The metrics and figures are typically not sufficiently explained, and some aspects of the method are not sufficiently explained.

    - Diverse persona generation: following sentence was unclear to me: “Furthermore, each trait is grounded in expert-validated gameplay logic, allowing personality differences to consistently manifest as observable behavioral patterns during interaction.” ([pdf](zotero://open-pdf/library/items/52T9FXET?page=3)) (pg 3).  How exactly was each trait grounded in the expert gameplay logic?

    - Environment - The task design is one of the key parts of a benchmark paper, but is glossed over. Did the authors entirely take the tasks from existing work, or do any original work? What is the difference in coordination challenge presented by CWAH and Overcooked and why do we need both?

    - Lack of measures of confidence / variability: Table 1 reports “std” for task score only.  Does std correspond to std dev or std error? It’s also not sufficient to report measure of confidence for only one metric. Where are the measures of confidence for all other results?

    - Table 1: what is the meaning of the #tokens(k) metric? They appear to be ratios rather than token counts.

    - Figs 4 and 9c: why are the score values so low? In particular, I am familiar with the Overcooked tasks, which provide a reward of 20 for delivering one dish. So why is the provided score range -1.4 to 1.8?

    - In Table 1, to contextualize performance can the authors report an upper bound on task scores from humans and RL agents?


- **Validation of Persona Generation**: There is insufficient work validating the persona generation pipeline.

    - In addition to comparing the trajectory diversity of CollabBench and CoELA / ProAgent, can the authors also compare diversity of CollabBench personas to diversity induced by the baseline LLMs (i.e. DeepSeek, GPT, Qwen)?

    - What is the relationship between personality-behavior mapping and Interaction-based filtering? Why do we need the former? Can the authors provide ablations validating the necessity of both?


*Minor Comments*

- **Where are the limitations of existing methods?** The paper claims to provide key collaborative limitations of existing methods (last sentence of abstract) but these appear to be buried in the extensive Appendix. There should be a discussion in the main paper.

- **Lacking contextualization** w.r.t. Concordia and existing work in AHT / ZSC. Baseline results with ZSC methods are reported in Tables 4 and 5 in the Appendix, but the  literature is not discussed in the main paper’s introduction or related work.

- **Typos**: throughout the paper, spaces are missing between some words (e.g. personalitybehavior, humanLLM) and there are some typos (Cwah-MultiPlayer on line 383, right)

---

> ### Author Rebuttal · Authors · 2026-03-31
>
> Thanks for your advice. ***All the additional experiments are shown `in this link`*** (**[https://anonymous.4open.science/r/CollabBench/pictures/To_Reviewer_8Jwd.md](https://anonymous.4open.science/r/CollabBench/pictures/To_Reviewer_8Jwd.md)**).
> >W1
>
> We carefully considered whether our work is a benchmark or a method. With LLM research increasingly emphasizing data, environments, and evaluation, the line between methods and benchmarks has blurred. Earlier strong methods [1] were tied to valuable environments and newly constructred benchmarks, while context and harness engineering show that components beyond core training can be equally important.
>
> From this perspective, our main contribution is systematically evaluating efficiency and affective capabilities in human–agent collaboration, which is underexplored yet closely aligned with real-world interactions. By revealing current LLM limitations and providing a scalable environment, we position our work as a benchmark, offering a foundation for future advancements rather than a standalone method.
> >W2
>
> 1. As described in Sec. 4.1.1, we model personality as observable behaviors using expert-informed mappings from the Big Five to task-specific strategies. For example, detail-oriented players check containers, while collaborative players frequently synchronize with teammates. For high neuroticism in CWAH-Multiplayer, we model behaviors such as anxiety-driven repeated confirmation, hesitation, and inefficient actions.
>
> 2.
> + **Originality:** Our approach introduces several key improvements:
>      + **Human-agent collaboration:** While based on multi-agent environments, we replace one agent with a high-quality persona to capture real-world human-like collaboration.
>      + **Affective evaluation:** We propose a framework to assess affective capabilities, addressing gaps in prior benchmarks; Figure 1 and Appendix A show it is more challenging, exposing limitations and guiding improvements.
>      + **Environment extension:** Overcooked originally lacks communication. We incorporate it as a core component, enabling LLMs to fully utilize collaborative abilities.
>
> + **CWAH vs. Overcooked:** The key difference is game pace, affecting decision speed and task difficulty. As analyzed in Section 5.2.1, CWAH’s slower pace allows more time per step, while Overcooked demands rapid, time-sensitive decisions. Table 1 supports these distinctions, showing the importance of including both environments for a comprehensive benchmark.
>
>  Thank you for your insightful question. It highlighted the value of a more detailed introduction and comparison of these two tasks, which we will include in the revised version.
>
> 3. Std. corresponds to std dev. Std. of other metircs are provided in `link`.
>
> 4. Tokens (k) denotes the average number of tokens generated by the model at each step during gameplay, measured in thousands (k).
>
> 5. We select five representative player types, compute mean and standard deviation across tasks, and normalize scores for color mapping.
>
> 6. In Table 1, we report a GPT5.2-based oracle to indicate the performance ceiling on our benchmark. The upper bounds for human participants and our RL agent are also presented and analyzed in Section 5.2.4 and Figure 6, covering all score dimensions.
>
> >W3
>
> 1. Using CWAH-Multiplayer, we further encode trajectory segments to measure diversity for CollabBench and baselines in `link`. Results show CollabBench achieves higher diversity, driven by explicit personality–behavior mappings, while baselines rely mainly on generation randomness.
>
> 2. We encode and cluster trajectories to extract behavior patterns and form personality–behavior profiles, then refine them via interaction-based filtering. Consistency metrics in Section 5.2.2, Appendix C.2 confirm improved diversity, robustness, and realism after filtering.
>
> >W4
>
> 1. The term “method” here includes both existing agent paradigms and training strategies. Detailed analyses are in Sections 5.2.1, 5.2.3, and 5.2.4, with finer case studies in Appendix D. We agree that incorporating more appendix discussion into the main text would improve clarity and will do so in the revised version.
>
> 2&3. Thank you for your suggestions. We will add discussions of ZSC and AHT in the related work section and correct the typos in the revised version.
> >Q1
>
> For a powerful model like GPT5.2, providing an accurate partner persona helps anticipate the partner’s next actions, enabling proactive coordination. This relates to Theory of Mind (ToM) [2]. The effect is especially pronounced in slower-paced environments like CWAH, where the model has sufficient time and information to reason, significantly improving task performance. This aligns with prior findings and suggests that richer partner information enhances collaborative reasoning.
> ```
> [1] COLLABLLM: From Passive Responders to Active Collaborators. ICML'25
> [2] Theory of Mind for Multi-Agent Collaboration via Large Language Models. EMNLP'23
> ```

---

> > ### Author Rebuttal · Reviewer_8Jwd · 2026-04-04
> >
> > Thank you for addressing my concerns. I encourage the authors to incorporate the results presented here and improve the discussion/framing of their benchmark based on the above points. I will keep my score since it already indicates accept.

---

> > > ### Author Response · Authors · 2026-04-04
> > >
> > > Thank you for your encouraging feedback and for recognizing the value of our work. We are pleased that our responses have addressed your concerns. We will include the results and analyses discussed here in the revised version and further improve the presentation of our work. Thank you again for your constructive comments and support.

---

### Official Review · Reviewer_h4en · 2026-03-11

**Soundness:** 2
**Presentation:** 3
**Significance:** 3
**Originality:** 3
**Overall Recommendation:** 4
**Confidence:** 3

**Summary:**

This paper introduces CollabBench, a comprehensive benchmarking platform designed to systematically evaluate the collaborative capabilities of LLMs in multi-agent environments.

**Compliance With Llm Reviewing Policy:**

Affirmed.

**Final Justification:**

My main concerns have been addressed.

**Key Questions For Authors:**

- I hope the authors could provide evaluation results for open-source LLMs with smaller parameter scales.

- A deeper analysis of these results is essential, as it would more effectively demonstrate the true value of this work.

- How does this benchmark distinguish between genuine collaboration and a scenario where a single powerful 'dominant' model completes the task primarily through its own capabilities, effectively overshadowing the other weaker models?

- Could the authors provide more details regarding prompt sensitivity? Specifically, how significantly does the collaboration success rate fluctuate when there are slight variations in the initial system prompts or role descriptions?

**Limitations:**

yes

**Strengths And Weaknesses:**

*Strengths*

- As the LLM evaluation landscape shifts toward multi-agent systems, providing a standardized and rigorous benchmark for collaboration and communication efficiency is vital. This paper fills a significant gap in the field.
- The proposed evaluation metrics offer a nuanced and methodologically sound assessment of model collaboration by incorporating factors such as communication overhead, conflict resolution success rates, and contribution balance.
- The paper is well-structured and highly readable. The clear categorization of collaborative tasks and the establishment of a robust leaderboard provide essential guidance for future researchers to build upon.

*Weaknesses*

- The evaluation results appear heavily skewed toward a few top-tier proprietary models. From my perspective, the paper lacks a sufficient analysis of smaller-scale, open-source LLMs.
- I think that many of the tasks in the paper's benchmark tests seem to be directly adapted from existing single-agent datasets, rather than being originally designed multi-agent challenges. This may fail to fully capture the complexity and asymmetry inherent in real-world human-like collaboration.
- While the quantitative results are satisfactory, the paper lacks an in-depth qualitative error analysis. It does not adequately explore the root causes of collaborative failures, such as hallucination loops, stubbornness, or context window overflow caused by protracted inter-agent debates.

---

> ### Author Rebuttal · Authors · 2026-03-31
>
> Thanks for your advice. ***All the additional experiments are shown `in this link`*** (**[https://anonymous.4open.science/r/CollabBench/pictures/To_Reviewer_h4en.md](https://anonymous.4open.science/r/CollabBench/pictures/To_Reviewer_h4en.md)**).
> >W1&Q1
>
> We appreciate your suggestion and agree that testing smaller-scale open-source models is important. In Appendix D.4.1 and Table 10, we further analyze Qwen2.5-3B-Instruct and Qwen3-8B before and after training. These results demonstrate our benchmark’s performance on small-scale models and **its meaningful differentiation across parameter scales and model types**.
>
> To further address your concern, we also evaluate additional small open-source models such as LLaMA, for five player types in CWAH tasks (`link`). Results show larger models enhance collaboration via better reasoning. Compared with LLaMA, interaction-aligned Qwen models excel, though reasoning-intensive models like Qwen3-8B may overthink, slightly reducing affective performance and highlighting the efficiency–affective trade-off (Appendix D.4.1). But our method **boosts Efficiency and Affective metrics for even model with low initial ability**.
> >W2
>
> We apologize for any misunderstanding. Our two game environments are originally designed for multi-agent cooperation [1,2] rather than derived from single-agent tasks. Original environments focus on agent-agent cooperation and do not capture human-agent interaction. **Our main contribution is actually modeling the complexity and asymmetry of real-world human-like collaboration via a high-fidelity pipeline simulating diverse player personalities**. To evaluate the often-overlooked affective capabilities, we introduce a **comprehensive benchmark**. As shown in Figure 1 and Appendix A, our personality-driven setting raises task difficulty, highlighting models’ current limitations and guiding optimization.
>
> While real-world human-AI collaboration poses additional challenges, we focus on efficiency and affectiveness as key interaction indicators, providing valuable insights for developing collaborative LLM-based agents.
> >W3&Q2
>
> First, we clarify that **our work focuses on human–agent interaction, not multi-agent systems**. Our setting involves a single human simulator and agent, so multi-agent issues (e.g., hallucination loops, stubbornness, context overflow) from prolonged inter-agent debates do not necessarily apply.
>
> For our settings, we actually provide a comprehensive, multi-perspective analysis of interactions between current models and human agents. In **Appendix D.4**, we present representative failure cases to illustrate limitations of existing models and training paradigms. Additionally, **Appendix G** includes complete trajectory examples of various personality types interacting with our trained agent, enabling future researchers to further explore current model limitations.
> >Q3
>
> Thank you for your question. **Our benchmark is explicitly designed to address this concern**. As you noted, a single powerful “dominant” model that completes tasks primarily through its own capabilities may achieve high efficiency, but this often comes at the cost of severely degraded affective performance. This trade-off is directly shown in **our ablation study in Section 5.2.3**. We trained a variant resembling the “dominant” model you describe, removing step-level affective rewards so the model focused solely on efficiency (i.e., game score). As expected, this agent achieved very high efficiency but performed poorly on affective metrics. **Appendix D.4.1** shows that optimizing purely for efficiency eliminates communication ability. This highlights the distinction between dominant and collaborative models. **A dominant model disregards human-related interactions and focuses solely on completing tasks independently, whereas a collaborative model communicates effectively, maintains the teammate’s emotional experience, and achieves high efficiency simultaneously**.
>
> **Our benchmark effectively distinguishes these two behaviors**. **Section 5.2.3** shows clear differences between model variants in efficiency and affective scores, while **Appendix D.4** presents failure cases with low affective scores, providing intuitive evidence of the benchmark’s ability to capture meaningful distinctions in genuine collaboration.
>
> >Q4
>
> Thank you for the suggestion. In forced_coordination task of COOK that require intensive collaboration, we apply limited prompt variations (e.g., synonyms, word order, slight behavior changes) while keeping goals fixed in `link`. Minor changes cause small fluctuations, while larger shifts (e.g., shifting role preferences toward over-preparing ingredients like onions) can significantly reduce performance (Reduced from 6 to 4 orders due to over-preparation).
> ```
> [1] ProAgent: Building Proactive Cooperative Agents with Large Language Models. AAAI'24
> [2] Building Cooperative Embodied Agents Modularly with Large Language Models. ICLR'24
> ```

---

> > ### Author Rebuttal · Reviewer_h4en · 2026-04-02
> >
> > Thank you for your comments, and my main concerns have been addressed. I have increased my score and have no further questions.

---

> > > ### Author Response · Authors · 2026-04-02
> > >
> > > Thank you for your positive feedback, recognizing the value of our work, and raising the score. We are glad that our responses have addressed your concerns, and we will carefully incorporate all your suggestions into the revised version. Thanks.

---

### Official Review · Reviewer_37WJ · 2026-03-11

**Soundness:** 3
**Presentation:** 4
**Significance:** 3
**Originality:** 2
**Overall Recommendation:** 4
**Confidence:** 4

**Summary:**

This paper introduces CollabBench (CB), a benchmark for evaluating and training LLM-based agents in cooperative game environments with diverse human-like partners. The work addresses three core challenges: simulating personality-diverse players, training agents that balance efficiency with affective responsiveness, and evaluating collaboration beyond task metrics. The authors propose a Diverse Player Profile Simulation pipeline grounded in Big Five personality theory, a Collaborative Agentic Training paradigm using hybrid rewards (trajectory-level efficiency + step-level affective), and a dual-metric evaluation protocol (CB-Efficiency and CB-Affective) tested with two game environments. There is also a user evaluation for sim-trained agents with real humans to test generalization.

**Compliance With Llm Reviewing Policy:**

Affirmed.

**Key Questions For Authors:**

- The authors may want to cite research from embodied AI/ human-robot collaboration, which also focuses on collaborating with diverse humans in grounded manner. Particularly, COOPERA (Neurips 2025: https://openreview.net/forum?id=wOSZVnYH5w) also simulates diverse humans driven by big-5 personality traits for grounded and longitudinal human-robot collaboration, while PARTNR (https://arxiv.org/abs/2411.00081) benchmarks a heterogeneous collaboration setting in which two agents cooperate to achieve a shared goal in embodied environments.
- It is not clear to me which LLM model is used as the judge. If the same model is used as both the player simulator and the reward judge, there could be shared inductive biases that inflate affective scores. I am also curious about how strong of a model the judge needs to be for effective eval and training. How sensitive are CB-Affective scores to the choice of judge LLM? Have the authors tested judge agreement across different model families?
- It would also be great to motivate why GIGPO instead of simpler alternatives like reward shaping on top of GRPO. Likewise, it would be good to elaborate on why the authors choose the specific testbeds for this work.
- Is it possible to report standard deviation on affectiveness scores as well in Tab. 1?
- Is there a pareto-frontier of efficiency vs. affectiveness? Is there a way to analyze what efficiency is being "sacrificed" for affective gains?
- CollabBench diversity is measured using the same embedding model (Qwen3-Embedding-4B) used to construct profiles. Does this potential bias the diversity metrics?

**Limitations:**

I enjoyed reading Appendix E. I would have liked to learn more about how the work could generalize/apply to other long horizon, realistic collaborative tasks. Likewise, it would be good for the authors to state the limitations of the userstudy such as no control of learning effects, small sample size and limited demohgraphics and potential implications of that on the takeways. Also, please see weakness described in the section above.

**Strengths And Weaknesses:**

### Strengths:
- I appreciated the well-motivated problem framing and push to go beyond task performance for training and evaluating human-AI collaboration.
- It was also good to get affirmation from user study that affective rewards were indeed useful when it came to real human-AI collaboration.
- The experimental insights and analysis (sec 5.2.1) were interesting and useful for future research in this space.

### Weakness:
- Anthropomorphic simulation: While I appreciated authors attempts to filter profiles and evaluate whether agents followed their personas using different consistency metrics, it seems to me that the absolute consistency scores remain relatively low, suggesting substantial remaining inconsistency even in filtered profiles. It would be good if the authors could be more upfront about how this potentially affects the validity of both the training signal and the evaluation protocol simultaneously.
- LLM-Judge calibration/human validation: I understand why an LLM is used for evaluating affectiveness. However, it is not clear to me whether the judge is a valid proxy for human judgment at all, which is a prerequisite for using it as a training signal. It would be useful to correlate llm judge performance with that of human to ensure that the judges are well-aligned and calibrated with humans.
- Limited generalization: The training setup perturbs test profiles to create training profiles (Section D.1.1), but the perturbation strategy is not clearly defined. This makes it unclear whether the trained model is genuinely generalizing and how strong is this generalization. Also, CB-Affective is used both as a training reward and as an evaluation metric, isn't that a big problem/circular? I'd love to hear more about this from the authors in the rebuttal.
- No SFT baseline: The generated trajectory data from player simulation could straightforwardly be used for SFT. Without this comparison, it's unclear whether RL is necessary or whether imitation learning would achieve similar gains at lower cost.

My overall take is that the framework, eval and training setup, and resultant experiments are interesting but many limitations remain. It might be useful to atleast provide SFT baseline, and some more discussion on generalization issues.

---

> ### Author Rebuttal · Authors · 2026-03-31
>
> Thanks for your advice. ***All the additional experiments are shown `in this link`*** (**[https://anonymous.4open.science/r/CollabBench/pictures/To_Reviewer_37WJ.md](https://anonymous.4open.science/r/CollabBench/pictures/To_Reviewer_37WJ.md)**).
> >W1
>
> We analyze consistency $ S^{\text{ALL}} $ under different filtering ratios (`link`): it rises sharply then plateaus. With Top 30 profiles / 50%, removing inconsistent profiles improves data quality, while further filtering yields limited gains. Our method remains robust due to large-scale data learning.
> >W2
>
> Providing reliable affective rewards is challenging; we use LLM judges [1,2] with several designs to ensure validity.
>
> + **Hybrid Reward Design:** As in Sec. 2.2, we combine trajectory-level efficiency with step-level rewards, using communication frequency as an auxiliary signal to offset LLM limits and enhance affective interaction quality.
> + **Rubric-Based Prompting:** The LLM Judge uses detailed rubric-based prompts (Appendix F) with defined behavioral patterns and human-verified examples, improving scoring accuracy [3].
>
> Empirical results (Section 5.2.3 and Appendix D.4) show that LLM Judge affective rewards substantially improve affective capabilities compared to using only objective rewards.
>
> + **Human Proxy:** We conduct LLM judge–human alignment experiments across five player types in CWAH tasks. The LLM shows high consistency with human ratings, supporting its feasibility.
>
> >W3
>
> + **Perturbation Strategy:** Behavioral, contextual, and personality perturbations create distributional variation while keeping training and testing semantically related. Stable performance on unseen settings (Tab. 1) demonstrates strong generalization.
>
> + **Homogeneity between Reward and Metric:** We acknowledge some overlap between the reward and evaluation metric but took steps to avoid full metric leakage:
>
>     + **Scoring Criteria:** During training, only the three major dimensions and a few representative scoring rules are provided, not the full evaluation rubrics.
>     + **Granularity of Scoring:** The LLM Judge produces a single step-level score (0–1), less detailed than the full trajectory-level metric. This balances training cost and avoids complete overlap.
>
>     Despite some overlap, our main contribution is a robust environment and scoring framework for training collaborative LLMs. A meaningful benchmark guides evaluation and model optimization, linking training rewards with evaluation criteria.
> >W4
>
> Additional SFT results (`link`) show modest gains, but limited generalization prevents it from outperforming RL, highlighting RL’s necessity and effectiveness.
> >Q1
>
> We appreciate these valuable references, which are highly relevant and will be cited and discussed in the revision to inform future work.
> >Q2
>
> Using DeepSeek-V3.1 as the judge, we analyze five player types across CWAH tasks in `link`, measuring rank consistency via Spearman correlation.
>
> + **Sensitivity on Model Size:** Larger models (Qwen2.5-72B, 32B, 14B) show higher alignment with human ratings, suggesting stronger reasoning better captures affective cues.
> + **Sensitivity on Model Family:** Alignment with human ratings across Qwen2.5-72B, DeepSeek-V3.1, and GPT-5.2 is high, showing CB-Affective is robust to judge model choice.
>
> >Q3
>
> + **GIGPO vs. GRPO:** GRPO relies on sparse trajectory-level rewards and struggles with efficiency and affective signals, while GIGPO integrates trajectory- and step-level rewards via hierarchical advantage estimation and verl-agent, providing a robust, extensible framework.
> + **Testbed Choice:** Cooperative games like CWAH and COOK require proactive multi-turn interaction, unlike single-agent benchmarks (e.g., ALFWorld), making them well-suited for CollabBench.
>
> >Q4
>
> Std. on affective metric are provided in `link`.
> >Q5
>
> This is insightful. Our hybrid reward balances efficiency and affectiveness: removing affective rewards boosts efficiency but reduces satisfaction, while removing efficiency rewards leads to unproductive communication.
>
> The trade-off is controlled by $\omega$ in Eq. 5, set to 1 for equal weighting of efficiency and interaction quality. A full hyperparameter study of $\omega$ is valuable and planned for the revised version.
> >Q6
>
> We use BGE-M3 to encode trajectories for diversity in CWAH (`link`); while embeddings may affect absolute values, our conclusions rely on relative differences, with limited self-reinforcing bias since embeddings are not optimized.
> >L1
>
> We report long-horizon collaborative results in writing and math [2] in `link`. While performance gains are limited, Interactivity shows clear affective improvement.
>
> The experiments and analyses discussed above will be clarified in the revision.
> ```
> [1] RLVER: Reinforcement Learning with Verifiable Emotion Rewards for Empathetic Agents. ICLR'26
> [2] COLLABLLM: From Passive Responders to Active Collaborators. ICML'25
> [3] Rubrics as Rewards: Reinforcement Learning Beyond Verifiable Domains. ICLR'26
> ```

---

> > ### Author Rebuttal · Reviewer_37WJ · 2026-04-06
> >
> > Thanks for answering my questions and providing additional experiments. I remain positive about this paper.

---

> > > ### Author Response · Authors · 2026-04-06
> > >
> > > Thank you for your positive feedback and for recognizing the value of our work. We are very pleased to have addressed your questions and will incorporate all suggestions into the revised version based on your comments. Thanks.

---

### Decision · Program_Chairs · 2026-04-30

**Decision:**

Accept (regular)

**Comment:**

Summary: CollabBench tries to create a benchmark to evaluate LLM agents on cooperative game environments with partners that exhibit various types of personas based on the Big Five theory in psychology. It proposes metrics related to both effectiveness at finishing tasks as well as how well they collaborate and shows that trained models are better at both metrics than baselines.

Pros:
- Human-agent collaboration is a very timely topic and one without too much contemporary work that has seen widespread adoption
- The ablation studies and in particular the persona diversity validation help disentangle the various sources of complexity in the benchmark
- The pipeline for creating the agent personas in grounded in (psych) literature and seemingly thus more amenable to successful reproduction

Cons:
- LLM as judge variance is not well discussed and though correlation to human evaluations is provided, this weakens the claims
- The writing makes disentangling the method vs the benchmark in terms of the overall contribution and situation wrt the literature